# LogiConBench: Benchmarking Logical Consistencies of LLMs

**Zheng Chen**[1,*]  **Chuan Zhou**[2,*]  **Fengxiang Cheng**[3]  **Yip Tin Po**[1]  **Fenrong Liu**[4]
**Yisen Wang**[2,5]  **Jiajun Chai**[6]  **Xiaohan Wang**[6]  **Guojun Yin**[6]  **Wei Lin**[6]
**Bo Li**[1,†]  **Haoxuan Li**[7,†]  **Zhouchen Lin**[2,5,†]

[1]Computer Science and Engineering, Hong Kong University of Science and Technology
[2]State Key Lab of General AI, School of Intelligence Science and Technology, Peking University
[3]Institute for Logic, Language and Computation, University of Amsterdam
[4]The Tsinghua-UvA JRC for Logic, Department of Philosophy, Tsinghua University
[5]Institute for Artificial Intelligence, Peking University    [6]Meituan
[7]Center for Data Science, Peking University
`zchenin@connect.ust.hk, zhouchuancn@pku.edu.cn,`
`bli@cse.ust.hk, hxli@stu.pku.edu.cn, zlin@pku.edu.cn`

## Abstract

Logical consistency, the requirement that statements remain non-contradictory under logical rules, is fundamental for trustworthy reasoning, yet current LLMs often fail to maintain it even on simple inference tasks. Existing benchmarks for LLM logical consistency are not scalable, not diverse, and not challenging, with state-of-the-art models already surpassing 95% accuracy. **LogiConBench** is the first benchmark that (1) generates unlimited logical rule combinations with precise labels, (2) provides controllable-depth graphs with explicit reasoning paths, and (3) remains challenging for state-of-the-art LLMs. To achieve this, LogiConBench automatically generates **logical graphs** where nodes represent symbolic propositions and edges denote reasoning relations. From these graphs, it samples lists of propositions, extracts **reasoning paths**, determines all **consistent label lists**, and translates them into diverse natural language expressions. While we release a 280K-sample corpus in this work, the framework can be scaled to generate unlimited data. To strengthen its evaluative significance, we evaluate 14 frontier LLMs on three tasks with varying difficulty levels, and find that the **Enumerative task** remains extremely challenging, with the best exact accuracy as only 34%. Our code and data are available at `https://github.com/Bellafc/LogiConBench.git`.

## 1 Introduction

Logical reasoning is a category of capabilities essential to trustworthy AI (Xu et al., 2024; 2025a). Among these, logical **consistency** refers to the property that a set of statements does not contain contradictions under logical rules (Huang & Chang, 2023; Liu et al., 2025). Maintaining model consistency is central for reliable conclusions and reasoning (Cheng et al., 2025). However, recent studies show that LLMs frequently generate self-contradictory reasoning or outputs, even for simple inference tasks (Calanzone et al., 2025; Ghosh et al., 2025; Paleka et al., 2025; Song et al., 2025). For example, suppose sentence *P* entails *H*, and *H* entails *Z*; by transitivity, one should infer that *P* entails *Z*. However, models may produce inconsistent judgments which violate logical principles (Li et al., 2019). Such inconsistencies worsen the local inference and can also propagate through reasoning chains, which ultimately disrupt the overall reasoning process.

Existing efforts benchmarking the logical consistency of LLMs can be summarized as follows. **BeliefBank** (Kassner et al., 2021) generates constraints over entities based on a commonsense knowledge base. **EntailmentBank** (Dalvi et al., 2021) provides multi-step entailment trees, where answers

---

*Equal contribution.
†Bo Li, Haoxuan Li, and Zhouchen Lin are the corresponding authors.

Table 1: Comparison of logical consistency datasets in terms of size, depth, operators, reasoning path availability, scalability, and rule count. The detailed explanations can be found in Appendix B.

| Dataset | Size | Depth | Operators | Reasoning Path | Scalability | # of Rule |
|---|---|---|---|---|---|---|
| BeliefBank | 12,525 | 1 | $\rightarrow, \neg, \leftrightarrow$ | No | No | 2 |
| EntailmentBank | 1,840 | avg. 6 | $\rightarrow$ | **Yes** | No | 6 |
| LFC | ~2,000 | up to 4 | $\wedge, \vee, \rightarrow, \neg, \leftrightarrow$ | No | No | 9 |
| Set-LConVQA & Set-SNLI | 13,779 | up to 5 | $\wedge, \vee, \rightarrow, \neg, \leftrightarrow$ | No | No | 51 |
| **LogiconBench** | **280K** | **32** | $\wedge, \vee, \rightarrow, \neg, \leftrightarrow$ | **Yes** | **Yes** | **280K** |

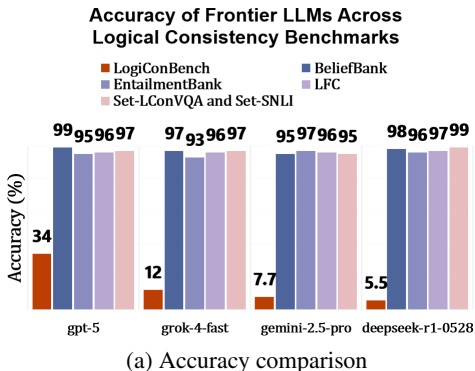

(a) Accuracy comparison

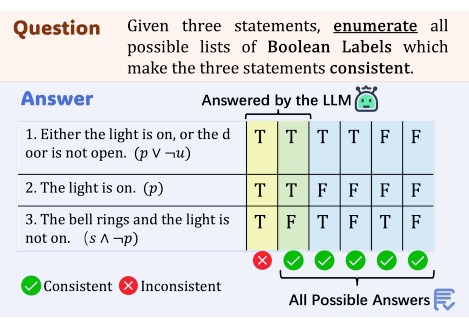

(b) Enumerative task

Figure 1: (a) Accuracy of frontier LLMs on LogiConBench vs. existing benchmarks. Existing benchmarks are saturated, while LogiConBench remains challenging and discriminative. (b) Illustration of the enumerative task. Given three statements, the LLM must list all consistent label assignments, but often outputs incomplete or incorrect lists.

are supported by explicit reasoning steps. **LFC datasets** (Ghosh et al., 2025) are built by transforming knowledge graphs into logical fact-checking queries. **Set-LConVQA and Set-SNLI** (Song et al., 2025) extend Visual Question Answering (VQA) dataset and Natural Language Inference (NLI) tasks into a set-level format and check for consistency across multiple sentences.

Despite their contributions, these datasets have several limitations. As shown in Table 1, **first**, their **sizes and rule counts** remain relatively small, which becomes inadequate for realistic logical reasoning scenarios. **Secondly**, their **depths are shallow and the reasoning paths are absent**, preventing the models from fully capturing multi-step reasoning chains. **Thirdly**, all of them are either human-written or derived from existing datasets, **highly limiting their scalability**. **Finally**, our empirical results in Figure 1a show that the most advanced LLMs such as `gpt-5` and `grok-4-fast` already achieve above 95% accuracy on these datasets, which suggests that they are **no longer sufficiently challenging** to be used as logical consistency benchmarks for frontier LLMs.

To address the aforementioned limitations, we propose a framework named **LogiConBench** that automatically constructs complex and large-scale datasets for logical consistency evaluation. **LogiConBench** first **generates logical graphs**, which can expand indefinitely and record reasoning relations, where nodes represent symbolic propositions and edges denote reasoning relations. From the graph, we **randomly sample** lists of propositions, extract their shortest reasoning paths, and propagate Boolean labels along the edges according to logical rules, by which we collect all truth-value assignments that keep the sampled propositions **consistent**. To enhance structural variety, we further apply symbolic rewriting techniques to **produce logically equivalent formulas**. Finally, propositions are translated into **natural language** through templates and lexical substitutions.

Through this construction, LogiConBench directly overcomes the above limitations. First, it can generate **unlimited logical rule combinations** automatically with precise consistency labels, which supports scaling up and covering a wider variety of logical rules. Second, graphs with controllable depths provide **explicit reasoning paths for multi-step inference**. Finally, our benchmark

remains **challenging** for state-of-the-art LLMs, as shown in Figure 1, which shows its significance for benchmarking the consistencies of current frontier models.

To systematically evaluate logical consistency reasoning, we design two primary benchmarking tasks: ***Discriminative task***, determining whether a given Boolean Label list can lead to contradiction for the given statements, and ***Enumerative task***, enumerating all consistent Boolean Label assignments for the given statements. To capture performance across different levels of difficulty, we further introduce variants for the three tasks. Our large-scale experiments with 14 frontier closed and open-source models reveal several consistent findings: Results show that frontier models (e.g., `gpt-5`, `grok-4-fast`, `deepseek-r1-0528`, `gemini-2.5-pro`) achieve 85–95% accuracy in *Discriminative task*, but *Enumerative task* remains extremely difficult, where only `gpt-5` reaches averagely 34% exact accuracy on the subset, while most other models stay below 1%, and the best model only reaches 42% consistency rate for the *Generative task*. In difficulty-based analysis, we found that the accuracy of Task 1 on hard samples drops to around 80% for frontier models and below 40% for smaller ones, while in Task 2 on Easy samples, the best model (`gpt-5`) improves average exact accuracy to 58%. Moreover, whether the natural-language statements are common-sense, counterfactual, or human-like has little impact on the results. We summarize the contributions as follows:

- **LogiConBench Framework.** We propose a novel framework that automatically constructs diverse and scalable logical consistency data through newly designed logical graphs, where the nodes represent propositions and logical relations are on the edges.
- **LogiConBench Dataset.** We produce a large-scale corpus of 280K samples with varying difficulty levels, which covers diverse and important logical reasoning rules.
- **Evaluation and Analysis.** We conduct experiments across varying levels of difficulty on 14 state-of-the-art LLMs, which shows that even strong models fail on more than half of the tasks, confirming that logical consistency reasoning remains highly challenging.

## 2 PRELIMINARIES

Our benchmark is grounded in standard natural deduction rules (Liu & Stokhof, 2024). In particular, we adopt the introduction rules for five logical operators: implication $I_\rightarrow$: $(\varphi \vdash \psi) \Rightarrow (\varphi \rightarrow \psi)$; disjunction $I_\vee$: $\varphi \Rightarrow (\varphi \vee \psi)$ or $\psi \Rightarrow (\varphi \vee \psi)$; conjunction $I_\wedge$: $\varphi, \psi \Rightarrow (\varphi \wedge \psi)$; negation $I_\neg$: $(\varphi \vdash \bot) \Rightarrow (\neg\varphi)$; and biconditional $I_\leftrightarrow$: $(\varphi \rightarrow \psi) \wedge (\psi \rightarrow \varphi) \Rightarrow (\varphi \leftrightarrow \psi)$. These rules serve as the fundamental logical primitives, which allow atomic propositions to be systematically combined and deduced into complex reasoning forms and axiomatic structures (Chiswell & Hodges, 2007; Westerstahl, 2022), hence, they serve as the foundation for our benchmark. A detailed illustration is given in Appendix C.

## 3 CONSTRUCTION OF LOGICONBENCH CORPUS

LogiConBench is constructed through a pipeline that generates datasets with $k = 2, 3, 4, 5$, where $k$ denotes the number of propositions in a single sample, and, as detailed in Appendix D, the data exhibit diverse distributions of atoms and logical operators.

### 3.1 THE LOGICAL GRAPH

**Granularity of the logical graph.** Constructing **multi-step reasoning trees or graphs** is an effective way to capture the reasoning process and to enable scaling across diverse corpora. Instead of relying on pre-defined axioms or costly natural language explanations, our approach deliberately starts from the level of **atomic propositions**, the finest granularity of reasoning. By composing these atomic elements through basic logical rules, we construct **Logical Graphs** that naturally reflect complex reasoning structures.

**Logical Graph Construction from Basic Rules.** Section 2 introduces five fundamental induction logical reasoning rules, which can be composed into infinitely many deductive forms, including axioms. Specifically, **Implication** ($\rightarrow$) and **Biconditional** ($\leftrightarrow$) are expressed only on graph edges, while **Disjunction** ($\vee$) and **Conjunction** ($\wedge$) are expressed only on nodes. **Negation** ($\neg$) can present

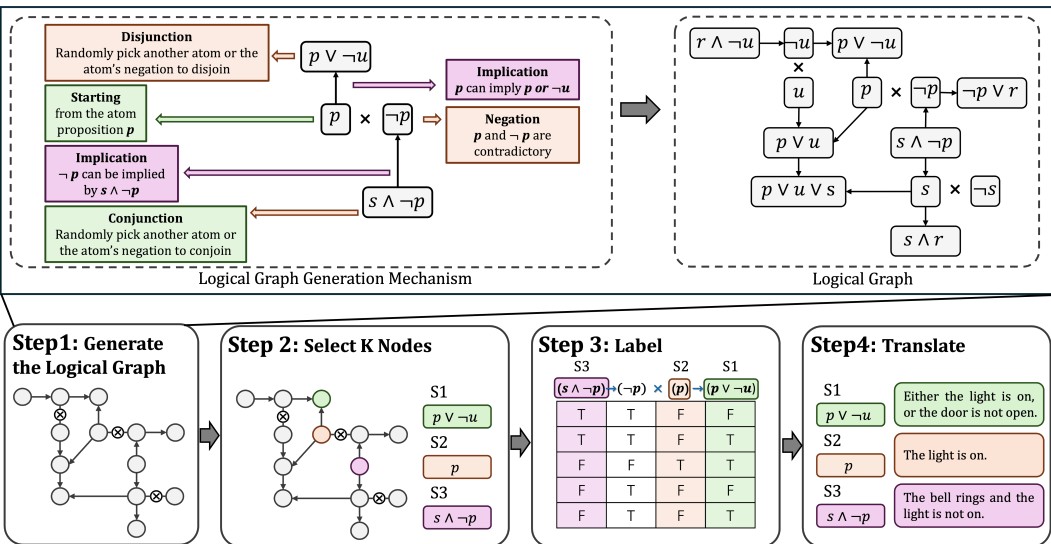

Figure 2: The overall pipeline of LogiconBench, including 4 steps: logical graph generation, node selection, truth labeling, and natural language translation.

both on nodes and edges, where an edge with contradiction is represented as "×". For example, we begin from a fixed atomic proposition $p$, and generate its negation $\neg p$, the conjunctive expansion $p \wedge q$ (where $q$ is a randomly sampled atom from a pre-defined atom list with length 8), and the disjunctive expansion $p \vee q$ that implies $p$. These yield the initial structure:

$$p \vee q \;\rightarrow\; p \;\rightarrow\; p \wedge q, \qquad p \times \neg p.$$

Subsequently, each newly added node can be further expanded in the same manner, which allows the **Logical Graph** to grow indefinitely through iterative application of these basic rules.

## 3.2 Random Sampling and Labeling Along Reasoning Paths

**Random sampling.** To stratify difficulty, we sample, for each $k \in \{2, 3, 4, 5\}$, exactly 10,000 examples where the target set contains $k$ distinct nodes. Since we require every example to admit **at least one inconsistent** label list, we avoid picking nodes that are too distant in the global graph, which are weakly constrained and satisfy all the truth labels. So we ensure they remain within a `bounded graph distance` set to 6, so that every example preserves sufficient local constraints. We also expect the formulas within an example to have diverse complexity, so we randomly sample $k$ distinct nodes $S = \{v_1, \ldots, v_k\}$ with a uniformly distributed number of atoms in their formulas.

**Path extraction.** After sampling from the logical graph, we obtain the target set $S$. We then extract a small subgraph that connects all targets by solving the Steiner tree problem (Hwang & Richards, 1992) with a permutation-based shortest-walk search, which results in an ordered edge list $\mathcal{E} = [(u_1, t_1, v_1), \ldots, (u_m, t_m, v_m)]$ that we use for labeling. The details can be found in Appendix E.

**Constraint semantics.** Edges encode pairwise truth-compatibility of their endpoints via the rule set

$$\text{LOGIC\_RULES} = \left\{ \begin{array}{l} \rightarrow: \{(T,T), (F,T), (F,F)\}, \\ \leftarrow: \{(T,T), (T,F), (F,F)\}, \\ \leftrightarrow: \{(T,T), (F,F)\}, \\ \times: \{(T,F), (F,T)\} \end{array} \right\}.$$

**Labeling via DFS propagation.** We perform a depth-first search (DFS) over $\mathcal{E}$ that incrementally assigns Boolean values to nodes according to `LOGIC_RULES`. The projections of all rule-consistent assignments onto the target nodes are collected as `consistent_lists`. Since each target list of $k$ nodes has $2^k$ possible Boolean assignments in total, the remaining assignments form the `inconsistent_lists`. We ensure that every sample contains both nonempty `consistent_lists` and `inconsistent_lists` for downstream evaluation. An example of labeling is shown in Appendix F.

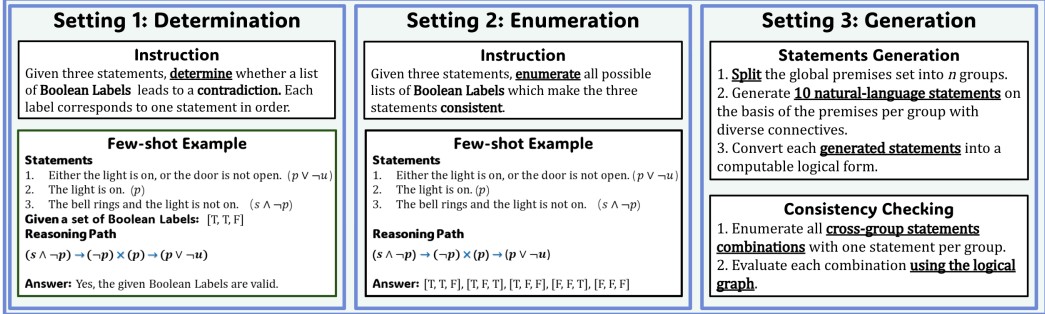

Figure 3: Illustration of the three task tasks in **LogiConBench**.

## 3.3 REWRITING

After sampling $k = 2/3/4/5$ target nodes (10,000 examples for each $k$) in the logical graph generated by the five induction rules, we apply the set of elimination rewrite rules as shown in Appendix G to every node in the sampled subgraphs (Liu & Stokhof, 2024). For each node, if a rewrite rule produces a valid transformed formula, the rewritten form is retained as an additional node. Several rewrite rules (e.g., *equivalence elimination*, conversion to conjunctive normal form (*CNF*) or disjunctive normal form (*DNF*)) are applied at the formula level, since nodes may contain multiple atoms, thereby producing multi-atom rewrites.

## 3.4 SYMBOLIC-TO-NATURAL LANGUAGE TRANSLATION

Following Morishita et al. (2023), we convert symbolic formulas into natural language sentences. For each atomic proposition, we randomly select NOUNs, ADJs, and VERBs drawn from Word-Net (Fellbaum, 2005), where we obtain a large variety of words to prevent models from overfitting to fixed wordings and better approximate real-world language diversity. We apply **negation rules** to cover both affirmative and negative variants of atomic statements. For composite formulas, we use **structural templates** that map logical connectives (e.g., $\wedge$, $\vee$, $\neg$) into corresponding natural language operators. The full set of rules and templates is provided in Appendix H.

## 4 BENCHMARKING LOGICAL CONSISTENCY

**Structure.** As shown in Figure 3, in this chapter, we introduce three primary benchmark tasks: **Task 1 (*Discriminative task*)**, **Task 2 (*Enumerative task*)**, and **Task 3 (*Generative task*)**. To capture more diverse performance patterns, we further refine them in Chapter 5. For Task 1, we evaluate on **hard samples**, where consistent and inconsistent label lists differ by only one element, and we also include a **label completion variant**. For Task 2, we evaluate on samples with **short reasoning paths** and **short statement length**, which enable a more fine-grained assessment of model reasoning. **We also evaluated task 2 on natural-language statements in common-sense, counterfactual, and human-like types, which shows little impact on the model performance.**

**Testing Models.** We evaluate a diverse collection of 14 models on the LogiConBench, covering state-of-the-art proprietary systems as well as large and small size open-source models: `grok-4-fast` (xAI, 2025), `qwen3-235b-a22b` (Yang et al., 2025), `qwen2.5-7b-instruct` (Yang et al., 2024), `gpt-5` (OpenAI, 2025a), `o3-mini` (OpenAI, 2025b), `mixtral-8x7b-instruct` (Jiang et al., 2024), `phi-4-reasoning-plus` (Abdin et al., 2025), `llama-3.1-8b-instruct` (Grattafiori et al., 2024), `llama-3.1-405b-instruct` (Grattafiori et al., 2024), `gpt-4o` (Hurst et al., 2024), `gemini-2.5-pro` (Comanici et al., 2025), `deepseek-r1-0528` (Guo et al., 2025), `claude-sonnet-4` (Anthropic, 2025), and `claude-3.5-haiku` (Anthropic, 2024).

For each experiment, we consider three evaluation settings: zero-shot (without examples in a prompt), few-shot (with three examples in a prompt), and few-shot with reasoning paths (with three examples plus corresponding ground-truth reasoning paths, as shown in Figure 3). Each setting is

Table 2: Performance on **Task 1 (Discriminative Task)** across different numbers of statements (2–5). Results are shown for *consistent samples*, *inconsistent samples*, and their *overall average accuracy* under three evaluation setups: zero-shot, 3-shot, and 3-shot with reasoning path. Background cell colors range from light to dark, indicating increasing values within each column.

| | Model | Accuracy on 2 Statements | | | Accuracy on 3 Statements | | | Accuracy on 4 Statements | | | Accuracy on 5 Statements | | |
|---|---|---|---|---|---|---|---|---|---|---|---|---|---|
| | | Con. | Incon. | Overall | Con. | Incon. | Overall | Con. | Incon. | Overall | Con. | Incon. | Overall |
| zero-shot learning | grok-4-fast | **95.30%** | 93.40% | **94.35%** | 90.70% | **97.30%** | **94.00%** | 80.00% | **97.20%** | 88.60% | 74.90% | **96.90%** | 85.90% |
| | gpt-5 | 88.40% | 91.70% | 90.05% | 87.30% | 92.50% | 89.90% | 87.30% | 92.80% | **90.05%** | 78.40% | 96.60% | **87.50%** |
| | deepseek-r1-0528 | 84.50% | **94.70%** | 89.60% | **91.60%** | 95.70% | 93.65% | 74.40% | 94.90% | 84.65% | 63.70% | 95.30% | 79.50% |
| | claude-sonnet-4 | 61.70% | 86.10% | 73.90% | 53.80% | 85.00% | 69.40% | 39.20% | 91.40% | 65.30% | 34.10% | 93.80% | 63.95% |
| | qwen3-235b-a22b | 49.00% | 87.90% | 68.45% | 60.00% | 84.30% | 72.15% | 84.20% | 54.70% | 69.45% | 54.30% | 58.80% | 56.55% |
| | gemini-2.5-pro | 58.00% | 59.30% | 58.65% | 58.90% | 67.10% | 63.00% | 51.60% | 67.30% | 59.45% | 47.60% | 66.50% | 57.05% |
| | llama-3.1-405b-instruct | 64.20% | 43.90% | 54.05% | 68.20% | 36.40% | 52.30% | 72.60% | 37.90% | 55.25% | 65.30% | 36.80% | 51.05% |
| | qwen2.5-7b-instruct | 17.50% | 91.80% | 54.65% | 41.70% | 66.80% | 54.25% | 49.60% | 55.60% | 52.60% | 43.60% | 42.80% | 43.20% |
| | phi-4-reasoning-plus | 11.10% | 94.60% | 52.85% | 27.90% | 86.50% | 57.20% | 33.60% | 73.00% | 53.30% | 25.30% | 65.40% | 45.35% |
| | mixtral-8x7b-instruct | 64.60% | 39.20% | 51.90% | 89.30% | 25.80% | 57.55% | 81.50% | 22.20% | 51.85% | 79.20% | 14.30% | 46.75% |
| | o3-mini | 58.10% | 41.50% | 49.80% | 56.60% | 35.40% | 46.00% | 61.30% | 27.10% | 44.20% | 64.00% | 16.40% | 40.20% |
| | claude-3.5-haiku | 40.50% | 44.60% | 42.55% | 82.80% | 15.80% | 49.30% | 93.40% | 8.80% | 51.10% | **95.10%** | 2.40% | 48.75% |
| | llama-3.1-8b-instruct | 36.00% | 32.00% | 34.00% | 52.70% | 18.40% | 35.55% | 54.70% | 10.10% | 32.40% | 56.90% | 6.70% | 31.80% |
| | gpt-4o | 36.90% | 29.00% | 32.95% | 44.00% | 19.40% | 31.70% | 58.50% | 11.80% | 35.15% | 47.40% | 14.90% | 31.15% |
| 3-shot learning | grok-4-fast | 93.70% | 91.30% | **92.50%** | 91.30% | **96.30%** | 93.80% | 80.40% | 95.30% | 87.85% | 76.30% | 96.20% | 86.25% |
| | gpt-5 | 87.10% | 92.70% | 89.90% | 86.40% | 87.50% | 86.95% | 86.40% | 96.10% | 91.25% | 73.90% | 95.10% | 84.50% |
| | deepseek-r1-0528 | 89.00% | 94.90% | 91.95% | 94.00% | 95.90% | **94.95%** | 89.90% | 96.70% | **93.30%** | 64.10% | 95.80% | 79.95% |
| | claude-sonnet-4 | 61.80% | 84.40% | 73.10% | 52.40% | 89.80% | 71.10% | 37.10% | 90.80% | 63.95% | 38.70% | 93.70% | 66.20% |
| | qwen3-235b-a22b | 58.70% | 86.00% | 72.35% | **95.90%** | 86.50% | 91.20% | **99.20%** | 64.90% | 82.05% | **98.90%** | 57.70% | 78.30% |
| | gemini-2.5-pro | 80.80% | 89.50% | 85.15% | 87.50% | 86.10% | 86.80% | 85.90% | **97.40%** | 91.15% | 76.20% | **97.30%** | **86.75%** |
| | llama-3.1-405b-instruct | 66.80% | 47.80% | 57.30% | 72.00% | 38.00% | 55.00% | 72.50% | 37.90% | 55.20% | 62.50% | 38.50% | 50.50% |
| | qwen2.5-7b-instruct | 16.80% | 91.70% | 54.25% | 45.00% | 62.80% | 53.90% | 54.10% | 50.90% | 52.50% | 48.70% | 46.60% | 47.65% |
| | phi-4-reasoning-plus | 14.20% | **96.90%** | 55.55% | 19.90% | 84.50% | 52.20% | 34.10% | 70.20% | 52.15% | 32.70% | 63.60% | 48.15% |
| | mixtral-8x7b-instruct | 65.40% | 44.00% | 54.70% | 90.40% | 23.70% | 57.05% | 88.60% | 29.30% | 58.95% | 87.00% | 14.50% | 50.75% |
| | o3-mini | **95.60%** | 56.10% | 75.85% | 95.00% | 46.30% | 70.65% | 94.50% | 19.90% | 57.20% | 94.20% | 16.70% | 55.45% |
| | claude-3.5-haiku | 41.00% | 52.80% | 46.90% | 88.40% | 20.40% | 54.40% | 93.60% | 6.70% | 50.15% | 94.20% | 3.10% | 48.65% |
| | llama-3.1-8b-instruct | 40.10% | 34.20% | 37.15% | 55.90% | 16.10% | 36.00% | 53.90% | 10.10% | 32.00% | 58.00% | 6.70% | 32.35% |
| | gpt-4o | 60.60% | 54.60% | 57.60% | 77.40% | 26.00% | 51.70% | 74.30% | 34.70% | 54.50% | 62.70% | 20.50% | 41.60% |
| 3-shot learning w/ reasoning path | grok-4-fast | **96.10%** | 95.20% | **95.65%** | 93.00% | 98.30% | **95.65%** | 83.50% | 99.50% | 91.50% | 79.30% | **100.00%** | 89.65% |
| | gpt-5 | 92.90% | 93.50% | 93.20% | 88.20% | 96.50% | 92.35% | 88.30% | 97.70% | **93.00%** | 79.60% | 99.80% | **89.70%** |
| | deepseek-r1-0528 | 90.60% | 94.50% | 92.55% | 93.70% | 96.60% | 95.15% | 87.90% | 95.40% | 91.65% | 69.60% | 98.80% | 84.20% |
| | claude-sonnet-4 | 67.80% | 92.30% | 80.05% | 54.10% | 91.00% | 72.55% | 46.10% | 93.00% | 69.55% | 40.20% | 95.00% | 67.60% |
| | qwen3-235b-a22b | 73.80% | **99.70%** | 86.75% | **98.70%** | 89.90% | 94.30% | **100.00%** | 76.30% | 88.15% | **99.40%** | 59.30% | 79.35% |
| | gemini-2.5-pro | 84.00% | 80.90% | 82.45% | 88.10% | **99.40%** | 93.75% | 82.80% | **99.70%** | 91.25% | 50.90% | 98.50% | 74.70% |
| | llama-3.1-405b-instruct | 71.90% | 51.80% | 61.85% | 86.80% | 37.80% | 62.30% | 77.60% | 41.30% | 59.45% | 70.80% | 45.50% | 58.15% |
| | qwen2.5-7b-instruct | 25.20% | 95.90% | 60.55% | 46.70% | 72.20% | 59.45% | 61.60% | 57.40% | 59.50% | 52.30% | 52.40% | 52.35% |
| | phi-4-reasoning-plus | 14.90% | 96.30% | 55.60% | 34.60% | 88.80% | 61.70% | 34.10% | 75.00% | 54.55% | 36.20% | 75.60% | 55.90% |
| | mixtral-8x7b-instruct | 69.80% | 50.20% | 60.00% | 91.20% | 31.90% | 61.55% | 86.40% | 22.00% | 54.20% | 81.00% | 16.50% | 48.75% |
| | o3-mini | 58.70% | 44.70% | 51.70% | 60.80% | 36.20% | 48.50% | 61.40% | 32.30% | 46.85% | 62.10% | 12.90% | 37.50% |
| | claude-3.5-haiku | 49.40% | 62.40% | 55.90% | 89.20% | 21.40% | 55.30% | 93.30% | 9.50% | 51.40% | 96.70% | 3.70% | 50.20% |
| | llama-3.1-8b-instruct | 55.70% | 45.30% | 50.50% | 84.00% | 27.60% | 55.80% | 89.70% | 12.20% | 50.95% | 85.90% | 13.90% | 49.90% |
| | gpt-4o | 34.30% | 59.30% | 46.80% | 46.90% | 30.20% | 38.55% | 54.80% | 35.00% | 44.90% | 45.50% | 32.30% | 38.90% |

applied across datasets of statement size $k = 2, 3, 4, 5$. All evaluations are conducted in a single-round format with the temperature fixed at 0, unless otherwise specified in the model card. For every configuration, we randomly sample 1,000 instances from the dataset for evaluation. The evaluation metrics details can be found in Appendix I.

## 4.1  TASK 1: DISCRIMINATIVE TASK

**Formulation of Task 1.** Task 1 focuses on determining whether a given list of **Boolean labels** assigned to $k$ logical statements leads to a contradiction, for statements size $k = 2, 3, 4, 5$.

**Findings of Task 1.** The experimental results for Task 1 is presented in Table 2. (1) The most advanced LLMs, including gpt-5, grok-4-fast, deepseek-r1-0528, and gemini-2.5-pro, consistently achieve accuracies in the 85–95% range. (2) Models show stable bias patterns across settings and number of statements $k$: some (e.g., claude-3.5-haiku) perform better on consistent than inconsistent statements, others (e.g., phi-4-reasoning-plus) show the opposite trend, while a third group (e.g., gpt-5) consistently favors inconsistent cases, with the gap widening as task size grows. (3) Prompting improves performance: average accuracy rises from 59.21% (zero-shot) to 65.70% (few-shot) and 67.58% (few-path), confirming the value of reasoning-path supervision. (4) Increasing $k$ substantially raises difficulty: accuracy drops from 66.87% at $k = 2$ to only 59.59% at $k = 5$.

Table 3: Performance in **Task 2 (Enumerative Task)** across models and prompting settings. We report *Format* (Executable rate), *Exact* (Exact accuracy of all consistent lists), and *F1* (partial correctness) under three evaluation setups: zero-shot, 3-shot, and 3-shot with reasoning path. The best model for each metric and setup is shown in **bold**, and the second-best is underlined.

| Model (mode) | Accuracy on 2 Statements | | | Accuracy on 3 Statements | | | Accuracy on 4 Statements | | | Accuracy on 5 Statements | | |
|---|---|---|---|---|---|---|---|---|---|---|---|---|
| | Format | Exact | F1 | Format | Exact | F1 | Format | Exact | F1 | Format | Exact | F1 |
| *zero-shot learning* | | | | | | | | | | | | |
| grok-4-fast | 0.280 | 0.072 | 0.459 | 0.433 | 0.167 | 0.301 | 0.517 | 0.034 | 0.230 | 0.517 | 0.000 | 0.194 |
| gpt-5 | 0.982 | **0.383** | **0.751** | 0.972 | **0.439** | **0.775** | 0.962 | **0.250** | **0.736** | 0.971 | 0.073 | **0.664** |
| deepseek-r1-0528 | 0.892 | 0.270 | 0.099 | 0.724 | 0.000 | 0.068 | 0.565 | 0.032 | 0.041 | 0.528 | 0.009 | 0.400 |
| claude-sonnet-4 | 0.880 | 0.043 | 0.494 | 0.843 | 0.000 | 0.317 | 0.813 | 0.000 | 0.193 | 0.754 | 0.000 | 0.123 |
| qwen3-235b-a22b | 0.959 | 0.021 | 0.523 | 0.875 | 0.031 | 0.437 | **1.000** | 0.000 | 0.207 | **1.000** | 0.000 | 0.135 |
| gemini-2.5-pro | 0.977 | 0.055 | 0.527 | 0.950 | 0.120 | 0.409 | 0.914 | 0.000 | 0.099 | 0.840 | **0.080** | 0.040 |
| llama-3.1-405b-instruct | 0.530 | 0.017 | 0.267 | 0.604 | 0.002 | 0.487 | 0.706 | 0.005 | 0.489 | 0.612 | 0.003 | 0.218 |
| qwen-2.5-7b-instruct | **1.000** | 0.009 | 0.455 | **1.000** | 0.006 | 0.452 | **1.000** | 0.000 | 0.367 | **1.000** | 0.000 | 0.240 |
| phi-4-reasoning-plus | 0.947 | 0.017 | 0.196 | 0.937 | 0.000 | 0.093 | 0.918 | 0.000 | 0.055 | 0.937 | 0.000 | 0.069 |
| mixtral-8x7b-instruct | **1.000** | 0.003 | 0.109 | 0.972 | 0.000 | 0.097 | 0.964 | 0.000 | 0.062 | **1.000** | 0.000 | 0.019 |
| o3-mini | **1.000** | 0.000 | 0.082 | **1.000** | 0.000 | 0.014 | 0.968 | 0.000 | 0.067 | **1.000** | 0.000 | 0.010 |
| claude-3.5-haiku | **1.000** | 0.027 | 0.406 | 0.940 | 0.000 | 0.263 | 0.957 | 0.000 | 0.141 | 0.890 | 0.000 | 0.059 |
| llama-3.1-8b-instruct | 0.351 | 0.002 | 0.287 | 0.430 | 0.000 | 0.326 | 0.333 | 0.000 | 0.074 | 0.433 | 0.000 | 0.058 |
| gpt-4o | **1.000** | 0.018 | 0.111 | 0.980 | 0.000 | 0.065 | **1.000** | 0.000 | 0.025 | **1.000** | 0.000 | 0.024 |
| *3-shot learning* | | | | | | | | | | | | |
| grok-4-fast | 0.233 | 0.087 | 0.147 | 0.448 | 0.034 | 0.255 | 0.533 | 0.100 | 0.298 | 0.300 | 0.000 | 0.174 |
| gpt-5 | 0.997 | **0.392** | **0.768** | 0.990 | **0.443** | **0.759** | 0.971 | **0.267** | **0.776** | 0.985 | **0.124** | **0.678** |
| deepseek-r1-0528 | 0.833 | 0.069 | 0.108 | 0.681 | 0.034 | 0.069 | 0.590 | 0.004 | 0.047 | 0.559 | 0.008 | 0.468 |
| claude-sonnet-4 | 0.900 | 0.050 | 0.494 | 0.773 | 0.003 | 0.315 | 0.803 | 0.000 | 0.218 | 0.720 | 0.000 | 0.124 |
| qwen3-235b-a22b | 0.897 | 0.006 | 0.564 | 0.914 | 0.006 | 0.470 | **1.000** | 0.000 | 0.179 | **1.000** | 0.000 | 0.122 |
| gemini-2.5-pro | 0.990 | 0.045 | 0.511 | 0.949 | 0.012 | 0.127 | 0.939 | 0.030 | 0.085 | **1.000** | 0.037 | 0.036 |
| llama-3.1-405b-instruct | 0.652 | 0.023 | 0.257 | 0.680 | 0.005 | 0.255 | 0.667 | 0.006 | 0.181 | 0.640 | 0.000 | 0.436 |
| qwen-2.5-7b-instruct | **1.000** | 0.028 | 0.518 | 0.995 | 0.000 | 0.478 | 0.995 | 0.005 | 0.370 | **1.000** | 0.000 | 0.257 |
| phi-4-reasoning-plus | 0.962 | 0.063 | 0.261 | 0.949 | 0.000 | 0.081 | 0.953 | 0.000 | 0.068 | 0.913 | 0.000 | 0.057 |
| mixtral-8x7b-instruct | **1.000** | 0.000 | 0.233 | 0.983 | 0.000 | 0.079 | **1.000** | 0.000 | 0.018 | **1.000** | 0.000 | 0.074 |
| o3-mini | 0.963 | 0.037 | 0.070 | **1.000** | 0.000 | 0.048 | 0.949 | 0.017 | 0.065 | **1.000** | 0.000 | 0.006 |
| claude-3.5-haiku | 0.943 | 0.042 | 0.489 | 0.953 | 0.003 | 0.301 | 0.984 | 0.000 | 0.144 | 0.907 | 0.000 | 0.063 |
| llama-3.1-8b-instruct | 0.293 | 0.010 | 0.229 | 0.458 | 0.000 | 0.327 | 0.300 | 0.000 | 0.044 | 0.400 | 0.000 | 0.029 |
| gpt-4o | 0.983 | 0.009 | 0.106 | 0.995 | 0.000 | 0.136 | **1.000** | 0.000 | 0.006 | 0.973 | 0.000 | 0.031 |
| *3-shot learning w/ reasoning path* | | | | | | | | | | | | |
| grok-4-fast | 0.133 | 0.204 | 0.290 | 0.533 | 0.167 | 0.381 | 0.533 | 0.100 | 0.249 | 0.552 | 0.000 | 0.293 |
| gpt-5 | 0.980 | **0.413** | **0.815** | 0.976 | **0.505** | **0.812** | 0.953 | **0.299** | **0.834** | 0.981 | **0.155** | **0.709** |
| deepseek-r1-0528 | 0.800 | 0.120 | 0.112 | 0.784 | 0.054 | 0.094 | 0.625 | 0.036 | 0.407 | 0.625 | 0.010 | 0.619 |
| claude-sonnet-4 | 0.900 | 0.053 | 0.507 | 0.800 | 0.010 | 0.333 | 0.760 | 0.000 | 0.229 | 0.790 | 0.000 | 0.225 |
| qwen3-235b-a22b | 0.963 | 0.009 | 0.460 | 0.915 | 0.017 | 0.492 | 0.964 | 0.000 | 0.278 | **1.000** | 0.000 | 0.170 |
| gemini-2.5-pro | 0.980 | 0.124 | 0.711 | 0.962 | 0.085 | 0.235 | 0.972 | 0.100 | 0.090 | **1.000** | 0.018 | 0.045 |
| llama-3.1-405b-instruct | 0.567 | 0.057 | 0.325 | 0.550 | 0.100 | 0.221 | 0.642 | 0.008 | 0.526 | 0.743 | 0.000 | 0.481 |
| qwen-2.5-7b-instruct | **1.000** | 0.032 | 0.539 | 0.991 | 0.012 | 0.479 | **1.000** | 0.009 | 0.394 | **1.000** | 0.004 | 0.263 |
| phi-4-reasoning-plus | 0.953 | 0.030 | 0.221 | 0.963 | 0.000 | 0.096 | 0.957 | 0.000 | 0.068 | 0.927 | 0.000 | 0.063 |
| mixtral-8x7b-instruct | **1.000** | 0.018 | 0.046 | **1.000** | 0.000 | 0.127 | **1.000** | 0.000 | 0.000 | 0.941 | 0.000 | 0.020 |
| o3-mini | **1.000** | 0.054 | 0.087 | 0.818 | 0.000 | 0.015 | **1.000** | 0.000 | 0.085 | 0.925 | 0.019 | 0.047 |
| claude-3.5-haiku | 0.938 | 0.047 | 0.547 | 0.957 | 0.003 | 0.302 | 0.947 | 0.000 | 0.152 | 0.903 | 0.000 | 0.075 |
| llama-3.1-8b-instruct | 0.334 | 0.023 | 0.283 | 0.410 | 0.006 | 0.342 | 0.241 | 0.000 | 0.052 | 0.333 | 0.000 | 0.045 |
| gpt-4o | 0.970 | 0.000 | 0.037 | **1.000** | 0.000 | 0.042 | **1.000** | 0.000 | 0.019 | **1.000** | 0.000 | 0.001 |

## 4.2 TASK 2: ENEMERATIVE TASK

**Formulation of Task 2.** Task 2 focuses on the task of enumeration. Given a set of logical statements, the model is required to enumerate all possible lists of Boolean label assignments that remain logically consistent. Unlike Task 1, which only verifies whether a specific label set leads to a contradiction, Task 2 demands a complete search over the label space. Therefore, LLMs must account for all logical consistency constraints, thereby eliminating the potential for shortcuts in consistency evaluation. As a result, Task 2 poses more challenges to the logical consistency reasoning of LLMs.

**Findings Task 2.** As shown in Table 3, Task 2 is considerably more challenging than Task 1, with exact accuracy for most models below 1%. Only the same top models identified in Task 1 perform competitively: gpt-5 under the 3-shot learning setting with reasoning paths achieves the best results (F1 ≈ 0.83, Exact accuracy ≈ 0.51), followed by grok-4-fast (F1 ≈ 0.29, Exact accuracy ≈ 0.20), gemini-2.5-pro (F1 ≈ 0.71, Exact accuracy ≈ 0.12), and deepseek-r1-0528 (F1 ≈ 0.11, Exact accuracy ≈ 0.12). All other systems remain far below 1% Exact accuracy, which highlights the challenge of exhaustive enumeration. Consistently, the models perform best under the 3-shot learning setting with reasoning paths, followed by 3-shot learning without reasoning paths, while the zero-shot setting sees the worst model performances. This observation confirms that reasoning-path supervision is especially critical for this task.

Table 4: Performance on **Task 3 (Generative Task)** across different group statement numbers (2–5). Results are shown for *Executive Rate* and *Consistency* under the zero-shot setup.

| Model | 2 statements | | 3 statements | | 4 statements | | 5 statements | |
|---|---|---|---|---|---|---|---|---|
| | Exec | Cons | Exec | Cons | Exec | Cons | Exec | Cons |
| grok-4-fast | 1 | 0.836 | 0.95 | 0.710 | 0.74 | 0.458 | 0.67 | 0.427 |
| gpt-5 | 1 | 0.826 | 0.94 | 0.686 | 0.77 | 0.436 | 0.66 | 0.381 |
| deepseek-r1-0528 | 1 | 0.802 | 0.88 | 0.649 | 0.63 | 0.372 | 0.46 | 0.283 |
| claude-sonnet-4 | 0.94 | 0.776 | 0.93 | 0.608 | 0.60 | 0.324 | 0.53 | 0.172 |
| qwen3-235b-a22b | 0.93 | 0.593 | 0.77 | 0.471 | 0.48 | 0.141 | 0.30 | 0.019 |
| gemini-2.5-pro | 1 | 0.817 | 0.93 | 0.663 | 0.76 | 0.416 | 0.65 | 0.348 |
| llama-3.1-405b | 0.92 | 0.696 | 0.91 | 0.639 | 0.67 | 0.369 | 0.54 | 0.247 |
| qwen2.5-7b | 0.88 | 0.575 | 0.82 | 0.452 | 0.53 | 0.101 | 0.38 | 0.120 |
| phi-4-reasoning-plus | 0.93 | 0.687 | 0.90 | 0.625 | 0.67 | 0.349 | 0.53 | 0.219 |
| mixtral-8x7b | 0.84 | 0.587 | 0.77 | 0.466 | 0.55 | 0.128 | 0.33 | 0.106 |
| o3-mini | 1 | 0.790 | 0.94 | 0.713 | 0.87 | 0.478 | 0.67 | 0.358 |
| claude-3.5-haiku | 0.94 | 0.696 | 0.91 | 0.638 | 0.68 | 0.369 | 0.43 | 0.257 |
| llama-3.1-8b | 0.89 | 0.668 | 0.88 | 0.596 | 0.66 | 0.308 | 0.33 | 0.138 |
| gpt-4o | 0.98 | 0.703 | 0.89 | 0.649 | 0.68 | 0.385 | 0.55 | 0.287 |

## 4.3 TASK 3: GENERATIVE TASK

**Formulation of Task 3.** This task evaluates whether, given n mutually consistent premises, LLMs can generate n new statements that remain logically consistent. The process begins by partitioning the global set of atomic propositions $\mathcal{P}$ and their truth assignments $\ell$ into $n$ disjoint groups $\{G_i\}_{i=1}^n$. For each group, a set of ten natural-language facts $\{\text{Prem}_i\}$ is generated under strict logical and linguistic constraints, including the mandatory use of multiple logical connectives and a rich vocabulary. Each generated fact is then translated into a symbolic logical form $f_{i,k}^{\text{sym}}$. The core of the task lies in the exhaustive consistency evaluation of all cross-group combinations $C_\alpha = (f_{1,a_1}^{\text{sym}}, \ldots, f_{n,a_n}^{\text{sym}})$ for $\alpha \in \{1, \ldots, 10\}^n$; a combination is deemed consistent if and only if every fact within it evaluates to true under the global assignment $\ell$. Further implementation details are provided in the Appendix.

**Formulation of Task 3.** As shown in Table 4, Execution rate, which means format correctness across n statements in a group, remains high for top models such as `grok-4-fast`, `gpt-5`, `gemini-2.5-pro`, and `o3-mini`, even when $n = 5$. Smaller models (e.g., `qwen2.5-7b`, `mixtral-8x7b`) show notable degradation as $n$ increases. Furthermore, performance follows a clear trend that higher $n$ leads to worse performance, which reflect the **difficulty still exists even for non-enumerative tasks**.

## 5 DIFFICULTY-BASED ANALYSIS

As introduced in Section 4, we design **three** benchmark tasks and extend them with variants to capture a deeper assessment of performance. For Task 1, we evaluate on **hard samples**, where consistent and inconsistent labels differ by only one position, and also introduce a **label completion variant**. For Task 2, we evaluate on **short reasoning edge samples**, **short statement length samples**, and **commonsense, counterfactual, and human-like natural language statements**. This design provides a finer view of when models succeed and when they fail.

### 5.1 TASK 1 ON HARD SAMPLES AND THE LABEL COMPLETION VARIANT

To better probe model limitations, we formulate two harder task variants using hard samples and label completion, as detailed in the Appendix K. Compared with the aggregate Task 1 results, model performances on hard samples reveal sharper contrasts. On hard samples (dark-colored bars in Figure 5a), all model performances drop remarkably. While on the completion variant (Figure 5b), model performances differ more significantly: strong models (`gpt-5`, `gemini-2.5-pro`) plateau around 80–86%, while smaller models (e.g., `claude-3.5-haiku`) collapse below 40%, near random guessing.

Table 5: Performance on three tasks and downstream benchmark correlations.

| Benchmark | Task 1 | | Task 2 | | Task 3 | |
|---|---|---|---|---|---|---|
| | Pearson | Spearman | Pearson | Spearman | Pearson | Spearman |
| livecodebench | 0.675 | 0.636 | 0.651 | 0.696 | 0.623 | 0.689 |
| infinite | 0.762 | 0.763 | 0.735 | 0.779 | 0.744 | 0.749 |
| aime | 0.786 | 0.643 | 0.653 | 0.678 | 0.690 | 0.652 |
| aa-lcr | 0.624 | 0.598 | 0.614 | 0.695 | 0.603 | 0.665 |
| acebench | 0.632 | 0.640 | 0.526 | 0.688 | 0.714 | 0.702 |

## 5.2 TASK 2 ON SAMPLES WITH SHORT REASONING PATH AND STATEMENT LENGTH

Performance shown in Figure 6 improves modestly but remains low overall. On the Short Path subset, under the 3-shot learning setting with reasoning paths, `gpt-5` averagely increases its Exact accuracy from about 34% to 58%, while `grok-4-fast`'s Exact accuracy averagely rises from 12% to 31%. On the Short Length subset, gains are smaller: for instance, `gpt-5` averagely reaches only 34% Exact accuracy, which shows that shorter statements provide limited benefit. Across both subsets, few-shot and especially few-path prompting remain the most effective strategies, with the best models retaining a clear advantage.

## 5.3 TASK 2 ON COMMONSENSE AND COUNTERFACTUAL SAMPLES

To address potential reasoning shortcuts from generated statements **coinciding with or contradicting** real-world facts, we conducted a controlled experiment. We used gpt-5.1 to generate 100 commonsense atomic propositions and 100 counterfactual ones. For both sets, we **randomly substituted** these atomic propositions into previously generated natural-language statement sets while preserving the sets' original logical labels, since the propositional structure remains unchanged. We then evaluated models on these modified statement sets. Our qualitative analysis reveals that the core reasoning strategy remains fundamentally unchanged between commonsense and counterfactual conditions (Appendix L Table 14 and Table 15). Models consistently translate sentences into symbolic representations, leading to highly similar performance patterns despite surface-level differences.

## 5.4 TASK 2 ON HUMANIZED NATURAL-LANGUAGE STATEMENTS

To address the lack of diversity and naturalness in templated text generation, we created 1000 "human-style" paraphrased statement sets. We sampled from each (k)-statement sets and used gpt-4o to rewrite the sentences into more natural English (see Appendix M for the prompt). As shown in Appendix L Table 16, large models exhibited almost no performance drop, in contrast, small models suffered a clear degradation, which confirms that the natural-language understanding component becomes significantly more challenging once the strong surface regularities of templated sentences are removed. The finding that only semantically capable large LLMs succeed when linguistic cues are removed provides strong evidence that LogiConBench genuinely measures the different reasoning capabilities.

## 6 REAL-WORLD SIGNIFICANCE

To clarify the practical value of LogiConBench, we evaluated the same set of models on several widely-used real-world downstream benchmarks, covering code generation (LiveCodeBench (Jain et al., 2025)), long-context writing (InfiniteBench (Zhang et al., 2024)), mathematical reasoning (aime (Maxwell-Jia, 2025)), long-horizon logical reasoning (AA-LCR (Artificial Analysis Team, 2025)), and agent collaboration (ACE Bench (Chen et al., 2025)), as shown in Appendix N Table 17. We additionally computed Pearson and Spearman correlations between LogiConBench's **three** tasks, and each downstream benchmark (as shown in Table 5). The correlations are consistently moderate to strong across all domains, showing that LogiConBench performance is tightly aligned with capabilities that models actually rely on in real-world use.

**Key observation.** (1) **All three tasks correlate strongly with long-context reasoning and agent-style collaborative tasks**, which indicates that LogiConBench captures a model's ability for agent planning, multi-step tool use, and delegated workflows. (2) **LogiConBench also correlates with math and code benchmarks**, which demonstrates that stable logical consistency is predictive of models' reliability in symbolic and algorithmic domains. Importantly, (3) **Task 2 correlates more strongly with long-horizon and multi-context benchmarks**, which reflects that cross-premise consistency (the core of Task 2) aligns with the demands of real agent systems, which often must reason coherently across multiple partial states or instructions.

# 7 UNDERSTANDING MODEL BEHAVIORS

## 7.1 QUANTITATIVE ERROR ANALYSIS

Across all tasks' quantitative performance, we observe three recurring failure modes. (1) **Enumeration breakdown**: in Task 2, models often omit some consistent label lists, produce duplicates, or output in the wrong format, which reflects a lack of systematic coverage. (2) **Error propagation**: as $k$ increases, small local mistakes compound along reasoning chains, which explains the sharp accuracy drop from $k = 2$ to $k = 5$ in Task 1 and the plateauing of frontier models in the Hard samples. (3) **Bias asymmetry**: many models exhibit skewed sensitivity, either over-predicting consistencies (e.g., `claude-3.5-haiku`) or over-detecting inconsistencies (e.g., `phi-4-reasoning-plus`), while a few frontier systems (e.g., `gpt-5`) lean toward inconsistency more systematically. Together, these trends suggest that contradiction detection (Task 1) is still manageable for frontier models, but exhaustive enumeration (Task 2) exposes deeper weaknesses in structured reasoning and systematic search, with large room for improvement.

## 7.2 QUALITATIVE ERROR ANALYSIS

**(1) Different-sized models have a clear difference.** Large models (e.g., GPT-4, Claude-Sonnet) first translate problems into symbolic form, enabling effective short-step logic. In contrast, smaller models (e.g., Llama-3.1-8B) tend to paraphrase the problem and jump to a conclusion, explaining their low accuracy despite moderate F1 scores.**(2) Shared Error Patterns in Large Models.** Even advanced models exhibit critical flaws. Their reasoning is often short-sighted, leading to three recurring errors: *Incomplete Enumeration* (checking too few cases), *One-Way Simplification* (failing to map symbolic results back to the original problem), and *Lost Goals* (losing sight of the main objective during reasoning).**(3) Task-Specific Failure Modes.** The two tasks expose distinct weaknesses. In Task 1, models struggle with *counterfactual exploration*, failing to systematically consider "what-if" scenarios. In Task 2, *semantic drift* occurs, where models output correct intermediate reasoning symbols as the final answer, confusing the tool with the solution.

# 8 CONCLUSIONS, LIMITATIONS AND FUTURE WORKS

**LogiConBench** advances the study of logical consistency in LLMs by addressing three key limitations of existing benchmarks: lack of scalability, absence of explicit reasoning structures, and insufficient difficulty. Our framework automatically generates unlimited logical rule combinations with precise labels, constructs graphs with controllable depths that provide explicit reasoning paths, and remains challenging for state-of-the-art LLMs. Large-scale experiments with 14 frontier models show that while *Discriminative Task* can reach 85–95% accuracy, *Enumerative Task* remains extremely difficult, with the best average exact accuracy only 34%, and *Generative Task* consistency drops sharply as group size increases, with even the best model falling from 83.6% to 42.7%. Moreover, the introduction of difficulty-based task variants reveals stable relative performance rankings across models, which highlights the benchmark's evaluative value. Despite its contributions, **LogiConBench** has certain limitations. First, the natural language generation process still relies on templates and lexical substitution, which may limit linguistic diversity compared to fully human-authored datasets. Second, our experiments mainly evaluate single-turn consistency reasoning. Future directions include enhancing the diversity of language formulations beyond templates to interactive multi-turn, multi-agent or multi-modal tasks where logical consistency plays a central role (Xu et al., 2025b). We also foresee applying LogiConBench for model training and alignment to strengthen consistency reasoning in frontier LLMs.

ETHICS STATEMENT

This work does not involve human subjects or sensitive personal data. All datasets are automatically generated using symbolic rules and publicly available lexical resources, with no privacy or security risks. The study does not raise foreseeable ethical concerns related to fairness, discrimination, or potential harmful use.

REPRODUCIBILITY STATEMENT

All datasets and preprocessing procedures, and evaluation metrics are described in detail in the main text and appendix. Complete implementation details are provided in an anonymized repository containing the code and reproduction instructions at `https://github.com/Bellafc/LogiConBench.git`.

ACKNOWLEDGMENTS

Zhouchen Lin was supported by the Beijing Natural Science Foundation (L257007), the NSF China (62276004), and the Beijing Major Science and Technology Project (Z251100008425006). Bo Li was supported in part by an NSFC grant 62432008, RGC RIF grant R6021-20, an RGC TRS grant T43-513/23N-2, RGC CRF grants C7004-22G, C1029-22G and C6015-23G, NSFC/RGC grant CRS_HKUST601/24 and RGC GRF grants 16207922, 16207423 and 16203824. Fenrong Liu is supported by the Beijing Natural Science Foundation (L257007) and Tsinghua University's Initiative for Advancing First-Class and World-Leading Disciplines in the Humanities and Social Sciences. Yisen Wang is supported by the Beijing Natural Science Foundation (L257007), the Beijing Major Science and Technology Project (Z251100008425006), the National Natural Science Foundation of China (92370129, 62376010), the Beijing Nova Program (20230484344, 20240484642), and State Key Laboratory of General Artificial Intelligence.

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

## USAGE OF AI

In this work, we made limited use of LLMs as an assistive writing tool. Specifically, we used LLMs to replace synonyms, restructure sentences, and brainstorm alternative ways of expressing ideas within paragraphs. All conceptual contributions, research design, experiments, analyses, and final writing decisions were made by the authors. The authors take full responsibility for the accuracy and originality of the content.

# A RELATED WORK

## A.1 DATASETS FOR LOGICAL CONSISTENCY

Several datasets have been introduced to study consistency, but they mostly fall into three categories: (1) *constraint-graph and knowledge-graph resources*, which encode logical relations over structured triples or constraint graphs; (2) *QA-style datasets*, which induce logical dependencies between answers to paired or sequential questions; and (3) *NLI-style corpora*, which frame consistency in terms of entailment, contradiction, or neutrality. While each line provides valuable insights, they are generally tied to specific domains, limited rule templates, or narrow label spaces. Importantly, many of these resources conflate *factual consistency* (whether statements are true with respect to external world knowledge) with *logical consistency* (whether statements are mutually non-contradictory under formal rules), whereas our work isolates and directly evaluates the latter.

**Constraint-graph and knowledge-graph datasets.** **BeliefBank** (Kassner et al., 2021) builds an explicit constraint graph from ConceptNet (Speer et al., 2017) and WordNet (Fellbaum, 2005), where structural rules define positive implications and mutual exclusivities. By instantiating entities into this graph, 12,525 "silver" truth-labeled facts are automatically propagated, and additional human-labeled calibration facts are introduced to refine consistency. The recently introduced **Logical Fact-checking Datasets** (Ghosh et al., 2025) (FreebaseLFC, NELLLFC, and WikiLFC) are derived from large knowledge graphs (Bordes et al., 2013; Carlson et al., 2010; Hu et al., 2021), where triplets are transformed into (Fact, Context) pairs. These benchmarks explicitly support propositional logic queries with negation, conjunction, and disjunction, thus enabling large-scale fact-checking with logical operators. **EntailmentBank** (Dalvi et al., 2021) provides the first dataset of multistep and full derivation entailment trees, a graph-structured form of explanation that links atomic facts to hypotheses through multipremise entailment steps. However, these graph-based resources are tied to specific domains, such as facts, and to specific rule templates, which make them less general for evaluating logical consistency in diverse open-domain settings.

**QA-style consistency datasets.** **ConVQA** (Ray et al., 2019) consists of visual question–answer pairs that are logically related, which naturally induces logical constraints between answers, making the dataset well suited for evaluating consistency in visual reasoning. Several commonsense and scientific QA datasets follow a similar design by introducing logically related statements or questions. **Com2Sense** (Singh et al., 2021) presents paired statements where only one is logically consistent with commonsense knowledge, requiring models to distinguish true from false assertions. **CREAK** (Onoe et al., 2021) further targets commonsense abduction, consisting of fact-verification questions that require linking claims to implicit commonsense knowledge. Likewise, **OBQA** and **QuaRTz** provide science and quantitative reasoning questions with built-in logical dependencies between answers. Beyond pairwise consistency, datasets such as **WIQA** (Tandon et al., 2019), **QuaRel** (Tafjord et al., 2019), and **HotpotQA** (Yang et al., 2018) emphasize causal reasoning. For example, WIQA asks about the effects of perturbations on processes described in procedural texts, requiring the model to trace causal chains and determine whether changes lead to positive, negative, or neutral outcomes. Nonetheless, these QA datasets typically test specific reasoning phenomena and do not provide systematic coverage of logical consistency rules across diverse contexts.

**Natural Language inference (NLI) datasets.** Standard NLI corpora such as **SNLI** (Bowman et al., 2015) and **MultiNLI** (Wang et al., 2018) can be adapted for logical consistency evaluation, as they contain premise–hypothesis pairs annotated with entailment, contradiction, or neutrality. Recent extensions have moved beyond pairs to sets of statements: **Set-SNLI and Set-LConVQA** (Song et al., 2025) require detecting whether an entire set of sentences is mutually consistent and identifying the specific statements that introduce conflict. **FOLIO** (Han et al., 2024) is the first expert-written dataset for first-order logic (FOL) reasoning, where each example pairs a set of natural language premises with a conclusion derived from NLI symbols, together with a parallel formalization in FOL. Yet these NLI-style resources are limited in scale and coverage of logical operators, which are contradiction, neutral, and entailment, and thus cannot fully capture the broad range of logical consistency phenomena.

### A.2 METHODS FOR LOGICAL CONSISTENCY REASONING

**Fact-checking tasks.** BeliefBank (Kassner et al., 2021) embeds a pretrained language model in a system with an evolving symbolic memory, using a weighted MaxSAT solver to reason over dependencies and a feedback mechanism to query the model with known beliefs as context. Ghosh et al. (2025) introduces logical fact-checking datasets over knowledge graphs, proposes measures of logical consistency on propositional logic queries, and applies supervised fine-tuning to improve performance. Calanzone et al. (2025) introduce a neuro-symbolic loss that enforces consistency with an external set of facts and rules, allowing multiple constraints to be combined in a principled way and improving generalization to unseen but semantically similar knowledge. Paleka et al. (2025) define consistency metrics for LLM forecasters based on arbitrage opportunities, generate logically related question sets, and demonstrate that instantaneous consistency metrics correlate with ground-truth forecasting performance.

**NLI-based tasks.** Li et al. (2019) present a framework that compiles knowledge stated in first-order logic into loss functions, reducing inconsistency in neural models. ConCoRD constructs a factor graph combining model predictions and NLI-based pairwise relations, then applies weighted MaxSAT to select globally consistent answers, boosting performance on QA and VQA benchmarks (Mitchell et al., 2022). Maieutic Prompting recursively generates trees of abductive explanations and frames inference as a satisfiability problem over these explanations and their logical relations, achieving improvements on commonsense reasoning benchmarks (Jung et al., 2022). RE-FLEX adds a rational, self-reflecting layer on top of LLMs: it builds belief graphs through backward chaining and uses a constraint reasoner to minimize contradictions, significantly improving consistency without harming accuracy (Kassner et al., 2023). LogicReward (Jundong Xu, 2026) constructs a reward function integrating step-level logic validity with outcome correctness to train LLMs using reinforcement learning, significantly enhancing the model's logical reasoning capability.

**Comparison QA approaches.** Asai & Hajishirzi (2020) propose logic-guided data augmentation and regularization, leveraging logical and linguistic knowledge to augment training data and constrain predictions, thereby improving global consistency across multiple QA tasks. REPAIR introduces a framework to quantify logical consistency via proxies such as transitivity, commutativity, and negation invariance, evaluates LLMs across multiple comparison tasks, and enhances consistency through data refinement and augmentation (Liu et al., 2025).

## B TABLE 1 EXPLANATION

In this section, we explain how the statistics in Table 1 were obtained.

**BeliefBank.** Section 5.3 of Kassner et al. (2021) mentions that the dataset contains 12,525 "silver" truth-labeled facts. The constraints used are only of two types: Positive Implications and Mutual Exclusivities. Thus, we consider the rule count as 2. Since all constraints are directly instantiated without multi-step reasoning, the depth is set to 1, and the operators involved are implication ($\rightarrow$), negation ($\neg$), and bidirectional implication ($\leftrightarrow$). The dataset does not explicitly provide reasoning paths. As it is developed from ConceptNet, its scalability is considered limited.

**EntailmentBank.** The dataset contains 1,840 QA pairs. Table 1 of Dalvi et al. (2021) reports that the average number of edges per inference is 6, which we use as the average reasoning depth. Only the implication operator ($\rightarrow$) is present, and reasoning paths are explicitly included. However, since all examples were annotated by experts, scalability remains limited.

**LFC.** According to Ghosh et al. (2025), the Logical Fact-checking datasets (FreebaseLFC, NEL-LLFC, WikiLFC) consist of around 2,000 examples. Rules are listed in Tables 13, 19, and 20 of the paper, totaling 9 distinct logical rules. The maximum reasoning depth among them is 4. Since the dataset is constructed from existing sources (Freebase, NELL, and Wiki), scalability is limited, and no reasoning paths are provided.

**Set-LConVQA & Set-SNLI.** Section 4 of Song et al. (2025) states that the dataset contains $6,754 + 6,225 + 200 \times 4 = 13,779$ instances. From Tables 6–9, we identify the largest rea-

Table 6: Natural deduction rules for the five logical operators.

| Operator | Introduction Rule | Elimination Rule |
|---|---|---|
| Implication ($\rightarrow$) | $I_\rightarrow$: If assuming $\varphi$ leads (possibly through several steps) to $\psi$, then infer $\varphi \rightarrow \psi$ (discharge assumption, make it part of the conclusion). | $E_\rightarrow$ (Modus Ponens): From $\varphi$ and $\varphi \rightarrow \psi$, infer $\psi$. |
| Disjunction ($\vee$) | $I_\vee$: From $\varphi$, infer $\varphi \vee \psi$; or from $\psi$, infer $\varphi \vee \psi$ (introduce $\vee$ by either side). | $E_\vee$: From $\varphi \vee \psi$, and the two derivations $\varphi \vdash \chi$ and $\psi \vdash \chi$, infer $\chi$. |
| Conjunction ($\wedge$) | $I_\wedge$: From $\varphi$ and $\psi$, infer $\varphi \wedge \psi$ (introduce $\wedge$ in the conclusion). | $E_\wedge$: From $\varphi \wedge \psi$, infer either $\varphi$ or $\psi$ (eliminate $\wedge$ from the premise). |
| Negation ($\neg$) | $I_\neg$: From assuming $\varphi$ leads to contradiction $\bot$, infer $\neg\varphi$. | $E_\neg$: From $\varphi$ and $\neg\varphi$, infer contradiction $\bot$. |
| Biconditional ($\leftrightarrow$) | $I_\leftrightarrow$: From $\varphi \rightarrow \psi$ and $\psi \rightarrow \varphi$, infer $\varphi \leftrightarrow \psi$. | $E_\leftrightarrow$: From $\varphi \leftrightarrow \psi$, infer either $\varphi \rightarrow \psi$ or $\psi \rightarrow \varphi$. |

soning depth as 5. The total number of rules is $36 + 6 + 3 + 6 = 51$. Operators involved include conjunction ($\wedge$), disjunction ($\vee$), implication ($\rightarrow$), negation ($\neg$), and bidirectional implication ($\leftrightarrow$). Since the dataset is derived from SNLI and LConVQA, scalability is limited, and reasoning paths are not included.

**LogiconBench.** Our benchmark contains 280k logical graphs, with maximum depth 32, involving all five standard operators. Each graph explicitly records reasoning paths, and the dataset construction process can be scaled indefinitely. Therefore, scalability is considered unlimited, and the number of rules grows with the dataset size.

## C  DETAILS OF THE PRELIMINARIES

Table 6 summarizes the introduction and elimination rules in the Natural Deduction for the five logical operators (Liu & Stokhof, 2024) considered in our work: implication, disjunction, conjunction, negation, and biconditional. We used introduction rules to construct the logical graph, and we included the elimination rules for rewriting. These rules specify how new propositions can be derived or simplified in a proof system, and thus provide the basis for generating and evaluating logical graphs. By grounding our benchmark in these standard rules, we ensure that the reasoning tasks are both formally precise and interpretable.

## D  DATASETS STATISTICS

To better characterize our dataset, we report the distribution of logical operators within the constructed nodes in Figure 4. Specifically, we count the occurrences of Not, And, and Or, since these operators are directly involved in forming the node-level expressions. In contrast, Implication ($\rightarrow$) and Equivalence ($\leftrightarrow$) are naturally reflected in the directed edges between nodes, and thus are not included in the node-level statistics. The results in Table 7 show that the distributions vary with $k$, but consistently exhibit a wide coverage across different counts, ensuring diversity in logical complexity.

## E  PATH EXTRACTION

The details of extracting the shortest path is described as follows:

1. For each permutation $\pi$ of $S$, concatenate undirected shortest paths between consecutive pairs $(\pi_i, \pi_{i+1})$ to form a walk covering all targets.

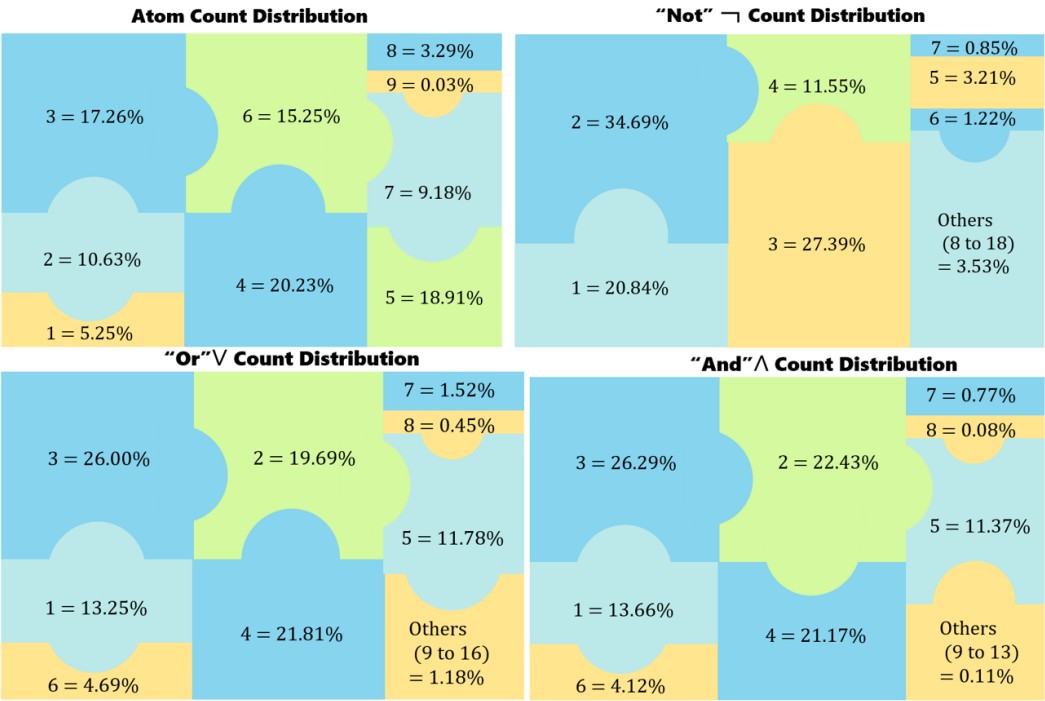

Figure 4: The distribution of logical operators within the constructed nodes.

Table 7: Distribution of operator counts

| Operator | | 1 | 2 | 3 | 4 | 5 | 6 | 7 | 8 | 9 | 10 | 11 | 12 | 13 | 14 | 15 | 16 | 17 | 18 |
|---|---|---|---|---|---|---|---|---|---|---|---|---|---|---|---|---|---|---|---|
| ATOM | k=2 | 2743 | 5302 | 10284 | 12524 | 12007 | 10230 | 6136 | 1366 | 0 | — | — | — | — | — | — | — | — | — |
| | k=3 | 11959 | 22465 | 34740 | 40529 | 38863 | 30806 | 19166 | 7423 | 98 | — | — | — | — | — | — | — | — | — |
| | k=4 | 16490 | 40684 | 60423 | 67757 | 58154 | 47196 | 28414 | 11506 | 52 | — | — | — | — | — | — | — | — | — |
| | k=5 | 28089 | 52362 | 84020 | 99497 | 96025 | 73891 | 43030 | 18548 | 203 | — | — | — | — | — | — | — | — | — |
| NOT | k=2 | 9883 | 18196 | 18284 | 8460 | 2719 | 1166 | 758 | 432 | 390 | 88 | 80 | 16 | 40 | 28 | 22 | 5 | 11 | 14 |
| | k=3 | 43066 | 70093 | 54328 | 24771 | 6539 | 2438 | 1777 | 1128 | 871 | 455 | 195 | 86 | 264 | 23 | 12 | 3 | 0 | 0 |
| | k=4 | 75087 | 112364 | 88805 | 37269 | 8972 | 2910 | 2110 | 1489 | 791 | 373 | 185 | 155 | 79 | 35 | 32 | 20 | 0 | 0 |
| | k=5 | 99746 | 178536 | 137921 | 55743 | 12151 | 4427 | 3137 | 1863 | 1020 | 170 | 383 | 288 | 60 | 21 | 30 | 67 | 43 | 59 |
| OR | k=2 | 7661 | 11386 | 15850 | 13479 | 7657 | 2661 | 948 | 300 | 253 | 122 | 101 | 58 | 58 | 11 | 10 | 37 | — | — |
| | k=3 | 28294 | 37021 | 51723 | 42465 | 26075 | 12697 | 4489 | 1417 | 794 | 374 | 398 | 89 | 213 | 0 | 0 | 0 | — | — |
| | k=4 | 49910 | 71836 | 85635 | 73407 | 32373 | 11421 | 2660 | 832 | 670 | 1177 | 565 | 62 | 128 | 0 | 0 | 0 | — | — |
| | k=5 | 57077 | 100481 | 133061 | 110012 | 59731 | 23579 | 7642 | 1763 | 800 | 522 | 373 | 233 | 254 | 137 | 0 | 0 | — | — |
| AND | k=2 | 7174 | 12532 | 16139 | 13768 | 7400 | 2821 | 636 | 67 | 30 | 9 | 6 | 3 | 7 | — | — | — | — | — |
| | k=3 | 33272 | 47867 | 55672 | 40465 | 20332 | 6955 | 1211 | 133 | 129 | 13 | 0 | 0 | 0 | — | — | — | — | — |
| | k=4 | 40469 | 71935 | 82740 | 71965 | 41676 | 17501 | 3426 | 386 | 73 | 193 | 312 | 0 | 0 | — | — | — | — | — |
| | k=5 | 71365 | 119122 | 131369 | 101983 | 53451 | 15553 | 1928 | 248 | 261 | 42 | 343 | 0 | 0 | — | — | — | — | — |

2. Among all permutations, pick the walk with the fewest distinct edges (ties broken by fewer nodes).

3. Recover edge labels from the original directed graph: if we traverse $u \to v$ along a stored edge, use its type; if we traverse against direction $v \to u$, flip $\to/\leftarrow$ and keep $\times$ or $\leftrightarrow$ unchanged.

This yields an ordered edge list $\mathcal{E} = [(u_1, t_1, v_1), \ldots, (u_m, t_m, v_m)]$ that we use for labeling.

## F LABEL EXAMPLE

Consider three target nodes $S = \{(p \vee \neg u),\ p,\ (s \wedge \neg p)\}$, connected by the path

$$(p \vee \neg u) \ \leftarrow \ p \ \times \ \neg p \ \leftarrow (s \wedge \neg p).$$

By labeling nodes according to LOGIC_RULES, the consistent assignments over all nodes on the path in order are

$$T, T, F, F;\ \ T, F, T, T;\ \ F, T, F, F;\ \ F, F, T, F;\ \ F, F, T, T.$$

Projecting these assignments onto the target list $S$ gives the `consistent_lists`:

$$[T, T, F], \ [T, F, T], \ [F, T, F], \ [F, F, F], \ [F, F, T].$$

Since $S$ has $2^3 = 8$ possible Boolean assignments, the other three, $[T, T, T], [T, F, F], [F, T, T]$ are `inconsistent_lists`.

## G  REWRITE RULES

The rewrite rules are shown in Table 8.

Table 8: Rewrite rules of the symbolic languages.

| ID | Rewrite Rule | Description | Example |
|----|-------------|-------------|---------|
| A | $\varphi \rightsquigarrow simplify(\varphi)$ | Simplification | $(p \wedge \top) \vee \neg\neg q \ \rightsquigarrow \ p \vee q$ |
| B | $\varphi \rightsquigarrow \text{NNF}(\varphi)$ | Negation Normal Form | $\neg(p \rightarrow q) \ \rightsquigarrow \ p \wedge \neg q$ |
| C | $(a \rightarrow b) \rightsquigarrow (\neg a \vee b)$ | Implication elimination | $p \rightarrow q \ \rightsquigarrow \ \neg p \vee q$ |
| D1 | $(a \leftrightarrow b) \rightsquigarrow (a \rightarrow b) \wedge (b \rightarrow a)$ | Equivalence elimination | $p \leftrightarrow q \ \rightsquigarrow \ (p \rightarrow q) \wedge (q \rightarrow p)$ |
| D2 | $(a \leftrightarrow b) \rightsquigarrow (a \wedge b) \vee (\neg a \wedge \neg b)$ | Equivalence elimination | $p \leftrightarrow q \ \rightsquigarrow \ (p \wedge q) \vee (\neg p \wedge \neg q)$ |
| E | $(\neg a \vee b) \rightsquigarrow (a \rightarrow b)$ | Implication introduction | $\neg p \vee q \ \rightsquigarrow \ p \rightarrow q$ |
| F | $(a \wedge b) \vee (\neg a \wedge \neg b) \rightsquigarrow (a \leftrightarrow b)$ | Equivalence introduction | $(p \wedge q) \vee (\neg p \wedge \neg q) \ \rightsquigarrow \ p \leftrightarrow q$ |
| G1 | $\varphi \rightsquigarrow \text{CNF}(\varphi)$ | Conjunctive Normal Form | $\neg(p \wedge q) \vee r \ \rightsquigarrow \ (\neg p \vee r) \wedge (\neg q \vee r)$ |
| G2 | $\varphi \rightsquigarrow \text{DNF}(\varphi)$ | Disjunctive Normal Form | $(p \vee q) \wedge r \ \rightsquigarrow \ (p \wedge r) \vee (q \wedge r)$ |

## H  SYMBOL LANGUAGE TO NATURAL LANGUAGE STRUCTURES

This appendix summarizes the templates used in data construction. First, we present the **negation rules**, which ensure coverage of both affirmative and negative variants of atomic statements.

- *The NOUN is ADJ"* $\mapsto$ *The NOUN is not ADJ"*,
- *The NOUN occurs"* $\mapsto$ *The NOUN does not occur"*,
- *The NOUN VERBs"* $\mapsto$ *The NOUN does not VERB"*,
- *The NOUN has …"* $\mapsto$ *The NOUN does not have …"*.

Second, we define **composite formula templates**, where structural operators correspond to logical connectives (e.g., conjunction, disjunction, negation), enabling natural language rendering of complex formulas.

- $A \wedge B$; ; $\mapsto$; ; *First, A. Second, B."*
- $A \vee B \vee C \ \mapsto \ $ *Either (i) A, or (ii) B, or (iii) C."*
- $\neg A$; ; $\mapsto$; ; *"It is not the case that the following holds: A."*

## I  EVALUATION METRICS

**Task 1 and Task 1 on Hard Samples.**  The evaluation metric is **accuracy**, defined as the percentage of correctly predicted labels. We report: (1) accuracy on consistent lists (i.e., given a consistent set, the model outputs "yes"), (2) accuracy on inconsistent lists (i.e., given an inconsistent set, the model outputs "no"), and (3) overall accuracy. Since our experiments include 500 consistent and 500 inconsistent lists, the overall accuracy is computed as the average of the two.

**Task 1 Variant: Label Completion.**  The evaluation metric is **overall accuracy**, which measures the correctness of label completion, i.e., whether the predicted Boolean assignment correctly completes all statements without contradiction.

**Task 2 and Task 2 on Easy Samples.** The evaluation metrics include **Format accuracy**, **Exact accuracy**, **Precision**, **Recall**, and **F1**.

- **Format accuracy**: the percentage of outputs that follow the required enumeration format.

- **Exact accuracy**: a strict metric that measures the proportion of samples where the model enumerates *exactly* all the consistent label lists in the dataset.

- **TP, TN, FP, FN**: we treat the unlisted assignments as the model's predicted inconsistent lists. Given the ground truth partition of assignments into consistent vs. inconsistent:

  - **TP (True Positive)** = percentage of assignments that are consistent in the ground truth and also predicted as consistent.
  - **FN (False Negative)** = percentage of assignments that are consistent in the ground truth but predicted as inconsistent.
  - **TN (True Negative)** = percentage of assignments that are inconsistent in the ground truth and also predicted as inconsistent.
  - **FP (False Positive)** = percentage of assignments that are inconsistent in the ground truth but predicted as consistent.

- **Precision**: intuitively, how many of the lists predicted as consistent are truly consistent.

- **Recall**: how many of the truly consistent lists are successfully identified by the model.

- **F1 score**: the harmonic mean of precision and recall, reflecting the balance between the two.

## J    EXPERIMENTAL RESULTS

### J.1    TASK 1 ON HARD SAMPLES

The performance of Task 1 on Hard Samples are detailed in Table 9.

### J.2    TASK 1 VARIANT: LABEL COMPLETION

The performance of Task 1 Variant is shown in Table 10

### J.3    TASK 2 PRECISION AND RECALL

The Precision and Recall Score of Task 2 is shown in Table 11.

### J.4    TASK 2 ON EASY SAMPLES (SHORT REASONING PATH SAMPLES AND SHORT STATEMENT LENGTH SAMPLES)

The Performance of Task 2 on Easy Samples is shown in Table 12 and Table 13.

## K    DIFFICULTY ONE AND TWO

**Formulation of Task 1 on hard samples.** In the original task, closed-source models such as `grok-4-fast` and `gpt-5` perform well, which suggests that the task may not fully expose their limitations. To probe this, we design a hard-sample variant where provided lists of Boolean labels come from consistent and inconsistent lists differing in only one element, which makes them more challenging to distinguish. For example, choosing [T, T, T] in `consistent_lists` or [T, T, F] in `inconsistent_lists` as the label list to discriminate.

**Formulation of Task 1 variant: Label Completion.** Label Completion is a harder variant of the Discriminative task. Instead of verifying a full label set, the model must recover the hidden label so that the full label list remains logically consistent, which increases reasoning difficulty.

Table 9: Performance decomposition across zero-shot, 3-shot, and 3-shot with path. Each block reports consistent (con.), inconsistent (incon.), and overall accuracy for $k = 2, 3, 4, 5$.

| Model (mode) | Accuracy on 2 Statements | | | Accuracy on 3 Statements | | | Accuracy on 4 Statements | | | Accuracy on 5 Statements | | |
|---|---|---|---|---|---|---|---|---|---|---|---|---|
| | con. | incon. | overall | con. | incon. | overall | con. | incon. | overall | con. | incon. | overall |
| **zero-shot learning** | | | | | | | | | | | | |
| claude-3.5-haiku (zero-shot) | 31.57 | 38.58 | 35.07 | 31.61 | 38.64 | 35.12 | 31.66 | 38.69 | 35.17 | 31.70 | 38.75 | 35.22 |
| claude-sonnet-4 (zero-shot) | 51.63 | 63.10 | 57.37 | 51.68 | 63.16 | 57.42 | 51.72 | 63.21 | 57.47 | 51.77 | 63.27 | 57.52 |
| deepseek-r1-0528 (zero-shot) | 67.86 | 82.93 | 75.40 | 67.90 | 82.99 | 75.45 | 67.95 | 83.04 | 75.50 | 67.99 | 83.10 | 75.55 |
| gemini-2.5-pro (zero-shot) | 35.68 | 43.61 | 39.64 | 35.73 | 43.66 | 39.69 | 35.77 | 43.72 | 39.74 | 35.82 | 43.77 | 39.79 |
| gpt-4o (zero-shot) | 17.39 | 21.25 | 19.32 | 17.43 | 21.30 | 19.37 | 17.48 | 21.36 | 19.42 | 17.52 | 21.41 | 19.47 |
| llama-3.1-405b-instruct (zero-shot) | 29.98 | 36.65 | 33.32 | 30.03 | 36.70 | 33.37 | 30.07 | 36.76 | 33.42 | 30.12 | 36.81 | 33.47 |
| llama-3.1-8b-instruct (zero-shot) | 21.78 | 26.62 | 24.20 | 21.82 | 26.67 | 24.25 | 21.87 | 26.73 | 24.30 | 21.91 | 26.78 | 24.35 |
| phi-4-reasoning-plus (zero-shot) | 24.80 | 30.31 | 27.56 | 24.85 | 30.37 | 27.61 | 24.89 | 30.42 | 27.66 | 24.94 | 30.48 | 27.71 |
| mixtral-8x7b-instruct (zero-shot) | 18.42 | 22.51 | 20.46 | 18.46 | 22.56 | 20.51 | 18.51 | 22.62 | 20.56 | 18.55 | 22.67 | 20.61 |
| o3-mini (zero-shot) | 19.53 | 23.87 | 21.70 | 19.58 | 23.93 | 21.75 | 19.62 | 23.98 | 21.80 | 19.67 | 24.04 | 21.85 |
| gpt-5 (zero-shot) | 79.57 | 97.25 | 88.41 | 79.61 | 97.30 | 88.46 | 79.66 | 97.36 | 88.51 | 79.70 | 97.41 | 88.56 |
| qwen-2.5-7b-instruct (zero-shot) | 27.85 | 34.04 | 30.95 | 27.90 | 34.10 | 31.00 | 27.94 | 34.15 | 31.05 | 27.99 | 34.21 | 31.10 |
| qwen-3-235b-a22b (zero-shot) | 43.91 | 53.67 | 48.79 | 43.95 | 53.72 | 48.84 | 44.00 | 53.78 | 48.89 | 44.04 | 53.83 | 48.94 |
| grok-4-fast (zero-shot) | 69.43 | 84.86 | 77.15 | 69.48 | 84.92 | 77.20 | 69.52 | 84.97 | 77.25 | 69.57 | 85.03 | 77.30 |
| **3-shot learning** | | | | | | | | | | | | |
| claude-3.5-haiku (3-shot) | 31.14 | 38.06 | 34.60 | 31.19 | 38.12 | 34.65 | 31.23 | 38.17 | 34.70 | 31.28 | 38.23 | 34.75 |
| claude-sonnet-4 (3-shot) | 52.85 | 64.60 | 58.73 | 52.90 | 64.65 | 58.78 | 52.94 | 64.71 | 58.83 | 52.99 | 64.76 | 58.88 |
| deepseek-r1-0528 (3-shot) | 77.23 | 94.39 | 85.81 | 77.27 | 94.44 | 85.86 | 77.32 | 94.50 | 85.91 | 77.36 | 94.55 | 85.96 |
| gemini-2.5-pro (3-shot) | 82.72 | 101.10 | 91.91 | 82.76 | 101.15 | 91.96 | 82.81 | 101.21 | 92.01 | 82.85 | 101.26 | 92.06 |
| gpt-4o (3-shot) | 28.40 | 34.71 | 31.56 | 28.45 | 34.77 | 31.61 | 28.49 | 34.82 | 31.66 | 28.54 | 34.88 | 31.71 |
| llama-3.1-405b-instruct (3-shot) | 34.55 | 42.22 | 38.38 | 34.59 | 42.28 | 38.43 | 34.64 | 42.33 | 38.48 | 34.68 | 42.39 | 38.53 |
| llama-3.1-8b-instruct (3-shot) | 19.72 | 24.10 | 21.91 | 19.76 | 24.16 | 21.96 | 19.81 | 24.21 | 22.01 | 19.85 | 24.27 | 22.06 |
| phi-4-reasoning-plus (3-shot) | 28.87 | 35.29 | 32.08 | 28.92 | 35.35 | 32.13 | 28.96 | 35.40 | 32.18 | 29.01 | 35.46 | 32.23 |
| mixtral-8x7b-instruct (3-shot) | 46.96 | 57.40 | 52.18 | 47.01 | 57.45 | 52.23 | 47.05 | 57.51 | 52.28 | 47.10 | 57.56 | 52.33 |
| o3-mini (3-shot) | 47.87 | 58.51 | 53.19 | 47.91 | 58.56 | 53.24 | 47.96 | 58.62 | 53.29 | 48.00 | 58.67 | 53.34 |
| gpt-5 (3-shot) | 82.87 | 101.29 | 92.08 | 82.92 | 101.34 | 92.13 | 82.96 | 101.40 | 92.18 | 83.01 | 101.45 | 92.23 |
| qwen-2.5-7b-instruct (3-shot) | 41.12 | 50.26 | 45.69 | 41.17 | 50.31 | 45.74 | 41.21 | 50.37 | 45.79 | 41.26 | 50.42 | 45.84 |
| qwen-3-235b-a22b (3-shot) | 68.57 | 83.81 | 76.19 | 68.62 | 83.87 | 76.24 | 68.66 | 83.92 | 76.29 | 68.71 | 83.98 | 76.34 |
| grok-4-fast (3-shot) | 82.49 | 100.82 | 91.65 | 82.53 | 100.87 | 91.70 | 82.58 | 100.93 | 91.75 | 82.62 | 100.98 | 91.80 |
| **3-shot learning w/ reasoning path** | | | | | | | | | | | | |
| claude-3.5-haiku (3-shot with path) | 46.76 | 57.15 | 51.95 | 46.80 | 57.20 | 52.00 | 46.85 | 57.26 | 52.05 | 46.89 | 57.31 | 52.10 |
| claude-sonnet-4 (3-shot with path) | 63.79 | 77.96 | 70.87 | 63.83 | 78.02 | 70.92 | 63.88 | 78.07 | 70.97 | 63.92 | 78.13 | 71.02 |
| deepseek-r1-0528 (3-shot with path) | 89.06 | 108.86 | 98.96 | 89.11 | 108.91 | 99.01 | 89.15 | 108.97 | 99.06 | 89.20 | 109.02 | 99.11 |
| gemini-2.5-pro (3-shot with path) | 82.66 | 101.02 | 91.84 | 82.70 | 101.08 | 91.89 | 82.75 | 101.13 | 91.94 | 82.79 | 101.19 | 91.99 |
| gpt-4o (3-shot with path) | 35.13 | 42.93 | 39.03 | 35.17 | 42.99 | 39.08 | 35.22 | 43.04 | 39.13 | 35.26 | 43.10 | 39.18 |
| llama-3.1-405b-instruct (3-shot with path) | 53.58 | 65.49 | 59.53 | 53.63 | 65.54 | 59.58 | 53.67 | 65.60 | 59.63 | 53.72 | 65.65 | 59.68 |
| llama-3.1-8b-instruct (3-shot with path) | 42.62 | 52.09 | 47.36 | 42.67 | 52.15 | 47.41 | 42.71 | 52.20 | 47.46 | 42.76 | 52.26 | 47.51 |
| phi-4-reasoning-plus (3-shot with path) | 46.89 | 57.32 | 52.11 | 46.94 | 57.37 | 52.16 | 46.98 | 57.43 | 52.21 | 47.03 | 57.48 | 52.26 |
| mixtral-8x7b-instruct (3-shot with path) | 43.24 | 52.85 | 48.05 | 43.29 | 52.91 | 48.10 | 43.33 | 52.96 | 48.15 | 43.38 | 53.02 | 48.20 |
| o3-mini (3-shot with path) | 49.12 | 60.04 | 54.58 | 49.17 | 60.09 | 54.63 | 49.21 | 60.15 | 54.68 | 49.26 | 60.20 | 54.73 |
| gpt-5 (3-shot with path) | 88.81 | 108.55 | 98.68 | 88.86 | 108.61 | 98.73 | 88.90 | 108.66 | 98.78 | 88.95 | 108.72 | 98.83 |
| qwen-2.5-7b-instruct (3-shot with path) | 47.49 | 58.04 | 52.76 | 47.53 | 58.09 | 52.81 | 47.58 | 58.15 | 52.86 | 47.62 | 58.20 | 52.91 |
| qwen-3-235b-a22b (3-shot with path) | 83.61 | 102.20 | 92.90 | 83.66 | 102.25 | 92.95 | 83.70 | 102.31 | 93.00 | 83.75 | 102.36 | 93.05 |
| grok-4-fast (3-shot with path) | 88.41 | 108.06 | 98.24 | 88.46 | 108.12 | 98.29 | 88.50 | 108.17 | 98.34 | 88.55 | 108.23 | 98.39 |

Table 10: Performance on Task 1 variant: label completion of accuracy (%).

| Model | 2 statements | | | 3 statements | | | 4 statements | | | 5 statements | | |
|---|---|---|---|---|---|---|---|---|---|---|---|---|
| | zero | few | few_path | zero | few | few_path | zero | few | few_path | zero | few | few_path |
| claude-3.5-haiku | 0.112 | 0.341 | 0.765 | 0.116 | 0.171 | 0.196 | 0.259 | 0.303 | 0.323 | 0.378 | 0.419 | 0.470 |
| claude-sonnet-4 | 0.312 | 0.587 | 0.672 | 0.518 | 0.617 | 0.654 | 0.413 | 0.534 | 0.713 | 0.319 | 0.576 | 0.654 |
| deepseek-r1-0528 | 0.582 | 0.595 | 0.608 | 0.646 | 0.667 | 0.747 | 0.570 | 0.598 | 0.646 | 0.523 | 0.613 | 0.622 |
| google-gemini-2.5-pro | 0.890 | 0.912 | 0.966 | 0.828 | 0.867 | 0.878 | 0.758 | 0.759 | 0.814 | 0.765 | 0.790 | 0.792 |
| gpt-4o | 0.276 | 0.291 | 0.363 | 0.260 | 0.274 | 0.294 | 0.243 | 0.307 | 0.341 | 0.302 | 0.357 | 0.366 |
| llama-3.1-405b-instruct | 0.280 | 0.335 | 0.490 | 0.227 | 0.263 | 0.327 | 0.205 | 0.262 | 0.275 | 0.158 | 0.162 | 0.189 |
| llama-3.1-8b-instruct | 0.321 | 0.546 | 0.645 | 0.145 | 0.194 | 0.264 | 0.138 | 0.216 | 0.290 | 0.150 | 0.177 | 0.234 |
| phi-4-reasoning-plus | 0.423 | 0.532 | 0.654 | 0.432 | 0.546 | 0.672 | 0.598 | 0.657 | 0.677 | 0.243 | 0.542 | 0.564 |
| mixtral-8x7b-instruct | 0.122 | 0.143 | 0.714 | 0.228 | 0.294 | 0.297 | 0.102 | 0.178 | 0.224 | 0.143 | 0.174 | 0.215 |
| o3-mini | 0.527 | 0.582 | 0.615 | 0.702 | 0.740 | 0.780 | 0.610 | 0.670 | 0.688 | 0.532 | 0.632 | 0.668 |
| gpt-5 | 0.876 | 0.924 | 0.934 | 0.835 | 0.883 | 0.891 | 0.789 | 0.791 | 0.809 | 0.778 | 0.790 | 0.808 |
| qwen2.5-7b-instruct | 0.469 | 0.497 | 0.501 | 0.552 | 0.553 | 0.638 | 0.489 | 0.601 | 0.619 | 0.473 | 0.513 | 0.526 |
| qwen3-235b-a22b | 0.461 | 0.512 | 0.522 | 0.460 | 0.503 | 0.538 | 0.392 | 0.395 | 0.483 | 0.168 | 0.178 | 0.200 |
| grok-4-fast | 0.613 | 0.637 | 0.640 | 0.665 | 0.761 | 0.823 | 0.720 | 0.728 | 0.813 | 0.686 | 0.708 | 0.776 |

**Formulation of Task 2: Easy Samples.** In the original Task 2, smaller open-source models such as `llama-3.1-8b-instruct` and `phi-4-reasoning-plus` struggle with complex dependency structures. To provide a controlled easier variant, we construct test sets by selecting the 1,000 samples with the shortest reasoning paths and the 1,000 samples with the shortest natural language statements to reduce logical and linguistic complexity, respectively.

Table 11: Precision and Recall in Task 2.

| Model | 2 Statements | | 3 Statements | | 4 Statements | | 5 Statements | |
|---|---|---|---|---|---|---|---|---|
| | Precision | Recall | Precision | Recall | Precision | Recall | Precision | Recall |
| claude-3.5-haiku_zero | 0.648 | 0.396 | 0.436 | 0.196 | 0.414 | 0.095 | 0.278 | 0.034 |
| claude-3.5-haiku_few | 0.451 | 0.444 | 0.505 | 0.219 | 0.398 | 0.087 | 0.316 | 0.044 |
| claude-3.5-haiku_fewpath | 0.588 | 0.528 | 0.522 | 0.227 | 0.385 | 0.091 | 0.302 | 0.036 |
| claude-sonnet-4_zero | 0.591 | 0.476 | 0.523 | 0.263 | 0.547 | 0.168 | 0.531 | 0.076 |
| claude-sonnet-4_few | 0.587 | 0.479 | 0.504 | 0.264 | 0.531 | 0.161 | 0.499 | 0.076 |
| claude-sonnet-4_fewpath | 0.615 | 0.486 | 0.534 | 0.283 | 0.487 | 0.136 | 0.482 | 0.066 |
| deepseek-r1-0528_zero | 0.109 | 0.110 | 0.081 | 0.069 | 0.064 | 0.035 | 0.429 | 0.375 |
| deepseek-r1-0528_few | 0.111 | 0.133 | 0.084 | 0.072 | 0.504 | 0.341 | 0.483 | 0.455 |
| deepseek-r1-0528_fewpath | 0.110 | 0.123 | 0.109 | 0.100 | 0.074 | 0.043 | 0.716 | 0.546 |
| gemini-2.5-pro_zero | 0.609 | 0.495 | 0.556 | 0.388 | 0.181 | 0.077 | 0.077 | 0.033 |
| gemini-2.5-pro_few | 0.584 | 0.489 | 0.172 | 0.116 | 0.158 | 0.071 | 0.084 | 0.027 |
| gemini-2.5-pro_fewpath | 0.595 | 0.481 | 0.318 | 0.220 | 0.178 | 0.072 | 0.012 | 0.004 |
| gpt-4o_zero | 0.133 | 0.104 | 0.109 | 0.049 | 0.054 | 0.016 | 0.091 | 0.014 |
| gpt-4o_few | 0.127 | 0.101 | 0.221 | 0.103 | 0.016 | 0.004 | 0.117 | 0.018 |
| gpt-4o_fewpath | 0.044 | 0.034 | 0.071 | 0.031 | 0.041 | 0.013 | 0.003 | 0.000 |
| gpt-5_zero | 0.852 | 0.702 | 0.875 | 0.739 | 0.886 | 0.667 | 0.913 | 0.566 |
| gpt-5_few | 0.859 | 0.730 | 0.856 | 0.720 | 0.912 | 0.716 | 0.894 | 0.590 |
| gpt-5_fewpath | 0.782 | 0.688 | 0.897 | 0.777 | 0.894 | 0.669 | 0.926 | 0.626 |
| grok-4-fast_zero | 0.000 | 0.000 | 0.309 | 0.299 | 0.364 | 0.187 | 0.340 | 0.157 |
| grok-4-fast_few | 0.178 | 0.133 | 0.355 | 0.214 | 0.434 | 0.259 | 0.255 | 0.143 |
| grok-4-fast_fewpath | 0.108 | 0.078 | 0.417 | 0.382 | 0.369 | 0.213 | 0.421 | 0.241 |
| llama-3.1-405b_zero | 0.517 | 0.386 | 0.576 | 0.421 | 0.520 | 0.461 | 0.427 | 0.444 |
| llama-3.1-405b_few | 0.592 | 0.403 | 0.623 | 0.453 | 0.523 | 0.453 | 0.447 | 0.466 |
| llama-3.1-405b_fewpath | 0.611 | 0.430 | 0.633 | 0.445 | 0.574 | 0.485 | 0.467 | 0.497 |
| llama-3.1-8b-instruct_zero | 0.287 | 0.002 | 0.326 | 0.000 | 0.117 | 0.056 | 0.186 | 0.036 |
| llama-3.1-8b-instruct_few | 0.229 | 0.010 | 0.316 | 0.000 | 0.083 | 0.032 | 0.101 | 0.017 |
| llama-3.1-8b-instruct_fewpath | 0.283 | 0.000 | 0.342 | 0.006 | 0.107 | 0.038 | 0.097 | 0.029 |
| mixtral-8x7b_zero | 0.110 | 0.121 | 0.139 | 0.076 | 0.128 | 0.046 | 0.083 | 0.010 |
| mixtral-8x7b_few | 0.180 | 0.333 | 0.131 | 0.060 | 0.033 | 0.013 | 0.102 | 0.058 |
| mixtral-8x7b_fewpath | 0.037 | 0.065 | 0.178 | 0.104 | 0.000 | 0.000 | 0.074 | 0.012 |
| o3-mini_zero | 0.099 | 0.076 | 0.023 | 0.011 | 0.176 | 0.046 | 0.040 | 0.006 |
| o3-mini_few | 0.085 | 0.063 | 0.083 | 0.035 | 0.165 | 0.044 | 0.024 | 0.003 |
| o3-mini_fewpath | 0.100 | 0.081 | 0.021 | 0.012 | 0.013 | 0.003 | 0.152 | 0.031 |
| phi-4-reasoning-plus_zero | 0.229 | 0.206 | 0.140 | 0.082 | 0.167 | 0.037 | 0.249 | 0.044 |
| phi-4-reasoning-plus_few | 0.296 | 0.266 | 0.134 | 0.074 | 0.135 | 0.053 | 0.171 | 0.038 |
| phi-4-reasoning-plus_fewpath | 0.251 | 0.246 | 0.151 | 0.085 | 0.142 | 0.052 | 0.214 | 0.039 |
| qwen-2.5-7b-instruct_zero | 0.544 | 0.391 | 0.504 | 0.410 | 0.364 | 0.371 | 0.204 | 0.292 |
| qwen-2.5-7b-instruct_few | 0.637 | 0.436 | 0.521 | 0.441 | 0.364 | 0.376 | 0.221 | 0.307 |
| qwen-2.5-7b-instruct_fewpath | 0.670 | 0.451 | 0.543 | 0.429 | 0.398 | 0.389 | 0.227 | 0.314 |
| qwen3-235b-a22b_zero | 0.667 | 0.430 | 0.529 | 0.372 | 0.480 | 0.163 | 0.570 | 0.078 |
| qwen3-235b-a22b_few | 0.758 | 0.450 | 0.599 | 0.386 | 0.459 | 0.113 | 0.601 | 0.069 |
| qwen3-235b-a22b_fewpath | 0.592 | 0.376 | 0.632 | 0.402 | 0.500 | 0.121 | 0.533 | 0.066 |

## L COMMONSENSE CONDITION RESULTS

## M HUMANIZED PROMPTS

We use the following prompt to humanize the statements.

```
Please rewrite the following sentences into natural human-style English.
Requirements:
- Do NOT keep any numeric labels (no "1)", "2)", "First,...", etc.).
- Do NOT place "not" or any negation operator at the beginning of a sentence.
  Negation must appear inside the clause in a natural way.
```

Table 12: Performance in Task 2 with short Reasoning Paths.

| Model (mode) | Accuracy on 2 Statements | | | Accuracy on 3 Statements | | | Accuracy on 4 Statements | | | Accuracy on 5 Statements | | |
|---|---|---|---|---|---|---|---|---|---|---|---|---|
| | Precision | Recall | F1 | Precision | Recall | F1 | Precision | Recall | F1 | Precision | Recall | F1 |
| claude-3.5-haiku_zero | 0.609 | 0.471 | 0.531 | 0.345 | 0.433 | 0.384 | 0.246 | 0.420 | 0.310 | 0.198 | 0.420 | 0.269 |
| claude-3.5-haiku_few | 0.643 | 0.489 | 0.556 | 0.370 | 0.449 | 0.406 | 0.249 | 0.419 | 0.312 | 0.255 | 0.474 | 0.332 |
| claude-3.5-haiku_few path | 0.753 | 0.541 | 0.630 | 0.378 | 0.460 | 0.415 | 0.294 | 0.470 | 0.362 | 0.280 | 0.500 | 0.359 |
| claude-sonnet-4_zero | 0.725 | 0.544 | 0.622 | 0.507 | 0.533 | 0.519 | 0.459 | 0.583 | 0.514 | 0.708 | 0.741 | 0.724 |
| claude-sonnet-4_few | 0.730 | 0.543 | 0.623 | 0.537 | 0.548 | 0.543 | 0.478 | 0.594 | 0.530 | 0.757 | 0.752 | 0.754 |
| claude-sonnet-4_few path | 0.745 | 0.550 | 0.633 | 0.537 | 0.557 | 0.546 | 0.487 | 0.609 | 0.541 | 0.797 | 0.770 | 0.783 |
| deepseek-r1-0528_zero | 0.613 | 0.439 | 0.512 | 0.258 | 0.257 | 0.257 | 0.523 | 0.444 | 0.480 | 0.499 | 0.430 | 0.462 |
| deepseek-r1-0528_few | 0.773 | 0.535 | 0.632 | 0.611 | 0.539 | 0.573 | 0.957 | 0.854 | 0.902 | 0.521 | 0.450 | 0.483 |
| deepseek-r1-0528_few path | 0.824 | 0.620 | 0.708 | 0.628 | 0.580 | 0.603 | 0.976 | 0.858 | 0.913 | 0.691 | 0.568 | 0.624 |
| gemini-2.5-pro_zero | 0.746 | 0.559 | 0.639 | 0.437 | 0.468 | 0.452 | 0.390 | 0.525 | 0.448 | 0.551 | 0.872 | 0.676 |
| gemini-2.5-pro_few | 0.744 | 0.564 | 0.642 | 0.464 | 0.508 | 0.485 | 0.431 | 0.585 | 0.497 | 0.993 | 0.949 | 0.970 |
| gemini-2.5-pro_few path | 0.765 | 0.558 | 0.645 | 0.626 | 0.566 | 0.594 | 0.488 | 0.611 | 0.542 | 0.997 | 0.952 | 0.974 |
| gpt-4o_zero | 0.450 | 0.430 | 0.439 | 0.184 | 0.260 | 0.216 | 0.251 | 0.451 | 0.323 | 0.235 | 0.457 | 0.311 |
| gpt-4o_few | 0.459 | 0.423 | 0.441 | 0.323 | 0.397 | 0.357 | 0.278 | 0.453 | 0.344 | 0.243 | 0.461 | 0.318 |
| gpt-4o_few path | 0.481 | 0.436 | 0.458 | 0.396 | 0.436 | 0.415 | 0.321 | 0.509 | 0.393 | 0.364 | 0.571 | 0.445 |
| llama-3.1-405b-instruct_zero | 0.498 | 0.499 | 0.498 | 0.236 | 0.456 | 0.311 | 0.742 | 0.812 | 0.775 | 0.307 | 0.703 | 0.427 |
| llama-3.1-405b-instruct_few | 0.498 | 0.499 | 0.499 | 0.501 | 0.497 | 0.499 | 0.750 | 0.811 | 0.779 | 0.333 | 0.747 | 0.461 |
| llama-3.1-405b-instruct_few path | 1.000 | 0.794 | 0.885 | 0.454 | 0.719 | 0.557 | 0.881 | 0.816 | 0.847 | 0.355 | 0.771 | 0.486 |
| llama-3.1-8b-instruct_zero | 0.526 | 0.484 | 0.505 | 0.147 | 0.224 | 0.178 | 0.145 | 0.308 | 0.197 | 0.070 | 0.308 | 0.114 |
| llama-3.1-8b-instruct_few | 0.633 | 0.488 | 0.551 | 0.402 | 0.425 | 0.413 | 0.202 | 0.431 | 0.276 | 0.083 | 0.358 | 0.135 |
| llama-3.1-8b-instruct_few path | 0.999 | 1.000 | 1.000 | 0.495 | 0.503 | 0.499 | 0.315 | 0.521 | 0.392 | 0.098 | 0.391 | 0.157 |
| phi-4-reasoning-plus_zero | 0.230 | 0.251 | 0.240 | 0.167 | 0.253 | 0.201 | 0.158 | 0.224 | 0.185 | 0.152 | 0.218 | 0.179 |
| phi-4-reasoning-plus_few | 0.287 | 0.279 | 0.283 | 0.204 | 0.291 | 0.240 | 0.176 | 0.243 | 0.204 | 0.167 | 0.246 | 0.199 |
| phi-4-reasoning-plus_few path | 0.325 | 0.346 | 0.335 | 0.283 | 0.365 | 0.319 | 0.259 | 0.335 | 0.292 | 0.239 | 0.319 | 0.273 |
| mixtral-8x7b-instruct_zero | 0.000 | 0.000 | 0.000 | 0.004 | 0.008 | 0.005 | 0.046 | 0.122 | 0.066 | 0.000 | 0.000 | 0.000 |
| mixtral-8x7b-instruct_few | 0.045 | 0.068 | 0.054 | 0.059 | 0.112 | 0.077 | 0.056 | 0.153 | 0.082 | 0.003 | 0.020 | 0.005 |
| mixtral-8x7b-instruct_few path | 0.050 | 0.073 | 0.059 | 0.169 | 0.264 | 0.206 | 0.149 | 0.343 | 0.208 | 0.080 | 0.346 | 0.130 |
| o3-mini_zero | 0.633 | 0.461 | 0.534 | 0.455 | 0.469 | 0.462 | 0.625 | 0.689 | 0.655 | 0.644 | 0.840 | 0.729 |
| o3-mini_few | 0.670 | 0.481 | 0.560 | 0.506 | 0.497 | 0.501 | 0.671 | 0.713 | 0.691 | 0.674 | 0.844 | 0.749 |
| o3-mini_few path | 0.722 | 0.530 | 0.611 | 0.536 | 0.510 | 0.523 | 0.993 | 0.821 | 0.899 | 0.751 | 0.786 | 0.768 |
| gpt-5_zero | 0.905 | 0.664 | 0.766 | 0.954 | 0.954 | 0.954 | 0.886 | 0.892 | 0.889 | 0.914 | 0.951 | 0.932 |
| gpt-5_few | 0.872 | 0.686 | 0.768 | 0.980 | 0.950 | 0.965 | 0.893 | 0.902 | 0.898 | 0.968 | 0.936 | 0.952 |
| gpt-5_few path | 1.000 | 0.689 | 0.816 | 0.999 | 0.994 | 0.997 | 0.912 | 0.899 | 0.905 | 1.000 | 0.992 | 0.996 |
| qwen2.5-7b-instruct_zero | 0.597 | 0.428 | 0.498 | 0.601 | 0.566 | 0.583 | 0.182 | 0.365 | 0.243 | 0.096 | 0.363 | 0.152 |
| qwen2.5-7b-instruct_few | 0.612 | 0.433 | 0.507 | 0.646 | 0.578 | 0.610 | 0.245 | 0.501 | 0.329 | 0.352 | 0.766 | 0.482 |
| qwen2.5-7b-instruct_few path | 0.647 | 0.446 | 0.528 | 0.693 | 0.606 | 0.646 | 0.295 | 0.497 | 0.370 | 0.406 | 0.802 | 0.539 |
| qwen3-235b-a22b_zero | 0.952 | 0.940 | 0.946 | 0.342 | 0.338 | 0.340 | 0.367 | 0.646 | 0.468 | 0.961 | 0.844 | 0.899 |
| qwen3-235b-a22b_few | 0.958 | 0.949 | 0.954 | 0.782 | 0.723 | 0.751 | 0.511 | 0.605 | 0.554 | 0.960 | 0.877 | 0.916 |
| qwen3-235b-a22b_few path | 0.962 | 0.955 | 0.958 | 0.813 | 0.715 | 0.761 | 0.514 | 0.671 | 0.582 | 0.974 | 0.884 | 0.927 |
| grok-4-fast_zero | 0.835 | 0.805 | 0.820 | 0.752 | 0.776 | 0.764 | 0.718 | 0.825 | 0.768 | 0.839 | 0.987 | 0.907 |
| grok-4-fast_few | 0.977 | 0.956 | 0.966 | 0.781 | 0.797 | 0.789 | 0.886 | 0.826 | 0.855 | 0.990 | 0.841 | 0.909 |
| grok-4-fast_few path | 0.979 | 0.965 | 0.972 | 0.814 | 0.801 | 0.808 | 0.986 | 0.897 | 0.940 | 0.978 | 0.967 | 0.973 |

– Preserve all logical relations exactly.

# N  REAL-WORLD BENCHMARK

# O  TASK 1 AND TASK 2 CORRELATION ANALYSIS

As shown in Figure 7, across all modes, Pearson correlation coefficients between Task 1 and Task 2 remain high (mostly > 0.6), which indicates that models that perform well in harder variants also tend to rank highly in easier ones. This suggests that difficulty calibration mainly shifts the absolute performance levels while preserving the relative ordering of models. In other words, all tasks probe a shared underlying reasoning capability at different levels of difficulty, which validates the robustness and coherence of our benchmark design. The correlation analysis results are shown in Figure 7.

# P  REINFORCEMENT LEARNING

# Q  REINFORCEMENT LEARNING EXPERIMENT ON TASK 1

Despite the fine-tuning experiment, we also conducted one RL experiment. We applied TRL von Werra et al. (2020) with GRPO Shao et al. (2024) on Task 1 using two base models (LLaMA-3.1-8b

Table 13: Performance in Task 2 with short Statement Length.

| Model (mode) | Accuracy on 2 Statements | | | Accuracy on 3 Statements | | | Accuracy on 4 Statements | | | Accuracy on 5 Statements | | |
|---|---|---|---|---|---|---|---|---|---|---|---|---|
| | Prec. | Rec. | F1 | Prec. | Rec. | F1 | Prec. | Rec. | F1 | Prec. | Rec. | F1 |
| claude-3.5-haiku_zero | 0.501 | 0.331 | 0.399 | 0.566 | 0.523 | 0.544 | 0.489 | 0.493 | 0.491 | 0.384 | 0.548 | 0.452 |
| claude-3.5-haiku_few | 0.499 | 0.333 | 0.400 | 0.589 | 0.534 | 0.560 | 0.507 | 0.499 | 0.503 | 0.452 | 0.593 | 0.513 |
| claude-3.5-haiku_few path | 0.504 | 0.333 | 0.401 | 0.630 | 0.561 | 0.593 | 0.603 | 0.549 | 0.574 | 0.497 | 0.622 | 0.552 |
| claude-sonnet-4_zero | 0.817 | 0.541 | 0.651 | 0.788 | 0.610 | 0.688 | 0.513 | 0.525 | 0.519 | 0.500 | 0.603 | 0.547 |
| claude-sonnet-4_few | 0.814 | 0.546 | 0.654 | 0.820 | 0.633 | 0.714 | 0.516 | 0.548 | 0.531 | 0.753 | 0.707 | 0.729 |
| claude-sonnet-4_few path | 0.830 | 0.546 | 0.659 | 0.829 | 0.657 | 0.733 | 0.555 | 0.557 | 0.556 | 1.000 | 0.756 | 0.861 |
| deepseek-r1-0528_zero | 0.827 | 0.597 | 0.693 | 0.739 | 0.591 | 0.657 | 0.773 | 0.691 | 0.730 | 0.947 | 0.611 | 0.743 |
| deepseek-r1-0528_few | 0.841 | 0.603 | 0.702 | 0.788 | 0.710 | 0.747 | 0.768 | 0.798 | 0.783 | 0.839 | 0.720 | 0.775 |
| deepseek-r1-0528_few path | 0.889 | 0.817 | 0.852 | 0.784 | 0.730 | 0.756 | 0.789 | 0.800 | 0.795 | 0.913 | 0.780 | 0.841 |
| gemini-2.5-pro_zero | 0.788 | 0.542 | 0.642 | 0.751 | 0.586 | 0.658 | 0.333 | 0.496 | 0.399 | 0.335 | 0.529 | 0.410 |
| gemini-2.5-pro_few | 0.879 | 0.615 | 0.723 | 1.000 | 0.713 | 0.833 | 0.532 | 0.517 | 0.524 | 0.672 | 0.717 | 0.694 |
| gemini-2.5-pro_few path | 0.941 | 0.640 | 0.762 | 1.000 | 0.715 | 0.834 | 0.537 | 0.633 | 0.581 | 0.676 | 0.740 | 0.706 |
| gpt-4o_zero | 0.641 | 0.469 | 0.541 | 0.606 | 0.532 | 0.567 | 0.480 | 0.500 | 0.490 | 0.368 | 0.540 | 0.438 |
| gpt-4o_few | 0.644 | 0.485 | 0.554 | 0.636 | 0.561 | 0.596 | 0.493 | 0.516 | 0.504 | 0.383 | 0.544 | 0.450 |
| gpt-4o_few path | 0.922 | 0.627 | 0.747 | 0.689 | 0.566 | 0.621 | 0.602 | 0.568 | 0.585 | 0.459 | 0.606 | 0.523 |
| llama-3.1-405b-instruct_zero | 0.697 | 0.545 | 0.612 | 0.537 | 0.522 | 0.529 | 0.285 | 0.358 | 0.318 | 0.479 | 0.604 | 0.534 |
| llama-3.1-405b-instruct_few | 0.745 | 0.583 | 0.654 | 0.745 | 0.572 | 0.647 | 0.503 | 0.537 | 0.519 | 0.585 | 0.668 | 0.624 |
| llama-3.1-405b-instruct_few path | 1.000 | 0.747 | 0.855 | 0.996 | 0.666 | 0.798 | 0.504 | 0.544 | 0.524 | 0.664 | 0.690 | 0.677 |
| llama-3.1-8b-instruct_zero | 0.325 | 0.389 | 0.354 | 0.350 | 0.408 | 0.377 | 0.990 | 0.625 | 0.766 | 0.503 | 0.436 | 0.467 |
| llama-3.1-8b-instruct_few | 0.452 | 0.427 | 0.439 | 0.499 | 0.479 | 0.489 | 1.000 | 0.725 | 0.840 | 0.693 | 0.610 | 0.649 |
| llama-3.1-8b-instruct_few path | 0.497 | 0.429 | 0.461 | 0.518 | 0.484 | 0.500 | 1.000 | 0.735 | 0.848 | 0.868 | 0.756 | 0.808 |
| phi-4-reasoning-plus_zero | 0.512 | 0.427 | 0.466 | 0.509 | 0.494 | 0.501 | 0.332 | 0.401 | 0.363 | 0.292 | 0.492 | 0.367 |
| phi-4-reasoning-plus_few | 0.527 | 0.440 | 0.480 | 0.564 | 0.516 | 0.539 | 0.362 | 0.418 | 0.388 | 0.332 | 0.505 | 0.401 |
| phi-4-reasoning-plus_few path | 0.540 | 0.438 | 0.484 | 0.582 | 0.531 | 0.555 | 0.379 | 0.446 | 0.410 | 0.408 | 0.582 | 0.480 |
| mixtral-8x7b-instruct_zero | 0.363 | 0.374 | 0.368 | 0.369 | 0.412 | 0.389 | 0.137 | 0.224 | 0.170 | 0.097 | 0.240 | 0.138 |
| mixtral-8x7b-instruct_few | 0.420 | 0.434 | 0.427 | 0.448 | 0.461 | 0.454 | 0.155 | 0.241 | 0.189 | 0.098 | 0.236 | 0.138 |
| mixtral-8x7b-instruct_few path | 0.750 | 1.000 | 0.857 | 0.502 | 0.487 | 0.494 | 0.168 | 0.260 | 0.204 | 0.190 | 0.383 | 0.254 |
| o3-mini_zero | 0.770 | 0.516 | 0.618 | 0.752 | 0.587 | 0.659 | 0.634 | 0.583 | 0.608 | 0.716 | 0.720 | 0.718 |
| o3-mini_few | 0.782 | 0.533 | 0.634 | 0.800 | 0.627 | 0.703 | 0.669 | 0.581 | 0.622 | 0.803 | 0.753 | 0.777 |
| o3-mini_few path | 0.907 | 0.611 | 0.730 | 0.848 | 0.865 | 0.856 | 0.671 | 0.600 | 0.634 | 0.837 | 0.759 | 0.796 |
| gpt-5_zero | 0.830 | 0.586 | 0.687 | 0.897 | 0.777 | 0.833 | 0.738 | 0.694 | 0.715 | 0.747 | 0.731 | 0.739 |
| gpt-5_few | 0.835 | 0.597 | 0.696 | 0.945 | 0.746 | 0.834 | 0.884 | 0.811 | 0.846 | 0.745 | 0.734 | 0.739 |
| gpt-5_few path | 0.949 | 0.798 | 0.867 | 0.937 | 0.796 | 0.861 | 0.866 | 0.844 | 0.855 | 0.887 | 0.839 | 0.862 |
| qwen2.5-7b-instruct_zero | 0.751 | 0.529 | 0.621 | 0.502 | 0.465 | 0.483 | 0.201 | 0.296 | 0.239 | 0.206 | 0.384 | 0.268 |
| qwen2.5-7b-instruct_few | 0.757 | 0.533 | 0.626 | 0.552 | 0.498 | 0.523 | 0.198 | 0.302 | 0.239 | 0.246 | 0.431 | 0.313 |
| qwen2.5-7b-instruct_few path | 0.797 | 0.539 | 0.643 | 0.600 | 0.521 | 0.558 | 0.249 | 0.353 | 0.292 | 0.254 | 0.441 | 0.322 |
| qwen3-235b-a22b_zero | 0.905 | 0.613 | 0.731 | 0.876 | 0.622 | 0.727 | 0.499 | 0.573 | 0.533 | 0.876 | 0.767 | 0.818 |
| qwen3-235b-a22b_few | 1.000 | 0.717 | 0.835 | 0.884 | 0.754 | 0.814 | 0.579 | 0.641 | 0.608 | 0.882 | 0.766 | 0.820 |
| qwen3-235b-a22b_few path | 0.999 | 0.725 | 0.840 | 0.999 | 0.717 | 0.835 | 0.633 | 0.702 | 0.666 | 0.995 | 0.793 | 0.883 |
| grok-4-fast_zero | 0.748 | 0.508 | 0.605 | 0.857 | 0.727 | 0.787 | 0.693 | 0.600 | 0.643 | 0.886 | 0.796 | 0.839 |
| grok-4-fast_few | 0.793 | 0.538 | 0.641 | 0.907 | 0.715 | 0.800 | 0.772 | 0.643 | 0.702 | 0.899 | 0.828 | 0.862 |
| grok-4-fast_few path | 0.850 | 0.560 | 0.675 | 0.921 | 0.709 | 0.801 | 0.769 | 0.658 | 0.709 | 0.891 | 0.847 | 0.868 |

Table 14: Results on the commonsense version.

| Model | k = 2 | | | | | k = 3 | | | | | k = 4 | | | | | k = 5 | | | | |
|---|---|---|---|---|---|---|---|---|---|---|---|---|---|---|---|---|---|---|---|---|
| | Fmt | Ex | Prec | Rec | F1 | Fmt | Ex | Prec | Rec | F1 | Fmt | Ex | Prec | Rec | F1 | Fmt | Ex | Prec | Rec | F1 |
| grok-4-fast | 0.62 | 0.13 | 0.83 | 0.73 | 0.78 | 0.71 | 0.10 | 0.79 | 0.82 | 0.81 | 0.54 | 0.03 | 0.65 | 0.70 | 0.67 | 0.49 | 0.00 | 0.78 | 0.63 | 0.69 |
| gpt-5 | 0.93 | 0.37 | 0.73 | 0.83 | 0.78 | 0.91 | 0.30 | 0.72 | 0.84 | 0.78 | 0.93 | 0.02 | 0.72 | 0.76 | 0.74 | 0.93 | 0.15 | 0.60 | 0.68 | 0.64 |
| deepseek-r1-0528 | 0.83 | 0.26 | 0.72 | 0.61 | 0.56 | 0.71 | 0.14 | 0.81 | 0.51 | 0.63 | 0.46 | 0.05 | 0.80 | 0.54 | 0.64 | 0.58 | 0.03 | 0.81 | 0.55 | 0.66 |
| claude-sonnet-4 | 0.83 | 0.04 | 0.54 | 0.63 | 0.44 | 0.72 | 0.03 | 0.42 | 0.61 | 0.50 | 0.89 | 0.00 | 0.53 | 0.62 | 0.57 | 0.63 | 0.00 | 0.67 | 0.74 | 0.70 |
| qwen3-235b-a22b | 0.63 | 0.15 | 0.56 | 0.63 | 0.45 | 0.83 | 0.13 | 0.65 | 0.32 | 0.43 | 0.68 | 0.00 | 0.83 | 0.64 | 0.72 | 0.93 | 0.03 | 0.57 | 0.53 | 0.55 |
| gemini-2.5-pro | 0.98 | 0.04 | 0.64 | 0.56 | 0.46 | 0.93 | 0.02 | 0.90 | 0.73 | 0.81 | 0.92 | 0.09 | 0.81 | 0.85 | 0.83 | 0.83 | 0.05 | 0.86 | 0.52 | 0.65 |
| llama-3.1-405b-instruct | 0.63 | 0.04 | 0.76 | 0.76 | 0.74 | 0.73 | 0.00 | 0.53 | 0.63 | 0.58 | 0.65 | 0.00 | 0.51 | 0.67 | 0.58 | 0.59 | 0.00 | 0.60 | 0.62 | 0.61 |
| qwen2.5-7b-instruct | 0.98 | 0.01 | 0.66 | 0.64 | 0.54 | 0.99 | 0.01 | 0.65 | 0.64 | 0.65 | 0.99 | 0.00 | 0.64 | 0.84 | 0.72 | 1.00 | 0.00 | 0.59 | 0.66 | 0.63 |
| phi-4-reasoning-plus | 0.94 | 0.04 | 0.69 | 0.62 | 0.55 | 0.95 | 0.02 | 0.42 | 0.42 | 0.42 | 0.96 | 0.00 | 0.51 | 0.64 | 0.57 | 0.87 | 0.00 | 0.69 | 0.75 | 0.72 |
| mixtral-8x7b-instruct | 1.00 | 0.01 | 0.71 | 0.71 | 0.65 | 0.97 | 0.00 | 0.71 | 0.53 | 0.61 | 0.93 | 0.00 | 0.68 | 0.86 | 0.76 | 1.00 | 0.00 | 0.54 | 0.62 | 0.58 |
| o3-mini | 1.00 | 0.01 | 0.63 | 0.74 | 0.60 | 1.00 | 0.01 | 0.53 | 0.63 | 0.58 | 0.97 | 0.00 | 0.63 | 0.53 | 0.57 | 1.00 | 0.00 | 0.61 | 0.73 | 0.66 |
| claude-3.5-haiku | 1.00 | 0.02 | 0.53 | 0.55 | 0.38 | 0.93 | 0.00 | 0.51 | 0.35 | 0.42 | 0.92 | 0.00 | 0.55 | 0.64 | 0.59 | 0.83 | 0.00 | 0.63 | 0.76 | 0.69 |
| llama-3.1-8b-instruct | 0.53 | 0.01 | 0.85 | 0.53 | 0.58 | 0.42 | 0.00 | 0.52 | 0.33 | 0.41 | 0.38 | 0.00 | 0.55 | 0.64 | 0.59 | 0.43 | 0.00 | 0.80 | 0.56 | 0.66 |
| gpt-4o | 1.00 | 0.04 | 0.53 | 0.52 | 0.36 | 0.98 | 0.02 | 0.53 | 0.62 | 0.57 | 1.00 | 0.00 | 0.62 | 0.52 | 0.56 | 1.00 | 0.00 | 0.80 | 0.63 | 0.71 |

and Qwen-2.5-7b) and found that even small-scale RL using Task 1 data can have **modest** improvement on Task 1, Task 2, and other logical-related tasks, which demonstrates that LogiConBench **can be used as a meaningful reward source**, but the inherent reasoning ability **cannot be easily mitigated** through data augmentation.

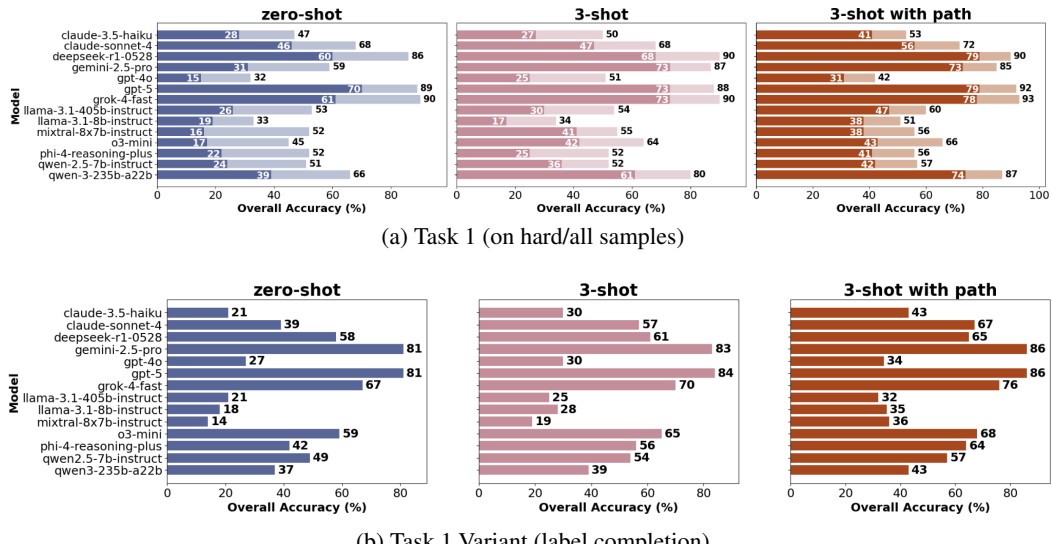

Figure 5: **(a) Task 1 (on hard/all samples)**. Hard samples are cases where the provided lists of Boolean labels come from consistent and inconsistent lists differing in only one element. **(b) Task 1 Variant (label completion)**. One label is missing, and the model must recover it to make the full label list consistent. Light color indicates results on all samples, and dark color is for hard samples.

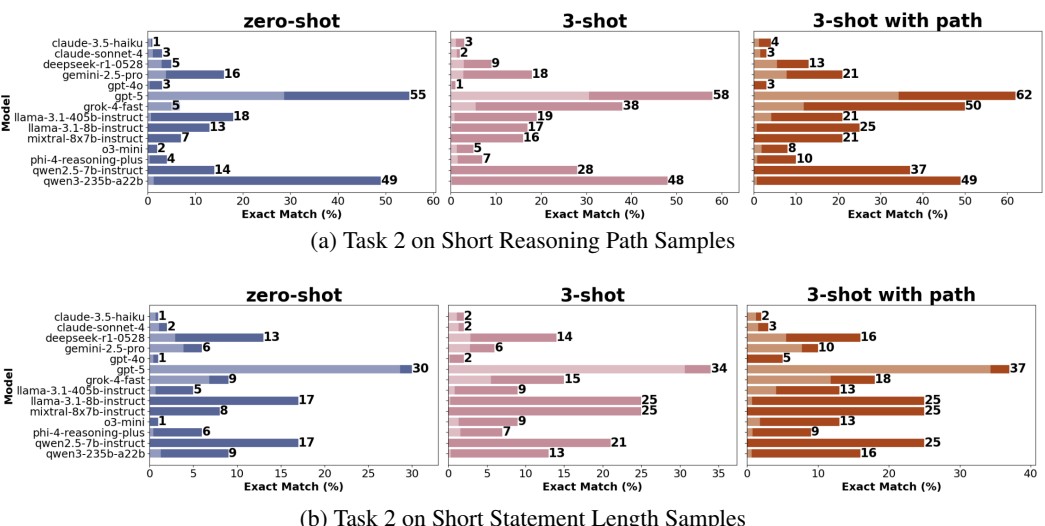

Figure 6: Performance on Task 2 under two easy conditions: (a) short reasoning paths and (b) short statement lengths. The lighter bars indicate the average Exact accuracy score on Task 2.

## Q.1 REWARD MODEL

We define a rule-based reward function that assigns scores based on the alignment between the model's response and the ground-truth label $y \in \{\texttt{correct}, \texttt{incorrect}, \texttt{unknown}\}$. Specifically, let $a$ denote the model's response and $\mathcal{Y} = \{\texttt{correct}, \texttt{incorrect}, \texttt{unknown}\}$. The reward is computed as:

$$r(a, y) = \begin{cases} 1.0 & \text{if } \texttt{contains}(a, y), \\ 0.5 & \text{if } \exists\, y' \in \mathcal{Y} \setminus \{y\}, \texttt{contains}(a, y'), \\ 0.2 & \text{otherwise,} \end{cases} \tag{1}$$

Table 15: Results on the counterfactual version.

| Model | k = 2 | | | | | k = 3 | | | | | k = 4 | | | | | k = 5 | | | | |
|---|---|---|---|---|---|---|---|---|---|---|---|---|---|---|---|---|---|---|---|---|
| | Fmt | Ex | Prec | Rec | F1 | Fmt | Ex | Prec | Rec | F1 | Fmt | Ex | Prec | Rec | F1 | Fmt | Ex | Prec | Rec | F1 |
| grok-4-fast | 0.59 | 0.19 | 0.85 | 0.75 | 0.80 | 0.69 | 0.10 | 0.39 | 0.42 | 0.41 | 0.51 | 0.02 | 0.25 | 0.31 | 0.28 | 0.49 | 0.00 | 0.13 | 0.21 | 0.16 |
| gpt-5 | 0.90 | 0.31 | 0.75 | 0.85 | 0.78 | 0.89 | 0.27 | 0.72 | 0.84 | 0.78 | 0.93 | 0.24 | 0.72 | 0.76 | 0.74 | 0.93 | 0.15 | 0.65 | 0.68 | 0.66 |
| deepseek-r1-0528 | 0.89 | 0.23 | 0.04 | 0.33 | 0.01 | 0.69 | 0.08 | 0.02 | 0.51 | 0.04 | 0.43 | 0.02 | 0.03 | 0.05 | 0.03 | 0.58 | 0.03 | 0.34 | 0.35 | 0.35 |
| claude-sonnet-4 | 0.89 | 0.02 | 0.56 | 0.65 | 0.44 | 0.70 | 0.01 | 0.42 | 0.41 | 0.42 | 0.88 | 0.00 | 0.53 | 0.32 | 0.40 | 0.63 | 0.00 | 0.14 | 0.24 | 0.18 |
| qwen3-235b-a22b | 0.50 | 0.15 | 0.58 | 0.65 | 0.45 | 0.81 | 0.00 | 0.65 | 0.42 | 0.51 | 0.66 | 0.00 | 0.23 | 0.53 | 0.33 | 0.93 | 0.03 | 0.52 | 0.13 | 0.21 |
| gemini-2.5-pro | 0.95 | 0.04 | 0.66 | 0.58 | 0.46 | 0.91 | 0.10 | 0.40 | 0.43 | 0.41 | 0.89 | 0.04 | 0.02 | 0.05 | 0.03 | 0.83 | 0.05 | 0.06 | 0.52 | 0.11 |
| llama-3.1-405b-instruct | 0.50 | 0.01 | 0.38 | 0.28 | 0.12 | 0.71 | 0.00 | 0.53 | 0.63 | 0.58 | 0.63 | 0.00 | 0.53 | 0.36 | 0.43 | 0.59 | 0.00 | 0.33 | 0.32 | 0.32 |
| qwen2.5-7b-instruct | 0.85 | 0.01 | 0.37 | 0.36 | 0.16 | 0.97 | 0.01 | 0.35 | 0.64 | 0.45 | 1.00 | 0.00 | 0.45 | 0.32 | 0.38 | 1.00 | 0.00 | 0.53 | 0.26 | 0.35 |
| phi-4-reasoning-plus | 0.80 | 0.01 | 0.21 | 0.44 | 0.10 | 0.93 | 0.04 | 0.04 | 0.12 | 0.06 | 0.93 | 0.00 | 0.02 | 0.34 | 0.04 | 0.87 | 0.00 | 0.21 | 0.05 | 0.08 |
| mixtral-8x7b-instruct | 0.92 | 0.00 | 0.33 | 0.33 | 0.12 | 0.95 | 0.00 | 0.03 | 0.53 | 0.06 | 0.99 | 0.00 | 0.07 | 0.35 | 0.11 | 1.00 | 0.00 | 0.04 | 0.02 | 0.03 |
| o3-mini | 0.93 | 0.09 | 0.05 | 0.76 | 0.03 | 0.98 | 0.03 | 0.05 | 0.06 | 0.06 | 1.00 | 0.00 | 0.04 | 0.53 | 0.08 | 1.00 | 0.00 | 0.01 | 0.02 | 0.01 |
| claude-3.5-haiku | 0.87 | 0.02 | 0.55 | 0.27 | 0.17 | 0.91 | 0.00 | 0.21 | 0.35 | 0.27 | 0.92 | 0.00 | 0.15 | 0.12 | 0.14 | 0.83 | 0.00 | 0.06 | 0.04 | 0.05 |
| llama-3.1-8b-instruct | 0.50 | 0.01 | 0.27 | 0.55 | 0.17 | 0.40 | 0.00 | 0.52 | 0.33 | 0.41 | 0.35 | 0.00 | 0.06 | 0.03 | 0.04 | 0.43 | 0.00 | 0.06 | 0.06 | 0.06 |
| gpt-4o | 0.87 | 0.04 | 0.55 | 0.54 | 0.36 | 0.96 | 0.03 | 0.05 | 0.03 | 0.04 | 1.00 | 0.00 | 0.03 | 0.32 | 0.06 | 1.00 | 0.00 | 0.03 | 0.05 | 0.04 |

Table 16: Results on the humanized version.

| Model | k = 2 | | | | | k = 3 | | | | | k = 4 | | | | | k = 5 | | | | |
|---|---|---|---|---|---|---|---|---|---|---|---|---|---|---|---|---|---|---|---|---|
| | Fmt | Ex | Prec | Rec | F1 | Fmt | Ex | Prec | Rec | F1 | Fmt | Ex | Prec | Rec | F1 | Fmt | Ex | Prec | Rec | F1 |
| grok-4-fast | 0.82 | 0.23 | 0.75 | 0.72 | 0.74 | 0.73 | 0.11 | 0.69 | 0.62 | 0.65 | 0.62 | 0.09 | 0.70 | 0.61 | 0.65 | 0.53 | 0.04 | 0.70 | 0.61 | 0.65 |
| gpt-5 | 0.98 | 0.29 | 0.65 | 0.63 | 0.64 | 0.91 | 0.20 | 0.73 | 0.83 | 0.78 | 0.93 | 0.13 | 0.68 | 0.83 | 0.75 | 0.90 | 0.09 | 0.68 | 0.83 | 0.75 |
| deepseek-r1-0528 | 0.82 | 0.05 | 0.06 | 0.41 | 0.04 | 0.90 | 0.08 | 0.83 | 0.70 | 0.76 | 0.85 | 0.04 | 0.86 | 0.69 | 0.76 | 0.87 | 0.02 | 0.86 | 0.69 | 0.76 |
| claude-sonnet-4 | 0.87 | 0.04 | 0.46 | 0.53 | 0.38 | 0.82 | 0.06 | 0.63 | 0.62 | 0.62 | 0.82 | 0.05 | 0.58 | 0.63 | 0.60 | 0.85 | 0.00 | 0.58 | 0.63 | 0.60 |
| qwen3-235b-a22b | 0.62 | 0.20 | 0.48 | 0.63 | 0.47 | 0.70 | 0.04 | 0.64 | 0.62 | 0.63 | 0.73 | 0.03 | 0.65 | 0.63 | 0.64 | 0.73 | 0.01 | 0.65 | 0.63 | 0.64 |
| gemini-2.5-pro | 0.85 | 0.05 | 0.61 | 0.32 | 0.42 | 0.90 | 0.18 | 0.79 | 0.73 | 0.76 | 0.92 | 0.12 | 0.82 | 0.75 | 0.78 | 0.97 | 0.06 | 0.82 | 0.75 | 0.78 |
| llama-3.1-405b-instruct | 0.51 | 0.00 | 0.33 | 0.28 | 0.30 | 0.70 | 0.00 | 0.54 | 0.63 | 0.58 | 0.79 | 0.00 | 0.49 | 0.63 | 0.55 | 0.81 | 0.00 | 0.49 | 0.63 | 0.55 |
| qwen2.5-7b-instruct | 0.89 | 0.02 | 0.39 | 0.36 | 0.37 | 0.95 | 0.00 | 0.75 | 0.64 | 0.69 | 0.93 | 0.00 | 0.77 | 0.63 | 0.69 | 0.89 | 0.00 | 0.77 | 0.63 | 0.69 |
| phi-4-reasoning-plus | 0.79 | 0.04 | 0.28 | 0.44 | 0.34 | 0.92 | 0.05 | 0.84 | 0.82 | 0.83 | 0.90 | 0.02 | 0.85 | 0.81 | 0.83 | 0.93 | 0.01 | 0.85 | 0.81 | 0.83 |
| mixtral-8x7b-instruct | 0.92 | 0.03 | 0.33 | 0.33 | 0.33 | 0.96 | 0.00 | 0.82 | 0.54 | 0.65 | 0.92 | 0.00 | 0.78 | 0.55 | 0.64 | 0.94 | 0.00 | 0.78 | 0.55 | 0.64 |
| o3-mini | 0.97 | 0.05 | 0.05 | 0.76 | 0.09 | 0.97 | 0.07 | 0.65 | 0.69 | 0.67 | 0.88 | 0.03 | 0.65 | 0.68 | 0.66 | 0.90 | 0.00 | 0.65 | 0.68 | 0.66 |
| claude-3.5-haiku | 0.89 | 0.00 | 0.55 | 0.27 | 0.36 | 0.90 | 0.02 | 0.82 | 0.55 | 0.66 | 0.89 | 0.00 | 0.87 | 0.55 | 0.68 | 0.87 | 0.00 | 0.87 | 0.55 | 0.68 |
| llama-3.1-8b-instruct | 0.62 | 0.01 | 0.27 | 0.55 | 0.36 | 0.60 | 0.06 | 0.53 | 0.43 | 0.47 | 0.67 | 0.03 | 0.53 | 0.42 | 0.47 | 0.69 | 0.00 | 0.53 | 0.42 | 0.47 |
| gpt-4o | 0.87 | 0.04 | 0.55 | 0.54 | 0.54 | 0.98 | 0.04 | 0.72 | 0.81 | 0.76 | 0.92 | 0.02 | 0.76 | 0.80 | 0.78 | 0.93 | 0.00 | 0.76 | 0.80 | 0.78 |

where contains$(a, y)$ indicates that the response $a$ contains the label $y$. Intuitively, a full reward of 1.0 is assigned when the response matches the ground-truth label, a partial reward of 0.5 is given when the response contains a plausible but incorrect label, and a default reward of 0.2 is assigned to all other responses.

## Q.2  EVALUATION RESULTS

**Table 1** reports Task 1 accuracy for both LLaMA and Qwen across different values of $k$, measured on consistent, inconsistent, and overall subsets.

**Tables 2 and 3** present Task 2 results for LLaMA and Qwen after RL training, evaluated on format accuracy, exact match, precision, recall, and F1.

**Table 4** compares general downstream performance before and after RL fine-tuning.

## Q.3  OVERALL ANALYSIS

RL training yields modest but consistent gains on Task 1 across most values of $k$, while Task 2 benefits only marginally, suggesting limited transfer to enumerative reasoning. On broader downstream benchmarks including LogiQA, LogicNLI, MathQA, HumanEval, and MMLU, improvements remain minimal or inconsistent, indicating little generalization beyond the training distribution. We attribute these limited gains to the relatively low task complexity and the tendency of RL to exploit surface-level reward patterns rather than develop deeper reasoning capabilities. Nevertheless, the fact that both RL and fine-tuning produce improvements on our benchmark suggests that LogiCon-Bench can serve as a useful training resource for enhancing logical reasoning in language models.

Table 17: Performance on downstream benchmarks.

| Model | LiveCodeBench | Infinite | AIME | AA-LCR | ACEBench (agent) |
|---|---|---|---|---|---|
| grok-4-fast | 79.00 | 65.80 | 89.70 | 64.70 | 73.00 |
| gpt-5 | 84.00 | 86.50 | 91.70 | 72.80 | 78.00 |
| deepseek-r1-0528 | 64.30 | 36.50 | 68.00 | 52.30 | 64.00 |
| claude-sonnet-4 | 55.90 | 64.60 | 74.30 | 64.70 | 53.00 |
| qwen3-235b-a22b | 79.00 | 53.20 | 91.00 | 67.00 | 51.00 |
| gemini-2.5-pro | 69.00 | 54.10 | 87.70 | 66.00 | 63.00 |
| llama-3.1-405b-instruct | 30.50 | 19.00 | **33.00** | 24.30 | 41.00 |
| qwen2.5-7b-instruct | 12.60 | **21.00** | **9.00** | **16.00** | **12.00** |
| phi-4-reasoning-plus | 23.10 | **32.00** | 21.00 | **20.00** | **15.00** |
| mixtral-8x7b-instruct | 6.60 | **13.00** | 3.00 | 8.00 | 6.00 |
| o3-mini | 71.70 | 28.70 | **25.00** | **30.00** | **65.00** |
| claude-3.5-haiku | 20.20 | **28.00** | 21.00 | 43.00 | 35.00 |
| llama-3.1-8b-instruct | 11.60 | 16.40 | 4.50 | 15.70 | 4.00 |
| gpt-4o | 42.50 | 25.10 | 25.70 | 53.00 | 71.50 |

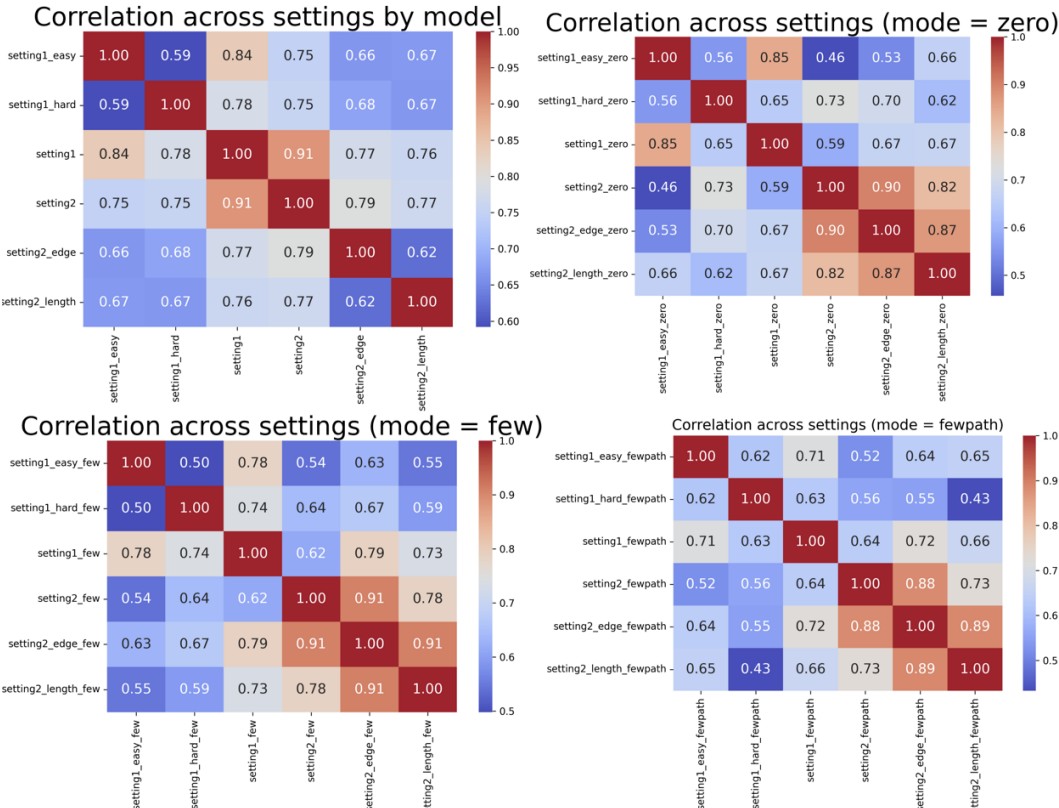

Figure 7: Correlations between different tasks.

# R FINE-TUNE

## R.1 FINE-TUNING IMPROVES PERFORMANCE BUT THE GAINS ARE LIMITED

We conducted a controlled fine-tuning experiment and found that although fine-tuning yields measurable improvements on the benchmark, the gains are fundamentally limited. Meanwhile, the storage and computation required for fine-tuning are substantial. These results suggest that the perfor-

Table 18: Accuracy on Task 1 (Consistent / Inconsistent / Overall).

| Model | k | Con | Incon | Overall |
|-------|---|------|--------|---------|
| LLaMA | 2 | 0.58 | 0.59 | 0.585 |
|       | 3 | 0.51 | 0.52 | 0.515 |
|       | 4 | 0.32 | 0.48 | 0.40 |
|       | 5 | 0.31 | 0.33 | 0.32 |
| Qwen  | 2 | 0.42 | 0.5436 | 0.4818 |
|       | 3 | 0.47 | 0.5308 | 0.5004 |
|       | 4 | 0.33 | 0.4673 | 0.3987 |
|       | 5 | 0.30 | 0.3406 | 0.3203 |

Table 19: Results on Task 2 after RL training.

| Model | k | Format | Exact | Precision | Recall | F1 |
|-------|---|--------|-------|-----------|--------|------|
| LLaMA | 2 | 0.691 | 0.285 | 0.7384 | 0.7962 | 0.7662 |
|       | 3 | 0.704 | 0.176 | 0.3894 | 0.8189 | 0.5278 |
|       | 4 | 0.613 | 0.104 | 0.5843 | 0.7320 | 0.6499 |
|       | 5 | 0.642 | 0.050 | 0.4830 | 0.5718 | 0.5237 |
| Qwen  | 2 | 0.989 | 0.183 | 0.4829 | 0.4899 | 0.4860 |
|       | 3 | 1.000 | 0.145 | 0.6348 | 0.4138 | 0.5010 |
|       | 4 | 0.977 | 0.058 | 0.4342 | 0.4280 | 0.4312 |
|       | 5 | 0.992 | 0.030 | 0.2143 | 0.2038 | 0.2090 |

mance bottleneck primarily lies in the models' inherent reasoning limitations rather than insufficient supervision.

### R.1.1 FINE-TUNING ON TASK 2 PROVIDES IMPROVEMENTS, BUT FAR FROM SOLVING THE TASK

We fine-tuned three small open-source models (Llama-3-8B, Qwen-2.5-7B, and Mistral-7B) on 1,000 synthetic Task 2 training samples generated by our pipeline (100 for $k=2$, 200 for $k=3$, 300 for $k=4$, and 400 for $k=5$). Each training instance contains both the full reasoning path and the final answer, providing complete in-domain supervision perfectly aligned with the evaluation format.

Across all models, fine-tuning improves Task 2 performance, but the gains remain limited (Table 21). Crucially, even after fine-tuning, none of the small models approach frontier-model performance.

### R.1.2 FINE-TUNING DOES NOT OVERFIT: IT GENERALIZES TO TASK 1 AND INDEPENDENT BENCHMARKS

Despite being trained solely on Task 2, all models exhibit improvements on Task 1 (Table 22). Since Task 1 involves a distinct output format and only partially overlapping reasoning skills, this demonstrates genuine reasoning transfer rather than overfitting.

Fine-tuned models also show improvements on multiple independent logical reasoning benchmarks (Table 23), including LogicNLI, LogiQA, MathQA, HumanEval, and MMLU—none of which appear in the training data. This demonstrates that synthetic data strengthens models' broader logical competence.

### R.2 SUBSTANTIAL STORAGE AND COMPUTATIONAL COSTS OF FINE-TUNING

Fine-tuning is not only limited in performance gains, but also extremely expensive in storage and computation.

**Storage Cost.** Each training instance requires storing its (i) symbolic representation, (ii) reasoning edges, (iii) atom expressions, (iv) consistency and non-consistency sets, and (v) the full logical graph. Since the logical graph grows with $\mathcal{O}(n^3)$ complexity as the number of atoms increases, the storage footprint escalates rapidly.

Table 20: General Downstream Evaluation (Before vs. After RL)

| Benchmark | LLaMA | LLaMA-FT | Qwen | Qwen-FT |
|-----------|-------|----------|------|---------|
| LogiQA | 39.6 | 42.1 | 33.0 | 32.5 |
| LogicNLI | 28.5 | 30.4 | 24.0 | 24.4 |
| MathQA | 42.8 | 38.2 | 36.0 | 33.6 |
| HumanEval | 22.6 | 25.3 | 32.9 | 33.8 |
| MMLU | 65.3 | 60.9 | 55.0 | 57.9 |

Table 21: Fine-tuning results on Task 2 across three models.

| Model | k | Format | Exact | Precision | Recall | F1 |
|-------|---|--------|-------|-----------|--------|-----|
| Llama-3 | 2 | 0.783 | 0.327 | 0.5626 | 0.9362 | 0.7028 |
| | 3 | 0.802 | 0.190 | 0.4457 | 0.9749 | 0.6117 |
| | 4 | 0.736 | 0.132 | 0.4349 | 0.5556 | 0.4879 |
| | 5 | 0.669 | 0.080 | 0.4247 | 0.4419 | 0.4331 |
| Qwen-2.5 | 2 | 1.00 | 0.213 | 0.6723 | 0.5773 | 0.6212 |
| | 3 | 1.00 | 0.161 | 0.7587 | 0.4635 | 0.5754 |
| | 4 | 1.00 | 0.066 | 0.4294 | 0.4593 | 0.4438 |
| | 5 | 1.00 | 0.063 | 0.3277 | 0.3393 | 0.3340 |
| Mistral-7B | 2 | 1.00 | 0.122 | 0.4307 | 0.8447 | 0.5705 |
| | 3 | 1.00 | 0.113 | 0.3326 | 0.2312 | 0.2728 |
| | 4 | 1.00 | 0.047 | 0.2626 | 0.2362 | 0.2487 |
| | 5 | 1.00 | 0.000 | 0.1457 | 0.2749 | 0.1905 |

**Computational Cost.** Even with only 1,000 training examples, fine-tuning an 8B-parameter model requires an `A800` GPU for approximately 4 hours. Achieving stronger performance would require orders of magnitude more computation, making fine-tuning impractical at scale.

Table 22: Fine-tuning on Task 2 and evaluating on Task 1.

| Model | k | Cons | Incons | Overall |
|---|---|---|---|---|
| Llama-3 | 2 | 0.69 | 0.47 | 0.58 |
| | 3 | 0.46 | 0.53 | 0.495 |
| | 4 | 0.52 | 0.35 | 0.435 |
| | 5 | 0.61 | 0.33 | 0.47 |
| Qwen-2.5 | 2 | 0.43 | 0.52 | 0.48 |
| | 3 | 0.63 | 0.62 | 0.63 |
| | 4 | 0.67 | 0.79 | 0.73 |
| | 5 | 0.64 | 0.78 | 0.71 |
| Mistral-7B | 2 | 0.81 | 0.61 | 0.71 |
| | 3 | 0.74 | 0.48 | 0.61 |
| | 4 | 0.85 | 0.38 | 0.615 |
| | 5 | 0.73 | 0.42 | 0.575 |

Table 23: Performance on general benchmarks with and without fine-tuning.

| Benchmark | Llama | Llama-FT | Qwen | Qwen-FT | Mistral | Mistral-FT |
|---|---|---|---|---|---|---|
| LogiQA | 39.6 | 45.2 | 33.0 | 35.2 | 34.0 | 33.0 |
| LogicNLI | 28.5 | 47.5 | 24.0 | 30.6 | 26.0 | 26.0 |
| MathQA | 42.8 | 49.6 | 36.0 | 28.0 | 33.0 | 36.2 |
| HumanEval | 22.6 | 29.3 | 32.9 | 41.0 | 28.7 | 32.0 |
| MMLU | 65.3 | 69.3 | 55.0 | 58.0 | 64.2 | 67.0 |

