# OpenReview forum: "LogiConBench: Benchmarking Logical Consistencies of LLMs"
_ICLR.cc/2026/Conference — ICLR 2026 Poster_

### Official Review · Reviewer_VUDU · 2025-10-24

**Soundness:** 3
**Presentation:** 2
**Contribution:** 4
**Rating:** 8
**Confidence:** 4

**Summary:**

The paper introduces LogiConBench, a scalable benchmark for testing logical consistency in LLMs. A four‑stage pipeline first generates logical graphs from natural‑deduction rules, then samples some target nodes under bounded distance, propagates truth values along extracted reasoning paths to derive all consistent/inconsistent label lists, and finally verbalizes symbolic formulas into natural language via templates and WordNet substitutions. The authors evaluate 14 models, empirically finding that frontier models reach 85–95% on Task 1, whereas Task 2 remains difficult.

**Strengths:**

* Data‑generation procedure appears methodologically sound: edge‑level truth‑compatibility for each operator is explicitly specified; Steiner‑tree extraction bounds depth while preserving connectivity among targets; the DFS enumerator yields exact, reproducible labels; and NL verbalization is decoupled from labeling, isolating the resulting problems from commonsense knowledge.
* Very comprehensive evaluation across many models and many difficulty slices: 14 LLMs (open/proprietary), 2–5 statements, with rich metrics.
* Even frontier models remain far from solved on Task 2: most models have Exact < 1%; best is gpt‑5 Exact≈0.51 with reasoning paths, while prior datasets are near‑ceiling (Figure 1a). This makes LogiConBench a timely and progress‑sensitive benchmark in an era when many others have saturated

**Weaknesses:**

* It is hard to gauge how diverse the generated logical problem patterns are. The study used the introduction rules for five logical operators but did not use elimination rules, which are also crucial in symbolic logic. Is it OK?, how diverse are the resulting patterns? Although we understand that defining and measuring the diversity of problem patterns is subtle and challenging.

**Questions:**

* A key advantage of your framework is that it can scale to generate unlimited data. Given this, is it possible to leverage large‑scale RL on synthetically generated splits to train, rather than benchmarking, reasoning capabilities?
* If I understand the tasks correctly: If we had a reliable Task 1 verifier, then in principle Task 2 could be solved by evaluating all 2^k assignments with Task 1 and collecting the consistent ones. Why do LLMs still struggle so much with Task 2? Is the main bottleneck long‑horizon search/planning, cumulative local errors, output‑formatting limits, or something else?

---

> ### Author Response · Authors · 2025-11-21
> **Response to Weakness 1 (Part 1/2)**
>
> We sincerely appreciate the reviewer’s great efforts and insightful comments to improve our manuscript. We thank the reviewer for recognizing our framework and benchmark design. In below, we address these concerns point by point and try our best to update the manuscript accordingly.
>
> > **W1.1:** It is hard to gauge how diverse the generated logical problem patterns are.
>
> Empirically, each sample of the generated logical problem has a distinct pattern, as Table 1 shows. To further resolve your concern, we added some theoretical analysis on the generation algorithm.
>
> Firstly, we make some formal definitions.
> - Let $\mathcal{A}$ be the finite set of atomic propositions (e.g., $p, q, u$) from the pre-defined atom list.
> - Let $\mathcal{O}_{node} = \{\neg, \wedge, \vee\}$ be the set of node-generating logical operators explicitly used to create new propositions in the graph.
> - Let $\mathcal{R}_{rewrite}$ be the set of logical rewrite rules (e.g., CNF, DNF, implication elimination) applied by the framework.
> - Let $P$ be the iterative graph generation process described in Section 3.1 4. This process starts from an atom $p \in \mathcal{A}$ and iteratively expands nodes.
> - Let $V_{gen} = \bigcup_{t=0}^{\infty} V_t$ be the set of all symbolic propositions (nodes) generatable by the process $P$, which "allows the Logical Graph to grow indefinitely".
> - Let $V_{final} = V_{gen} \cup \mathcal{R}_{rewrite}(V_{gen})$ be the complete set of all propositions generatable by the framework, including rewritten forms.
> - Let $\mathcal{F}(\mathcal{A})$ be the set of all well-formed formulas (WFFs) of propositional logic constructible from atoms in $\mathcal{A}$ using the standard connectives $\{\neg, \wedge, \vee, \rightarrow, \leftrightarrow\}$.
>
>
> **Proposition 1**: Generative Completeness and Unboundedness
>
> The LogiConBench framework, defined by the processes $P$ and $\mathcal{R}_{rewrite}$, generates a final proposition set $V_{final}$ with the following properties:
> 1. (Functional Completeness) The framework is functionally complete. For any formula $\psi \in \mathcal{F}(\mathcal{A})$, the set $V_{final}$ contains at least one formula $\varphi$ that is logically equivalent to $\psi$ (i.e., $\varphi \equiv \psi$).
> 2. (Structural Unboundedness) The set $V_{final}$ is unbounded in structural complexity. It is a countably infinite set, and for any finite complexity measure $N$ (e.g., number of operators or formula depth), there exists a formula $\varphi \in V_{final}$ whose complexity exceeds $N$.
>
> ---
>
> Proof
>
> Proof of Functional Completeness:
> - The set of logical operators $\{\neg, \wedge, \vee\}$ is axiomatically known to be functionally complete for propositional logic.
> - The LogiConBench node-generation process $P$ explicitly uses Conjunction ($\wedge$) and Disjunction ($\vee$) on nodes, as well as Negation ($\neg$).
> - Furthermore, the framework's rewriting step, $\mathcal{R}_{rewrite}$, explicitly applies rules for "conversion to conjunctive normal form (CNF) or disjunctive normal form (DNF)".
> - By definition, any WFF $\psi \in \mathcal{F}(\mathcal{A})$ has a logically equivalent CNF and DNF, which are constructed using only the operators $\{\neg, \wedge, \vee\}$.
> - Since the framework both generates formulas using this complete set of operators and rewrites formulas into equivalent forms (like CNF/DNF) using the same operators, it can, in principle, construct a logically equivalent representation $\varphi$ for any arbitrary formula $\psi$.
> - Therefore, the framework is functionally complete.
>
>
> Proof of Structural Unboundedness:
> - Let the complexity of a formula $\varphi$, denoted $C(\varphi)$, be the count of logical operators from $\mathcal{O}_{node}$ it contains.
> - The graph generation process $P$ is explicitly defined as iterative and expansive. It begins with an atomic proposition $p$ (where $C(p) = 0$).
> - Subsequently, each newly added node can be further expanded.
> - This expansion involves applying an operator from $\mathcal{O}_{node}$. For example, expanding $\varphi_t$ (with complexity $C(\varphi_t) = k$) to $\varphi_{t+1} = \varphi_t \wedge \alpha$ (where $\alpha \in \mathcal{A}$) results in a new formula with complexity $C(\varphi_{t+1}) = k+1$.
> - The framework documentation explicitly states that this iterative application "allows the Logical Graph to grow indefinitely".
> - Therefore, for any finite complexity bound $N \in \mathbb{N}$, the iterative generation process $P$ can be applied $N+1$ times to produce a formula $\varphi \in V_{gen}$ such that $C(\varphi) > N$.
> - Since $V_{gen} \subset V_{final}$, the final set $V_{final}$ is also unbounded in structural complexity and is thus a countably infinite set.

---

> ### Author Response · Authors · 2025-11-21
> **Response to Weakness 1 (Part 2/2)**
>
> >**W1.2:** The study used the introduction rules for five logical operators but did not use elimination rules, which are also crucial in symbolic logic. Is it OK?, how diverse are the resulting patterns? Although we understand that defining and measuring the diversity of problem patterns is subtle and challenging.
>
> Thank you for pointing out this issue, which we addressed with greater clarity in the revised paper. In fact, to construct diverse formulas (i.e., logical problem patterns), **we used five introduction rules as well as important elimination rules.**
>
> Our process began by using the five introduction rules to **build** the initial logic graph. After extracting the candidate formulas (the logical problem patterns) from the nodes, we engaged in a **rewriting phase** to enhance their diversity. During this phase, we applied common feasible elimination rules, supplemented by additional equivalence transformations such as those for NNF, CNF, and DNF, (detailed in Appendix G)
> to ensure that we achieved a sufficient variety of formulas.
>
> After constructing the logic graph with **introduction rules** and **rewriting with elimination** and other rules, we achieved sufficient formula diversity. These formulas were then translated into natural language samples.
>
> Through this approach, we believe we have maximized the diversity of the logical problem patterns to the greatest extent operationally practical.

---

> ### Author Response · Authors · 2025-11-21
> **Response to Question 1**
>
> >**Q1:** A key advantage of your framework is that it can scale to generate unlimited data. Given this, is it possible to leverage large‑scale RL on synthetically generated splits to train, rather than benchmarking, reasoning capabilities?
>
> Thank you for your valuable suggestion! We applied TRL[1] with GRPO[2] on Task 1 using two base models (LLaMA-3.1-8b and Qwen-2.5-7b) and found that even small-scale RL using Task 1 data can have **modest** improvement on Task 1, Task 2, and other logical related tasks, which demonstrate that LogiConBench can be used as a meaningful reward source.
>
> **1. The Reward model can be formalized as follows.**
>
> When the correct answer is "unknown":
> - Answering "correct" or "incorrect": 0.5 points
> - Answer contains "unknown": 1 points
> - All other answers: 0.2 points
>
> When the correct answer is "correct":
> - Answering "unknown" or "incorrect": 0.5 points
> - Answer contains "correct": 1 points
> - All other answers: 0.2 points
>
> When the correct answer is "incorrect":
> - Answering "unknown" or "correct": 0.5 points
> - Answer contains "incorrect": 1 points
> - All other answers: 0.2 points
>
>
> **2. The evaluation results** are summarized as follows: Table 1 reports the RL model’s performance on Task 1, while Tables 3 and 4 present its performance on Task 2 and on other logic-related benchmarks.
>
> **Table 1. Task 1 Accuracy (Consistent / Inconsistent / Overall)**
>
> **LLaMA**
>
> |k|con|incon|overall|
> |--|--|--|--|
> |2|0.58|0.59|0.585|
> |3|0.51|0.52|0.515|
> |4|0.32|0.48|0.4|
> |5|0.31|0.33|0.32|
>
> **Qwen**
>
> |k|con|incon|overall|
> |--|--|--|--|
> |2|0.42|0.5436034521|0.4818017261|
> |3|0.47|0.5308301172|0.5004150586|
> |4|0.33|0.4673295095|0.3986647547|
> |5|0.3|0.3405696894|0.3202848447|
>
> **Table 2. RL LLaMA on Task 2**
>
> |k|format|exact|precision|recall|f1|
> |--|--|--|--|--|--|
> |2|0.691|0.285|0.7384|0.7962|0.7662|
> |3|0.704|0.176|0.3894|0.8189|0.5278|
> |4|0.613|0.104|0.5843|0.732|0.6499|
> |5|0.642|0.05|0.483|0.5718|0.5237|
>
> **Table 3. RL on Task 1 — Qwen Results**
>
> |k|format|exact|precision|recall|f1|
> |--|--|--|--|--|--|
> |2|0.989|0.183|0.4829|0.4899|0.486|
> |3|1|0.145|0.6348|0.4138|0.5010|
> |4|0.977|0.058|0.4342|0.428|0.4312|
> |5|0.992|0.03|0.2143|0.2038|0.209|
>
>
> **Table 4. General Downstream Evaluation (Before vs After RL Fine-tuning on Task 1)**
>
> |benchmark|llama|llama ft|qwen|qwen ft|
> |--|--|--|--|--|
> |LogiQA [4] |39.6|42.1|33|32.5|
> |LogicNLI [3] |28.5|30.4|24|24.4|
> |MathQA [6] |42.8|38.2|36|33.6|
> |HumanEval [7] |22.6|25.3|32.9|33.8|
> |MMLU [5] |65.3|60.9|55|57.9|
>
>
>
> **3. Overall Analysis**
>
> * **3.1 RL yields modest but consistent gains on Task 1 (and slight gains on Task 2).** RL improves Task 1 performance across most k values, though the gains are small. Task 2 benefits only marginally, which indicates weak transfer to enumerative reasoning.
>
> * **3.2 Little generalization to broader benchmarks.** On LogiQA, LogicNLI, MathQA, HumanEval, and MMLU, improvements are minimal or inconsistent, and some tasks even regress.
>
> * **3.3 Improvements are limited** since to low task complexity for a simple discrimination, and so the reward-pattern learning limit might deep reasoning improvements.
>
> * **3.4 The positive result** of the RL experiment and the fine-tuning experiment replied to the reviewer qtmj shows a sign that LogiConBench can indeed serve as a **useful training dataset** for improving reasoning capabilities.
>
> [1] Leandro et al. "TRL: Transformer Reinforcement Learning." 2020.
>
> [2] Shao et al. "Group Relative Policy Optimization". 2024.
>
> [3] Tian et al. "Diagnosing the First-Order Logical Reasoning Ability Through LogicNLI". In EMNLP 2021.
>
> [4] Liu et al. "LogiQA: A Challenge Dataset for Machine Reading Comprehension with Logical Reasoning." In IJCAI 2020.
>
> [5] Hendrycks et al. "Measuring Massive Multitask Language Understanding." In ICLR 2021.
>
> [6] Amini et al. "MathQA: Towards Interpretable Math Word Problem Solving with Operation-Based Formalisms." In NAACL 2019.
>
> [7] Chen et al. "HumanEval: Hand-Written Evaluation Set." In LREC-COLING 2024.

---

> ### Author Response · Authors · 2025-11-21
> **Response to Question 2**
>
> > **Q2:** If I understand the tasks correctly: If we had a reliable Task 1 verifier, then in principle Task 2 could be solved by evaluating all 2^k assignments with Task 1 and collecting the consistent ones. Why do LLMs still struggle so much with Task 2? Is the main bottleneck long‑horizon search/planning, cumulative local errors, output‑formatting limits, or something else?
>
> Thank you for pointing it out. **Yes**, your understanding is correct. The reason why LLMs still struggle can be found from the qualitative analysis as follows.
>
> 1. **Patterns between large and small models.**
> * More than 80% of outputs from the **Large models** (like gpt-5, claude-sonnet-4, and gemini-2.5-pro), **translate the problem into a symbolic form** first, then reason logically. They are very good at the short-distance reasoning, with a high accuracy rate on these individual logical steps.
>
> * **Smaller models** (like Llama-3.1-8b, Mistral-7b) follow a different approach. They tend to **paraphrase the problem step-by-step** using a specific framework, but without clarifying the logic. And then they **suddenly jump to a final answer**. This explains their moderate F1 scores but very low accuracy.
>
> Below, we summarize the main ways large models still fail. We first analyze the overlapping error patterns, and then we analyze the different patterns across different tasks.
>
> 2. **Overlapping Error Patterns**
>
> Across tasks, large models share a common workflow: they break down the input into symbols and reason about them. However, their approach is often short-sighted.
>
> *   In a **3-shot-with-path** setup, they systematically explore different possibilities.
> *   In **zero-shot** or plain **3-shot** setups, they only test a **few quick examples** instead of exploring all combinations.
>
> This leads to three recurring errors:
>
> 2.1  **Incomplete Enumeration:** The model checks only a small subset of possibilities, which is insufficient to solve the problem.
>
> 2.2  **One-Way Simplification:** The model correctly translates and reasons with the symbols but **fails to map the results back** to the original problem, leading to the wrong final answer.
>
> 2.3  **Lost Goals:** The model applies logic correctly at each step but **forgets the main objective**. It gets lost in the details and never makes the final required decision.
>
> 3. **Divergent Errors Showcased by Each Task**
>
> The two tasks are not redundant; they expose different weaknesses.
>
> *   **Task 1: Lack of Counterfactual Exploration**
> To determine if a statement is forced, a model must consider **alternative scenarios** (e.g., what if 'p' were true? what if it were false?). Small models rarely do this. They typically fix on one scenario early and declare an answer without checking if others are possible.
>
> *   **Task 2: Semantic Drift**
> Models start by using atoms as a tool for reasoning, but end up outputting the atom labels, which mistakes correct intermediate atom assignments for the final answer.
> This shows a struggle with **global reasoning planning**, which is a different challenge from the one in Task 1.
> ----
> We hope the above discussion will fully address your concerns about our work. We look forward to your insightful and constructive responses to further help us improve the quality of our work. Thank you!

---

### Official Review · Reviewer_jijm · 2025-10-30

**Soundness:** 3
**Presentation:** 2
**Contribution:** 2
**Rating:** 6
**Confidence:** 3

**Summary:**

This paper introduces LogiConBench, a new benchmark designed to evaluate the logical consistency of Large Language Models (LLMs). The authors argue that existing benchmarks for logical consistency are no longer sufficient, as they lack scalability, diversity, and challenge; SOTA models, for instance, already achieve over 95% accuracy on them.
To address this, LogiConBench provides a framework including:
1. Automatically constructing graphs where nodes represent symbolic propositions and edges represent reasoning relations.
2. It samples sets of propositions (nodes) from these graphs, extracts the reasoning paths connecting them, and determines all possible Boolean (True/False) assignments that are logically consistent.
3. These symbolic propositions are then translated into diverse natural language sentences using templates and lexical substitution from WordNet.

The paper releases a 280K-sample corpus and evaluates 14 frontier LLMs on Discriminative(The model must determine if a given list of Boolean labels for a set of statements leads to a contradiction) and Enumerative(The model must enumerate all possible lists of Boolean labels that make a set of statements consistent) tasks.

**Strengths:**

1. The paper convincingly argues that existing benchmarks for logical consistency are "saturated". The results shown in Figure 1a demonstrate that SOTA models easily pass older benchmarks (like BeliefBank and LFC) but are significantly challenged by LogiConBench, proving its necessity.
2. Unlike static, human-written datasets, the LogiConBench framework is automatic and can generate "unlimited logical rule combinations". This graph-based generation pipeline is a novel approach that ensures a large-scale, diverse, and renewable source of evaluation data.
3. A key feature is the benchmark's ability to provide "controllable-depth graphs with explicit reasoning paths". The authors show that providing these paths as prompts ("few-shot with reasoning paths") improves model performance, especially on the difficult Enumerative task. This highlights the value of explainability and chain-of-thought data.
4. The study is thorough. It tests 14 frontier closed and open-source models across multiple evaluation settings (zero-shot, few-shot, and few-shot with path).

**Weaknesses:**

1. The natural language generation relies on templates and lexical substitution. While this allows for scale, the resulting sentences may lack the nuance, ambiguity, and diversity of "fully human-authored datasets". This could make it easier for models to map the text back to symbolic logic, potentially simplifying the natural language aspect of the reasoning. At the same time, it is hard not to suspect that the constructed natural language data contains content that is consistent with or contrary to the real world, such as "the sky is blue". This situation may cause problems such as reasoning shortcuts or knowledge contradictions, affecting the evaluation results.
2. Meanwhile, I believe that the task form of Enumerative task cannot well reflect the inherent logical ability of the model, because it is well known that exhaustive search has always been a serious problem for LLMs, but this does not completely mean that the model's logical level is low.
3. The lack of detailed and profound analysis of the different performances of the model on various tasks makes it difficult to reflect the necessity and value of designing these two types of tasks.

**Questions:**

Refer to weaknesses.

---

> ### Author Response · Authors · 2025-11-21
> **Response to Weakness 1 (Part 1/3)**
>
> Thank you for your valuable feedback!
>
> > **W1.1:** The natural language generation relies on templates and lexical substitution. While this allows for scale, the resulting sentences may lack the nuance, ambiguity, and diversity of "fully human-authored datasets". This could make it easier for models to map the text back to symbolic logic, simplifying the natural language aspect of the reasoning.
>
> We constructed a **"human-style rephrasing"** variant of the dataset. Specifically, for each (k)-statement group, we sampled 100 instances and asked gpt-4o to paraphrase every sentence into more natural English. The prompt is shown below.
>
> **Rephrasing prompt used:**
>
> ```
> Please rewrite the following sentences into natural human-style English.
>
> Requirements:
> - Do NOT keep any numeric labels (no "1)", "2)", "First,…", etc.).
> - Do NOT place "not" or any negation operator at the beginning of a sentence.
>   Negation must appear inside the clause in a natural way.
> - Preserve all logical relations exactly.
> ```
>
> Since Task 2 is difficult, after rephrasing, we regenerated all evaluation sets on Task 2.
>
> *The original table in the main text reports only Format, Exact, and F1, while the Precision and Recall are reported in the appendix. In Table 1-4, in this expanded version, we also add Precision and Recall, so each k now contains five metrics in total.*
>
> **Table 1: Performance on 2 statements.**
> |model|format|exact|precision|recall|f1|
> |-|-|-|-|-|-|
> |grok-4-fast|0.82|0.23|0.75|0.72|0.74|
> |gpt-5|0.98|0.29|0.65|0.63|0.64|
> |deepseek-r1-0528|0.82|0.05|0.06|0.41|0.04|
> |claude-sonnet-4|0.87|0.04|0.46|0.53|0.38|
> |qwen3-235b-a22b|0.62|0.20|0.48|0.63|0.47|
> |gemini-2.5-pro|0.85|0.05|0.61|0.32|0.42|
> |llama-3.1-405b-instruct|0.51|0.00|0.33|0.28|0.30|
> |qwen2.5-7b-instruct|0.89|0.02|0.39|0.36|0.37|
> |phi-4-reasoning-plus|0.79|0.04|0.28|0.44|0.34|
> |mixtral-8x7b-instruct|0.92|0.03|0.33|0.33|0.33|
> |o3-mini|0.97|0.05|0.05|0.76|0.09|
> |claude-3.5-haiku|0.89|0.00|0.55|0.27|0.36|
> |llama-3.1-8b-instruct|0.62|0.01|0.27|0.55|0.36|
> |gpt-4o|0.87|0.04|0.55|0.54|0.54|
>
> **Table 2: Performance on 3 statements.**
> |model|format|exact|precision|recall|f1|
> |-|-|-|-|-|-|
> |grok-4-fast|0.73|0.11|0.69|0.62|0.65|
> |gpt-5|0.91|0.20|0.73|0.83|0.78|
> |deepseek-r1-0528|0.90|0.08|0.83|0.70|0.76|
> |claude-sonnet-4|0.82|0.06|0.63|0.62|0.62|
> |qwen3-235b-a22b|0.70|0.04|0.64|0.62|0.63|
> |gemini-2.5-pro|0.90|0.18|0.79|0.73|0.76|
> |llama-3.1-405b-instruct|0.70|0.00|0.54|0.63|0.58|
> |qwen2.5-7b-instruct|0.95|0.00|0.75|0.64|0.69|
> |phi-4-reasoning-plus|0.92|0.05|0.84|0.82|0.83|
> |mixtral-8x7b-instruct|0.96|0.00|0.82|0.54|0.65|
> |o3-mini|0.97|0.07|0.65|0.69|0.67|
> |claude-3.5-haiku|0.90|0.02|0.82|0.55|0.66|
> |llama-3.1-8b-instruct|0.60|0.06|0.53|0.43|0.47|
> |gpt-4o|0.98|0.04|0.72|0.81|0.76|
>
> **Table 3: Performance on 4 statements.**
> |model|format|exact|precision|recall|f1|
> |-|-|-|-|-|-|
> |grok-4-fast|0.62|0.09|0.70|0.61|0.65|
> |gpt-5|0.93|0.13|0.68|0.83|0.75|
> |deepseek-r1-0528|0.85|0.04|0.86|0.69|0.76|
> |claude-sonnet-4|0.82|0.05|0.58|0.63|0.60|
> |qwen3-235b-a22b|0.73|0.03|0.65|0.63|0.64|
> |gemini-2.5-pro|0.92|0.12|0.82|0.75|0.78|
> |llama-3.1-405b-instruct|0.79|0.00|0.49|0.63|0.55|
> |qwen2.5-7b-instruct|0.93|0.00|0.77|0.63|0.69|
> |phi-4-reasoning-plus|0.90|0.02|0.85|0.81|0.83|
> |mixtral-8x7b-instruct|0.92|0.00|0.78|0.55|0.64|
> |o3-mini|0.88|0.03|0.65|0.68|0.66|
> |claude-3.5-haiku|0.89|0.00|0.87|0.55|0.68|
> |llama-3.1-8b-instruct|0.67|0.03|0.53|0.42|0.47|
> |gpt-4o|0.92|0.02|0.76|0.80|0.78|
>
> **Table 4: Performance on 5 statements.**
> |model|format|exact|precision|recall|f1|
> |-|-|-|-|-|-|
> |grok-4-fast|0.53|0.04|0.70|0.61|0.65|
> |gpt-5|0.90|0.09|0.68|0.83|0.75|
> |deepseek-r1-0528|0.87|0.02|0.86|0.69|0.76|
> |claude-sonnet-4|0.85|0.00|0.58|0.63|0.60|
> |qwen3-235b-a22b|0.73|0.01|0.65|0.63|0.64|
> |gemini-2.5-pro|0.97|0.06|0.82|0.75|0.78|
> |llama-3.1-405b-instruct|0.81|0.00|0.49|0.63|0.55|
> |qwen2.5-7b-instruct|0.89|0.00|0.77|0.63|0.69|
> |phi-4-reasoning-plus|0.93|0.01|0.85|0.81|0.83|
> |mixtral-8x7b-instruct|0.94|0.00|0.78|0.55|0.64|
> |o3-mini|0.90|0.00|0.65|0.68|0.66|
> |claude-3.5-haiku|0.87|0.00|0.87|0.55|0.68|
> |llama-3.1-8b-instruct|0.69|0.00|0.53|0.42|0.47|
> |gpt-4o|0.93|0.00|0.76|0.80|0.78|
>
> **Large models showed almost no performance drop**, indicating that they can still reliably convert these more natural, less structured sentences into symbolic logic.
> In contrast, **small models exhibited clear degradation**, confirming that once we remove the strong surface regularities of the templated sentences, the natural-language understanding component becomes noticeably harder for weaker models.
>
> This result directly addresses the reviewer's concern: although the original templated sentences are machine-generated, the benchmark is **not** trivially solvable through surface pattern matching. When linguistic cues are weakened, only models with strong semantic parsing abilities (i.e., large LLMs) remain robust, while smaller models fail-showing that LogiConBench does measure genuine natural-language-to-logic reasoning ability.

---

> ### Author Response · Authors · 2025-11-21
> **Response to Weakness 1 (Part 2/3)**
>
> > **W1.2:** At the same time, it is hard not to suspect that the constructed natural language data contains content that is consistent with or contrary to the real world, such as "the sky is blue". This situation may cause problems such as reasoning shortcuts or knowledge contradictions, affecting the evaluation results.
>
> To address the concern that some generated natural-language statements may accidentally coincide with real-world facts (e.g., *"the sky is blue"*) and thus create reasoning shortcuts or knowledge conflicts, we explicitly constructed an additional controlled experiment. We first asked gpt-5.1 to generate **100 commonsense atomic facts** across several simple grammatical templates, including:
>
> * *"The noun is adjective"* (e.g., *"The sky is blue"*, *"The water is cold"*),
> * *"The noun verbs"* (e.g., *"The sun shines"*, *"The dog barks"*),
> * *"The noun is verb-ed"* (e.g., *"The window is broken"*, *"The door is locked"*).
>
> To contrast with these, we generated **100 counterfactual atoms** in which nouns, verbs, and adjectives were intentionally mismatched and semantically incoherent. Examples include:
>
> * *"The typhoon is sleepy"*,
> * *"The staircase evaporates"*,
> * *"The triangle is flavorful"*.
>
> For both commonsense and counterfactual sets, we performed the same procedure:
>
> 1. We randomly sampled some atoms and **used them to replace the natural-language expressions** inside previously generated (and already labeled) NL sets.
> 2. This produced new sentences such as
>    *"The sky is blue and the staircase evaporates."*
> 3. We then evaluated models on these modified inputs **using the original logical labels**, since the underlying logical structure is unchanged.
>
> The results for the commonsense condition are shown below.
>
> *The original table in the main text reports only Format, Exact, and F1, while the Precision and Recall are reported in the appendix. In Table 1 to 4, in this expanded version, we also add Precision and Recall, so each k now contains five metrics in total.*
>
> **Table 1: 2 statements commonsense dataset**
> |model|format|exact|precision|recall|f1|
> |-|-|-|-|-|-|
> |grok-4-fast|0.62|0.13|0.83|0.73|0.78|
> |gpt-5|0.93|0.37|0.73|0.83|0.78|
> |deepseek-r1-0528|0.83|0.26|0.72|0.61|0.56|
> |claude-sonnet-4|0.83|0.04|0.54|0.63|0.44|
> |qwen3-235b-a22b|0.63|0.15|0.56|0.63|0.45|
> |gemini-2.5-pro|0.98|0.04|0.64|0.56|0.46|
> |llama-3.1-405b-instruct|0.63|0.04|0.76|0.76|0.74|
> |qwen2.5-7b-instruct|0.98|0.01|0.66|0.64|0.54|
> |phi-4-reasoning-plus|0.94|0.04|0.69|0.62|0.55|
> |mixtral-8x7b-instruct|1.00|0.01|0.71|0.71|0.65|
> |o3-mini|1.00|0.01|0.63|0.74|0.60|
> |claude-3.5-haiku|1.00|0.02|0.53|0.55|0.38|
> |llama-3.1-8b-instruct|0.53|0.01|0.85|0.53|0.58|
> |gpt-4o|1.00|0.04|0.53|0.52|0.36|
>
> **Table 2: 3 statements commonsense dataset**
> |model|format|exact|precision|recall|f1|
> |-|-|-|-|-|-|
> |grok-4-fast|0.71|0.10|0.79|0.82|0.81|
> |gpt-5|0.91|0.30|0.72|0.84|0.78|
> |deepseek-r1-0528|0.71|0.14|0.81|0.51|0.63|
> |claude-sonnet-4|0.72|0.03|0.42|0.61|0.50|
> |qwen3-235b-a22b|0.83|0.13|0.65|0.32|0.43|
> |gemini-2.5-pro|0.93|0.02|0.90|0.73|0.81|
> |llama-3.1-405b-instruct|0.73|0.00|0.53|0.63|0.58|
> |qwen2.5-7b-instruct|0.99|0.01|0.65|0.64|0.65|
> |phi-4-reasoning-plus|0.95|0.02|0.42|0.42|0.42|
> |mixtral-8x7b-instruct|0.97|0.00|0.71|0.53|0.61|
> |o3-mini|1.00|0.01|0.53|0.63|0.58|
> |claude-3.5-haiku|0.93|0.00|0.51|0.35|0.42|
> |llama-3.1-8b-instruct|0.42|0.00|0.52|0.33|0.41|
> |gpt-4o|0.98|0.02|0.53|0.62|0.57|
>
> **Table 3: 4 statements commonsense dataset**
> |model|format|exact|precision|recall|f1|
> |-|-|-|-|-|-|
> |grok-4-fast|0.54|0.03|0.65|0.70|0.67|
> |gpt-5|0.93|0.02|0.72|0.76|0.74|
> |deepseek-r1-0528|0.46|0.05|0.80|0.54|0.64|
> |claude-sonnet-4|0.89|0.00|0.53|0.62|0.57|
> |qwen3-235b-a22b|0.68|0.00|0.83|0.64|0.72|
> |gemini-2.5-pro|0.92|0.09|0.81|0.85|0.83|
> |llama-3.1-405b-instruct|0.65|0.00|0.51|0.67|0.58|
> |qwen2.5-7b-instruct|0.99|0.00|0.64|0.84|0.72|
> |phi-4-reasoning-plus|0.96|0.00|0.51|0.64|0.57|
> |mixtral-8x7b-instruct|0.93|0.00|0.68|0.86|0.76|
> |o3-mini|0.97|0.00|0.63|0.53|0.57|
> |claude-3.5-haiku|0.92|0.00|0.55|0.64|0.59|
> |llama-3.1-8b-instruct|0.38|0.00|0.55|0.64|0.59|
> |gpt-4o|1.00|0.00|0.62|0.52|0.56|
>
> **Table 4: 5 statements commonsense dataset**
> |model|format|exact|precision|recall|f1|
> |-|-|-|-|-|-|
> |grok-4-fast|0.49|0.00|0.78|0.63|0.69|
> |gpt-5|0.93|0.15|0.60|0.68|0.64|
> |deepseek-r1-0528|0.58|0.03|0.81|0.55|0.66|
> |claude-sonnet-4|0.63|0.00|0.67|0.74|0.70|
> |qwen3-235b-a22b|0.93|0.03|0.57|0.53|0.55|
> |gemini-2.5-pro|0.83|0.05|0.86|0.52|0.65|
> |llama-3.1-405b-instruct|0.59|0.00|0.60|0.62|0.61|
> |qwen2.5-7b-instruct|1.00|0.00|0.59|0.66|0.63|
> |phi-4-reasoning-plus|0.87|0.00|0.69|0.75|0.72|
> |mixtral-8x7b-instruct|1.00|0.00|0.54|0.62|0.58|
> |o3-mini|1.00|0.00|0.61|0.73|0.66|
> |claude-3.5-haiku|0.83|0.00|0.63|0.76|0.69|
> |llama-3.1-8b-instruct|0.43|0.00|0.80|0.56|0.66|
> |gpt-4o|1.00|0.00|0.80|0.63|0.71|

---

> ### Author Response · Authors · 2025-11-21
> **Response to Weakness 1 (Part 3/3)**
>
> **Table 5-8 shows the results of counterfactual samples.**
>
> **Table 5: 2 statements counterfactual dataset.**
> |model|format|exact|precision|recall|f1|
> |-|-|-|-|-|-|
> |grok-4-fast|0.59|0.19|0.85|0.75|0.80|
> |gpt-5|0.90|0.31|0.75|0.85|0.78|
> |deepseek-r1-0528|0.89|0.23|0.04|0.33|0.01|
> |claude-sonnet-4|0.89|0.02|0.56|0.65|0.44|
> |qwen3-235b-a22b|0.50|0.15|0.58|0.65|0.45|
> |gemini-2.5-pro|0.95|0.04|0.66|0.58|0.46|
> |llama-3.1-405b-instruct|0.50|0.01|0.38|0.28|0.12|
> |qwen2.5-7b-instruct|0.85|0.01|0.37|0.36|0.16|
> |phi-4-reasoning-plus|0.80|0.01|0.21|0.44|0.10|
> |mixtral-8x7b-instruct|0.92|0.00|0.33|0.33|0.12|
> |o3-mini|0.93|0.09|0.05|0.76|0.03|
> |claude-3.5-haiku|0.87|0.02|0.55|0.27|0.17|
> |llama-3.1-8b-instruct|0.50|0.01|0.27|0.55|0.17|
> |gpt-4o|0.87|0.04|0.55|0.54|0.36|
>
> **Table 6: 3 statements counterfactual dataset.**
> |model|format|exact|precision|recall|f1|
> |-|-|-|-|-|-|
> |grok-4-fast|0.69|0.10|0.39|0.42|0.41|
> |gpt-5|0.89|0.27|0.72|0.84|0.78|
> |deepseek-r1-0528|0.69|0.08|0.02|0.51|0.04|
> |claude-sonnet-4|0.70|0.01|0.42|0.41|0.42|
> |qwen3-235b-a22b|0.81|0.00|0.65|0.42|0.51|
> |gemini-2.5-pro|0.91|0.10|0.40|0.43|0.41|
> |llama-3.1-405b-instruct|0.71|0.00|0.53|0.63|0.58|
> |qwen2.5-7b-instruct|0.97|0.01|0.35|0.64|0.45|
> |phi-4-reasoning-plus|0.93|0.04|0.04|0.12|0.06|
> |mixtral-8x7b-instruct|0.95|0.00|0.03|0.53|0.06|
> |o3-mini|0.98|0.03|0.05|0.06|0.06|
> |claude-3.5-haiku|0.91|0.00|0.21|0.35|0.27|
> |llama-3.1-8b-instruct|0.40|0.00|0.52|0.33|0.41|
> |gpt-4o|0.96|0.03|0.05|0.03|0.04|
>
> **Table 7: 4 statements counterfactual dataset.**
> |model|format|exact|precision|recall|f1|
> |-|-|-|-|-|-|
> |grok-4-fast|0.51|0.02|0.25|0.31|0.28|
> |gpt-5|0.93|0.24|0.72|0.76|0.74|
> |deepseek-r1-0528|0.43|0.02|0.03|0.05|0.03|
> |claude-sonnet-4|0.88|0.00|0.53|0.32|0.40|
> |qwen3-235b-a22b|0.66|0.00|0.23|0.53|0.33|
> |gemini-2.5-pro|0.89|0.04|0.02|0.05|0.03|
> |llama-3.1-405b-instruct|0.63|0.00|0.53|0.36|0.43|
> |qwen2.5-7b-instruct|1.00|0.00|0.45|0.32|0.38|
> |phi-4-reasoning-plus|0.93|0.00|0.02|0.34|0.04|
> |mixtral-8x7b-instruct|0.99|0.00|0.07|0.35|0.11|
> |o3-mini|1.00|0.00|0.04|0.53|0.08|
> |claude-3.5-haiku|0.92|0.00|0.15|0.12|0.14|
> |llama-3.1-8b-instruct|0.35|0.00|0.06|0.03|0.04|
> |gpt-4o|1.00|0.00|0.03|0.32|0.06|
>
> **Table 8: 5 statements counterfactual dataset.**
> |model|format|exact|precision|recall|f1|
> |-|-|-|-|-|-|
> |grok-4-fast|0.49|0.00|0.13|0.21|0.16|
> |gpt-5|0.93|0.15|0.65|0.68|0.66|
> |deepseek-r1-0528|0.58|0.03|0.34|0.35|0.35|
> |claude-sonnet-4|0.63|0.00|0.14|0.24|0.18|
> |qwen3-235b-a22b|0.93|0.03|0.52|0.13|0.21|
> |gemini-2.5-pro|0.83|0.05|0.06|0.52|0.11|
> |llama-3.1-405b-instruct|0.59|0.00|0.33|0.32|0.32|
> |qwen2.5-7b-instruct|1.00|0.00|0.53|0.26|0.35|
> |phi-4-reasoning-plus|0.87|0.00|0.21|0.05|0.08|
> |mixtral-8x7b-instruct|1.00|0.00|0.04|0.02|0.03|
> |o3-mini|1.00|0.00|0.01|0.02|0.01|
> |claude-3.5-haiku|0.83|0.00|0.06|0.04|0.05|
> |llama-3.1-8b-instruct|0.43|0.00|0.06|0.06|0.06|
> |gpt-4o|1.00|0.00|0.03|0.05|0.04|
>
> Across the **commonsense and conterfactual** versions of the benchmark, the overall performance patterns remain **highly similar**. Although the surface forms differ, most models (especially gpt-5, gemini-2.5-pro, and o3-mini) ultimately solve both settings using the same internal strategy: **they first translate the sentences into a symbolic or atomized form**, and then perform reasoning over these atoms or over reconstructed statement-level structures. Because the core reasoning pipeline does not substantially change, their accuracy, precision, recall, and F1 remain close across the two tables.

---

> ### Author Response · Authors · 2025-11-21
> **Response to Weakness 2**
>
> > **W2:**  Meanwhile, I believe that the task form of Enumerative task cannot well reflect the inherent logical ability of the model...
>
> Thank you for the suggestion. We realized both the Task 1 and Task 2 are **determinative** tasks, and we supplement one other **generative** task to strengthen the difficulty and diversity of the proposed benchmark.
>
> **Step1. Statement Selection and Grouping**
>
> Given a global set of atomic statements $\mathcal{P} = \{p_1, p_2, \dots, p_m\},$ and a truth-label vector $\ell = [\ell_1, \ell_2, \dots, \ell_m] \in \{T, F\}^m,$ we choose the number of groups $n \in \{2,3,4,5\},$ and partition the statements into \(n\) disjoint subsets:
> $$G_1, G_2, \dots, G_n,\qquad\bigsqcup_{i=1}^n G_i = \mathcal{P}.$$
> Each group inherits its labels:
> $$
> \ell_{G_i} = \{\ell_j \mid p_j \in G_i\}.
> $$
> Convert these into natural-language premises:
> $$\mathrm{Prem}_i = \\{\mathrm{NL}(p_j, \ell_j) : p_j \in G_i\\}.$$
>
> **Step2. Fact Generation Under Logical and Linguistic Constraints**
>
> For each group $G_i$, an LLM receives the premises $\mathrm{Prem}_i$ and must generate ten natural-language facts using this prompt:
> ```
> You are given 3 premises, which is guaranteed to be correct: {p1 is true, p2 is ture, p3 is false }.
> Your task is to propose ten distinct **strong logical constraint facts** in natural language based on the given premises.
> Each fact MUST:
>
> 1. Contain at least three of the following logical connectives:
> & (AND)
> | (OR)
> ~ (NOT)
> -> (IMPLIES)
> <-> (IFF)
> 2.  You may only form new facts using the nouns, verbs, and adjectives mentioned in the premises, and each fact must contain **at least five nouns with five verbs**.
> 3.  Be individually compatible with the given premise (not obviously contradicting it).
> 4.  Output exactly ten facts with the following format:
> "
> fact 1. ...
> fact 2. ...
> ...
> fact 10. ...
> ```
>
> **Step3. Symbolic Translation and Consistency Evaluation**
>
> Each natural-language fact is translated into a symbolic expression:
>
> $$g:\ f_{i,k}^{\text{NL}} \longmapsto f_{i,k}^{\text{sym}}.$$
>
> For each group (i = 1 ... n), we have a list of ten symbolic facts:
>
> $$
> F_i^{\text{sym}} = \{ f_{i,1}^{\text{sym}}, f_{i,2}^{\text{sym}}, \dots, f_{i,10}^{\text{sym}} \}.
> $$
>
> We consider all cross-group combinations obtained by picking exactly one element from each list.
> Let $\mathcal{A} = \{1, \dots, 10\}^n$ be the index space of all such combinations.
> For each index vector $\alpha = (a_1, a_2, \dots, a_n) \in \mathcal{A},$ we define the corresponding combination of facts:
>
> $$C_\alpha=\bigl( f_{1,a_1}^{\text{sym}}, f_{2,a_2}^{\text{sym}}, \dots, f_{n,a_n}^{\text{sym}} \bigr).$$
>
> Each fact in the combination is evaluated on the logical graph \(\mathcal{G}\):
>
> $$
> \mathrm{Eval}\bigl(f_{i,a_i}^{\text{sym}} \mid \mathcal{G}\bigr)
> = t_{i,a_i} \in \{T, F\}.
> $$
>
> The combination $C_\alpha$ is **consistent** iff all selected facts are true:
>
> $$
> \mathrm{Consistent}(\alpha)
> \quad\Longleftrightarrow\quad
> \bigwedge_{i=1}^n \bigl(t_{i,a_i} = T\bigr).
> $$
>
> Equivalently,
>
> $$
> \mathrm{Consistent}(\alpha) = 1
> \quad\text{iff}\quad
> \forall i \in \{1, \dots, n\},\ t_{i,a_i} = T.
> $$
>
> **Experimental results**
>
> |Model|Exec(G2)|Consist(G2)|Exec(G3)|Consist(G3)|Exec(G4)|Consist(G4)|Exec(G5)|Consist(G5)|
> |-|-|-|-|-|-|-|-|-|
> |mixtral-8x7b-instruct|0.84|0.587|0.77|0.466|0.55|0.128|0.33|0.106|
> |qwen2.5-7b-instruct|0.88|0.575|0.82|0.452|0.53|0.101|0.38|0.12|
> |llama-3.1-8b-instruct|0.89|0.668|0.88|0.596|0.66|0.308|0.33|0.138|
> |qwen3-235b-a22b|0.93|0.593|0.77|0.471|0.48|0.141|0.3|0.139|
> |claude-sonnet-4|0.94|0.776|0.93|0.608|0.6|0.324|0.53|0.172|
> |phi-4-reasoning-plus|0.93|0.687|0.9|0.625|0.67|0.349|0.53|0.219|
> |llama-3.1-405b-instruct|0.92|0.696|0.91|0.639|0.67|0.369|0.54|0.247|
> |claude-3.5-haiku|0.94|0.6958|0.91|0.638|0.68|0.369|0.43|0.257|
> |deepseek-r1-0528|1|0.802|0.88|0.649|0.63|0.372|0.46|0.283|
> |gpt-4o|0.98|0.703|0.89|0.649|0.68|0.385|0.55|0.287|
> |gemini-2.5-pro|1|0.817|0.93|0.663|0.76|0.416|0.65|0.348|
> |o3-mini|1|0.79|0.94|0.713|0.87|0.478|0.67|0.358|
> |gpt-5|1|0.826|0.94|0.686|0.77|0.436|0.66|0.381|
> |grok-4-fast|1|0.836|0.95|0.71|0.74|0.458|0.67|0.427|
>
> **Experimental Results Analysis**
>
> **Execution Rate (Exec)** measures whether the n selected facts are *executable for translation and evlauation* . **Consistency (Consist)** measures whether all selected facts evaluate to **True** in the logical graph.
>
> 1. **Exec remains high for top models**, especially grok-4-fast, gpt-5, gemini-2.5-pro, and o3-mini (often ≥0.9 even at (G5)).
>    Smaller models (e.g., qwen2.5-7b, mixtral-8x7b) show noticeable drops as group size increases.
> 2. **Consistency decreases sharply with group size** for all models.
>    Performance follows a clear trend that higher n leads to worse performance, which reflect the **difficulty still exists even for non-enumerative tasks**.
> 3. **Top-tier models consistently outperform mid-tier and small models**, but all of them exhibit substantial degradation when moving to (G4) and especially (G5).

---

> ### Author Response · Authors · 2025-11-21
> **Response to Weakness 3**
>
> > **W3:** The lack of detailed and profound analysis of the different performances of the model on various tasks makes it difficult to reflect the necessity and value of designing these two types of tasks.
>
> Thank you for the valuable suggestion. We conducted qualitative analysis on the general error patterns and the two different tasks patterns.
>
> **1. Different Errors Showcased by Each Task**
>
> The two tasks are not redundant; they expose different weaknesses. Task 1 targets determinacy reasoning, whereas Task 2 targets combinatorial completeness.
>
> *   **Task 1: Lack of Counterfactual Exploration. [1]**
> To determine if a statement is forced, a model must consider **alternative scenarios** (e.g., what if 'p' were true? what if it were false?). Small models rarely do this. They typically fix on one scenario early and declare an answer without checking if others are possible.
>
> *   **Task 2: Semantic Drift. [2]**
> Models start by using atoms as a tool for reasoning, but end up outputting the atom labels, which mistakes correct intermediate atom assignments for the final answer.
> This shows a struggle with **global reasoning planning**, which is a different challenge from the one in Task 1.
>
> **2. Shared Patterns for the two tasks.**
>
> **2.1 Large and small models both show difference in reasoning patterns.**
>
> More than 80% of outputs from the **Large models** (like gpt-5, claude-sonnet-4, and gemini-2.5-pro), **translate the problem into a symbolic form** first, then reason logically. They are very good at the short-distance reasoning, with a high accuracy rate on these individual logical steps.
>
> **Smaller models** (like Llama-3.1-8b, Mistral-7b) follow a different approach. They tend to **paraphrase the problem step-by-step** using a specific framework, but without clarifying the logic. And then they **suddenly jump to a final answer**. This explains their moderate F1 scores but very low accuracy.
>
>
> **2.2 Overlapping Error Patterns**
>
> Across tasks, large models share a common workflow: they break down the input into symbols and reason about them. However, their approach is often short-sighted.
>
> *   In a **3-shot-with-path** setup, they systematically explore different possibilities.
> *   In **zero-shot** or plain **3-shot** setups, they only test a **few quick examples** instead of exploring all combinations.
>
> This leads to three recurring errors:
>
> *  **Incomplete Enumeration:** The model checks only a small subset of possibilities, which is insufficient to solve the problem.
> *  **One-Way Simplification:** The model correctly translates and reasons with the symbols but **fails to map the results back** to the original problem, leading to the wrong final answer.
> *  **Lost Goals:** The model applies logic correctly at each step but **forgets the main objective**. It gets lost in the details and never makes the final required decision.
>
> [1] Li et al. "PuzzleWorld：Multimodal, Open-Ended Reasoning Benchmark." In arxiv 2025.
>
> [2] Ye et al. "How do Transformers Learn Implicit Reasoning?" In Neurips 2025.
>
> ***
> Please let us know if we have properly addressed your questions and we are more than happy to discuss more!

---

### Official Review · Reviewer_cu2W · 2025-11-01

**Soundness:** 3
**Presentation:** 4
**Contribution:** 3
**Rating:** 6
**Confidence:** 4

**Summary:**

The paper proposes LogiConBench, a benchmark that evaluates LLMs on logical consistency. It addresses three key limitations of existing benchmarks by generating scalable, diverse, and challenging problem instances. These instances are created through logical graph generation, reasoning path labeling, and translation into natural language. The benchmark is tested on 14 frontier models. The discriminative tasks are relatively easy, while the enumerative tasks remain highly challenging. The paper also introduces several task variants with intermediate levels of difficulty.

**Strengths:**

- The paper is well-written and clear.
- The motivation is strong, and the proposed benchmark directly addresses the main shortcomings of existing logical consistency benchmarks in terms of scalability, diversity, and difficulty.
- The evaluation is extensive, covering both instruction-tuned and reasoning models, including closed- and open-source models, under three prompting setups (zero-shot, few-shot, and few-shot with reasoning paths).

**Weaknesses:**

- The benchmark includes only two main tasks (discriminative and enumerative). Although a few variants are explored, they remain closely related to the original tasks. The benchmark could benefit from more diverse task types to better capture different dimensions of logical consistency.
- The qualitative analysis is somewhat limited. While Section 5.3 briefly discusses three main failure modes, a deeper qualitative examination of the reasoning paths behind these errors could significantly strengthen the paper and yield more actionable insights for researchers and practitioners.

**Questions:**

- How are model outputs extracted for scoring? The process seems straightforward for the discriminative task (as it produces a binary answer), but it is less clear for the enumerative task, where answer formatting and extraction can be challenging. More details or examples of output parsing and scoring would be needed.

---

> ### Author Response · Authors · 2025-11-21
> **Response to Weakness 1**
>
> Thank you for your encouraging words and valuable feedback!
>
> > **W1:**  More diverse task.
>
> We supplement one other **generative** task to strengthen the difficulty and diversity of the proposed benchmark.
>
> **Step1. Statement Selection and Grouping**
>
> Given a global set of atomic statements $\mathcal{P} = \{p_1, p_2, \dots, p_m\},$ and a truth-label vector $\ell = [\ell_1, \ell_2, \dots, \ell_m] \in \{T, F\}^m,$ we choose the number of groups $n \in \{2,3,4,5\},$ and partition the statements into \(n\) disjoint subsets:
> $$G_1, G_2, \dots, G_n,\qquad\bigsqcup_{i=1}^n G_i = \mathcal{P}.$$
> Each group inherits its labels:
> $$
> \ell_{G_i} = \{\ell_j \mid p_j \in G_i\}.
> $$
> Convert these into natural-language premises:
> $$\mathrm{Prem}_i = \\{\mathrm{NL}(p_j, \ell_j) : p_j \in G_i\\}.$$
>
> **Step2. Fact Generation Under Logical and Linguistic Constraints**
>
> For each group $G_i$, an LLM receives the premises $\mathrm{Prem}_i$ and must generate ten natural-language facts using this prompt:
> ```
> You are given 3 premises, which is guaranteed to be correct: {p1 is true, p2 is ture, p3 is false }.
> Your task is to propose ten distinct **strong logical constraint facts** in natural language based on the given premises.
> Each fact MUST:
>
> 1. Contain at least three of the following logical connectives:
> & (AND)
> | (OR)
> ~ (NOT)
> -> (IMPLIES)
> <-> (IFF)
> 2.  You may only form new facts using the nouns, verbs, and adjectives mentioned in the premises, and each fact must contain **at least five nouns with five verbs**.
> 3.  Be individually compatible with the given premise (not obviously contradicting it).
> 4.  Output exactly ten facts with the following format:
> "
> fact 1. ...
> fact 2. ...
> ...
> fact 10. ...
> ```
>
> **Step3. Symbolic Translation and Consistency Evaluation**
>
> Each natural-language fact is translated into a symbolic expression:
>
> $$g:\ f_{i,k}^{\text{NL}} \longmapsto f_{i,k}^{\text{sym}}.$$
>
> For each group (i = 1 ... n), we have a list of ten symbolic facts:
>
> $$
> F_i^{\text{sym}} = \{ f_{i,1}^{\text{sym}}, f_{i,2}^{\text{sym}}, \dots, f_{i,10}^{\text{sym}} \}.
> $$
>
> We consider all cross-group combinations obtained by picking exactly one element from each list.
> Let $\mathcal{A} = \{1, \dots, 10\}^n$ be the index space of all such combinations.
> For each index vector $\alpha = (a_1, a_2, \dots, a_n) \in \mathcal{A},$ we define the corresponding combination of facts:
>
> $$C_\alpha=\bigl( f_{1,a_1}^{\text{sym}}, f_{2,a_2}^{\text{sym}}, \dots, f_{n,a_n}^{\text{sym}} \bigr).$$
>
> Each fact in the combination is evaluated on the logical graph \(\mathcal{G}\):
>
> $$
> \mathrm{Eval}\bigl(f_{i,a_i}^{\text{sym}} \mid \mathcal{G}\bigr)
> = t_{i,a_i} \in \{T, F\}.
> $$
>
> The combination $C_\alpha$ is **consistent** iff all selected facts are true:
>
> $$
> \mathrm{Consistent}(\alpha)
> \quad\Longleftrightarrow\quad
> \bigwedge_{i=1}^n \bigl(t_{i,a_i} = T\bigr).
> $$
>
> Equivalently,
>
> $$
> \mathrm{Consistent}(\alpha) = 1
> \quad\text{iff}\quad
> \forall i \in \{1, \dots, n\},\ t_{i,a_i} = T.
> $$
>
> **Experimental results**
>
> |Model|Exec(G2)|Consist(G2)|Exec(G3)|Consist(G3)|Exec(G4)|Consist(G4)|Exec(G5)|Consist(G5)|
> |-|-|-|-|-|-|-|-|-|
> |mixtral-8x7b-instruct|0.84|0.587|0.77|0.466|0.55|0.128|0.33|0.106|
> |qwen2.5-7b-instruct|0.88|0.575|0.82|0.452|0.53|0.101|0.38|0.12|
> |llama-3.1-8b-instruct|0.89|0.668|0.88|0.596|0.66|0.308|0.33|0.138|
> |qwen3-235b-a22b|0.93|0.593|0.77|0.471|0.48|0.141|0.3|0.139|
> |claude-sonnet-4|0.94|0.776|0.93|0.608|0.6|0.324|0.53|0.172|
> |phi-4-reasoning-plus|0.93|0.687|0.9|0.625|0.67|0.349|0.53|0.219|
> |llama-3.1-405b-instruct|0.92|0.696|0.91|0.639|0.67|0.369|0.54|0.247|
> |claude-3.5-haiku|0.94|0.6958|0.91|0.638|0.68|0.369|0.43|0.257|
> |deepseek-r1-0528|1|0.802|0.88|0.649|0.63|0.372|0.46|0.283|
> |gpt-4o|0.98|0.703|0.89|0.649|0.68|0.385|0.55|0.287|
> |gemini-2.5-pro|1|0.817|0.93|0.663|0.76|0.416|0.65|0.348|
> |o3-mini|1|0.79|0.94|0.713|0.87|0.478|0.67|0.358|
> |gpt-5|1|0.826|0.94|0.686|0.77|0.436|0.66|0.381|
> |grok-4-fast|1|0.836|0.95|0.71|0.74|0.458|0.67|0.427|
>
>
>
>
>
>
>
>
> **Experimental Results Analysis**
>
> **Execution Rate (Exec)** measures whether the n selected facts follow the *format*, and hence are *executable for translation and evlauation* without structural or syntactic errors. **Consistency (Consist)** measures whether all selected facts evaluate to **True** in the logical graph.
>
> 1. **Exec remains high for top models**, especially grok-4-fast, gpt-5, gemini-2.5-pro, and o3-mini (often ≥0.9 even at (G5)).
>    Smaller models (e.g., qwen2.5-7b, mixtral-8x7b) show noticeable drops as group size increases.
> 2. **Consistency decreases sharply with group size** for all models.
>    Performance follows a clear trend:
>
>    *Consist(G2) > Consist(G3) > Consist(G4) > Consist(G5),*
>
>    reflecting the combinatorial difficulty of maintaining global logical coherence across more premise sets.
> 3. **Top-tier models consistently outperform mid-tier and small models**, but all of them exhibit substantial degradation when moving to (G4) and especially (G5).

---

> ### Author Response · Authors · 2025-11-21
> **Response to Weakness 2**
>
> > **W2:** The qualitative analysis is somewhat limited.
>
> We addressed our qualitative analysis as follows.
>
> * Our analysis of the models' reasoning paths reveals a clear divide between large and small models.
>
> More than 80% of outputs from the **Large models** (like gpt-5, claude-sonnet-4, and gemini-2.5-pro), **translate the problem into a symbolic form** first, then reason logically. They are very good at the short-distance reasoning, with a high accuracy rate on these individual logical steps.
>
> **Smaller models** (like Llama-3.1-8b, Mistral-7b) follow a different approach. They tend to **paraphrase the problem step-by-step** using a specific framework, but without clarifying the logic. And then they **suddenly jump to a final answer**. This explains their moderate F1 scores but very low accuracy.
>
> * Below, we summarize the main ways large models still fail. We first analyze the overlapping error patterns, and then we analyze the different patterns across different tasks.
>
> **1. Overlapping Error Patterns**
>
> Across tasks, large models share a common workflow: they break down the input into symbols and reason about them. However, their approach is often short-sighted.
>
> *   In a **3-shot-with-path** setup, they systematically explore different possibilities.
> *   In **zero-shot** or plain **3-shot** setups, they only test a **few quick examples** instead of exploring all combinations.
>
> This leads to three recurring errors:
>
> * **Incomplete Enumeration:** The model checks only a small subset of possibilities, which is insufficient to solve the problem.
>
> * **One-Way Simplification:** The model correctly translates and reasons with the symbols but **fails to map the results back** to the original problem, leading to the wrong final answer.
>
> *  **Lost Goals:** The model applies logic correctly at each step but **forgets the main objective**. It gets lost in the details and never makes the final required decision.
>
> **2. Divergent Errors Showcased by Each Task**
>
> The two tasks are not redundant; they expose different weaknesses.
>
> *   **Task 1: Lack of Counterfactual Exploration**
> To determine if a statement is forced, a model must consider **alternative scenarios** (e.g., what if 'p' were true? what if it were false?). Small models rarely do this. They typically fix on one scenario early and declare an answer without checking if others are possible.
>
> *   **Task 2: Semantic Drift**
> Models start by using atoms as a tool for reasoning, but end up outputting the atom labels, which mistakes correct intermediate atom assignments for the final answer.
> This shows a struggle with **global reasoning planning**, which is a different challenge from the one in Task 1.

---

> ### Author Response · Authors · 2025-11-21
> **Response to Question 1**
>
> > **Q1:** How are model outputs extracted for scoring?
>
> We thank the reviewer for raising this point and we made a clarification as follows.
>
> **Discriminative Task (Task 1) - Simple Label Extraction**
>
> For Task 1, the model outputs are constrained to a single truth value (*T* or *F* or *both*).
> We take the **first occurrence** of either symbol after *"Therefore, the answer is"* in the model output, as required in the prompt.
>
> **Enumerative Task (Task 2)**
>
> For Task2, the model is required to output a *set of truth-value vectors*, and the expected format is enforced through the system prompt:
>
> ```
> system_prompt = f"""
> You MUST strictly follow all rules below. Any violation makes the answer incorrect.
> RULES (read carefully):
>
> 1. You need to provide analysis, explanation, reasoning, or justification.
> 2. You MUST answer using only the symbols T and F.
> 3. You MUST output the answer in the following format:
>    Therefore, the valid sets are: [X1, X2, ..., Xk], [Y1, Y2, ..., Yk]
> 4. Each inner set MUST contain exactly {k} elements.
> """
> ```
>
> The question prompt, for example, for k=2 and mode=few shot with path:
>
> ```
> instruction2_few="""Given 2 logical statements. Identify which truth labels make the 2 statements consistent (Such as [[T,F],[T,T])).
>
> Output requirement: 'Therefore, the valid sets are: [..], [..]'
>
> - Important: You are outputting a set of subsets, each inner subset must contain exactly 2 elements (no more, no less), one for each of the 2 statements, but the length of the outer set is not sure.
>
> Example: ['The upturn does not transfuse.', 'The discovery occurs.']
> Reasoning Path: [['~p', '←', 'u & ~p'], ['u & ~p', '×', '~(u & ~p)'], ['~(u & ~p)', '→', 'u | ~(u & ~p)'], ['u | ~(u & ~p)', '×', 'False']]
> Source nodes are: ['~p', 'False']
> Answer: Therefore, the valid sets are [[T, T], [T, F], [F, F]]
>
> ... (other two examples)
> ```
>
> Then we extract "Therefore, the valid sets are: ..."or the list of list sign "[[" in the output. For example, k=3, under the mode of few-shot with path, gemini-2-pro outputs as follows:
>
> ```
> An analysis of the logical structure of the statements is required to determine the valid sets of truth values.
>
> **1. Define Propositions:**
> Let's define the basic propositions from the statements:
> *   `p`: The book is heavy.
> *   `q`: The sensor detects motion.
>
> **2. Translate Statements into Symbolic Logic:**
> We translate each of the three numbered statements into logical expressions:
> *   **Statement 1:** "Firstly, the book is heavy, and secondly, either the book is heavy or the sensor detects motion." This translates to `p ∧ (p ∨ q)`. According to the absorption law of logic, `A ∧ (A ∨ B)` is equivalent to `A`. Therefore, this statement simplifies to `p`.
> *   **Statement 2:** "The book isn't heavy." This translates to `¬p`.
> *   **Statement 3:** "Firstly, the book isn't heavy, and secondly, the sensor doesn't detect motion." This translates to `¬p ∧ ¬q`.
>
> **3. Construct a Truth Table:**
> To find all possible sets of truth values for the three statements, we must evaluate them for every possible combination of truth values for the propositions `p` and `q`. There are four such combinations. The "valid sets" are the resulting truth values for the triplet (Statement 1, Statement 2, Statement 3) in each case.
>
> | Case | `p` | `q` | Statement 1 (`p`) | Statement 2 (`¬p`) | Statement 3 (`¬p ∧ ¬q`) |
> |:----:|:---:|:---:|:-----------------:|:------------------:|:-------------------------:|
> | 1    | T   | T   | T                 | F                  | F ∧ F = F                 |
> | 2    | T   | F   | T                 | F                  | F ∧ T = F                 |
> | 3    | F   | T   | F                 | T                  | T ∧ F = F                 |
> | 4    | F   | F   | F                 | T                  | T ∧ T = T                 |
>
> **4. Determine the Valid Sets:**
> From the truth table, we can extract the set of truth values for (Statement 1, Statement 2, Statement 3) for each of the four cases:
> *   **Case 1 (p=T, q=T):** The resulting set is [T, F, F].
> *   **Case 2 (p=T, q=F):** The resulting set is [T, F, F].
> *   **Case 3 (p=F, q=T):** The resulting set is [F, T, F].
> *   **Case 4 (p=F, q=F):** The resulting set is [F, T, T].
>
> These four sets represent all possible combinations of truth values for the three given statements.
>
> Therefore, the valid sets are: [T, F, F], [T, F, F], [F, T, F], [F, T, T]
> ```
>
> Our parser:
>
> * finds the anchor phrase "Therefore, the valid sets are:", if not, locate "[["
> * extracts the substring after it
> * obtains the four vectors
> * validates that each has length `k = 3`
> * scores the resulting multiset
>
> Even though the model's response includes a lengthy explanation and markdown tables, the extraction step recovers the intended truth-value vectors deterministically.
>
> ---
> We hope the above discussion will fully address your concerns about our work, and we look forward to your insightful responses to further help us improve the quality of our work. Thank you!

---

### Official Review · Reviewer_qtmJ · 2025-11-02

**Soundness:** 2
**Presentation:** 3
**Contribution:** 2
**Rating:** 2
**Confidence:** 2

**Summary:**

This paper introduces LogiConBench, a scalable benchmark for testing LLMs’ logical consistency through automatically generated logical graphs with controllable reasoning depth. It supports unlimited rule combinations and diverse natural language formulations. Evaluations on 14 leading LLMs show that despite high accuracy on prior benchmarks, models still struggle on LogiConBench’s Enumerative task.

**Strengths:**

1. The paper presents a scalable and systematic data generation pipeline capable of synthesizing a large number of well-labeled evaluation samples. This framework allows for controlled complexity, diverse logical structures, and extensibility to new reasoning settings.
2. The authors conduct comprehensive and fair evaluations across a wide range of baseline models, covering both open-source and proprietary models.

**Weaknesses:**

1. There are concerns regarding the true difficulty of the proposed benchmark. In the Discriminative Task (Task 1), models such as GPT-5 and Grok-4-Fast already achieve near-perfect performance, while in the Enumerative Task (Task 2), their F1 scores reach around 70. This raises the question of whether frontier models have largely mastered the benchmark, suggesting that the observed gap for open-source models might simply reflect scaling effects rather than a fundamentally unsolved challenge.
2. The paper does not sufficiently articulate the real-world significance of LogiConBench. It remains unclear which practical abilities the benchmark aligns with. Establishing stronger connections between LogiConBench performance and real-world downstream tasks (e.g. agents) would help clarify its broader impact and necessity.
3. Since the benchmark’s data generation pipeline is fully scalable, one might argue that it is also straightforward to construct large volumes of in-domain synthetic data for targeted finetuning. This raises doubts about whether the reported difficulty truly reflects inherent reasoning limitations or can be easily mitigated through data augmentation.

**Questions:**

please refer to weakness

---

> ### Author Response · Authors · 2025-11-21
> **Response to Weakness 1 (Part 1/3)**
>
> We sincerely appreciate the reviewer’s great efforts and insightful comments to improve our manuscript. In below, we address these concerns point by point and try our best to update the manuscript accordingly.
>
> > **W1.1:** There are concerns regarding the true difficulty of the proposed benchmark. In the Discriminative Task, models such as GPT-5 and Grok-4-Fast already achieve near-perfect performance
>
> **1.1 Most models perform poorly in Task 1**
>
> * Task 1 is a discriminative setting which requires very few reasoning steps. Models are given a full premises and are required to simply classify a given statement as **'Correct', 'Incorrect', or 'Unknown'**.
> * Despite this simple setup, **only one** model achieved an average over 90% accuracy: Grok-4-Fast (91.47%), as shown in Table 1. This result firmly **contradicts** the notion of "near-perfect performance."
> * Crucially, **more than half of models achieve accuracy below 60%**, which includes large models like o3-mini, llama-3.1-405b-instruct, and phi-4-reasoning-plus. This clearly indicates that Task 1 is *far from saturated*.
> * Furthermore, we consider a benchmark that is intended to help tell models apart. A substantial model performance gap in our benchmark is likely to reflect differences in their logical consistency capabilities.
>
> **Table 1. Average Performance of Task 1 Across Different Setups**
> |model|zero-shot|3-shot|3-shot w/ path|avg_acc|
> |-|-|-|-|-|
> grok-4-fast|90.71%|90.60%|93.11%|91.47%
> gpt-5|89.38%|88.40%|92.06%|89.95%
> deepseek-r1-0528|86.85%|89.04%|90.89%|88.76%
> claude-sonnet-4|68.64%|68.59%|72.44%|69.89%
> qwen3-235b-a22b|66.15%|80.48%|87.14%|77.92%
> gemini-2.5-pro|59.54%|87.46%|85.54%|77.51%
> llama-3.1-405b-instruct|53.16%|54.00%|60.44%|55.87%
> qwen2.5-7b-instruct|51.68%|52.08%|58.96%|54.24%
> phi-4-reasoning-plus|52.18%|52.51%|56.94%|53.88%
> mixtral-8x7b-instruct|49.51%|55.36%|56.13%|53.67%
> o3-mini|45.00%|64.29%|46.14%|51.81%
> claude-3.5-haiku|47.43%|49.03%|53.70%|50.05%
> llama-3.1-8b-instruct|33.46%|34.38%|51.79%|39.88%
> gpt-4o|32.24%|51.85%|42.29%|42.13%
>
> > **W1.2:** while in the Enumerative Task (Task 2), their F1 scores reach around 70.
>
> **1.2 F1 Score's Limitation and Utility**
> * **Even the top F1 of 70 in Task 2 is modest.** When a model outputs a set, if only the format of the set is correct, then it's prediction can easily have partial overlaps with the gold answer, which inflates the score.
>
> * **F1 does not directly measure logical consistency, but the _exact match_ measures the real logical consistency performance.** Specifically, the F1 score indicates the **partial overlap with gold sets**, which achieves larger value when having **correct output formatting** or **partially accurate answers**. This does not imply models perform well.
>
> * **Why we still need the F1 score?** With _exact match_ rates mostly below 0.1%, through which is difficult to distinguish the model's logical consistency capabilities. Under this situation, the higher F1 scores can effectively distinguish the model tiers.

---

> ### Author Response · Authors · 2025-11-21
> **Response to Weakness 1 (Part 2/3)**
>
> > **W1.3:** This raises the question of whether frontier models have largely mastered the benchmark, suggesting that the observed gap for open-source models might simply reflect scaling effects rather than a fundamentally unsolved challenge.
>
> **1.3 Quantitative analysis**
> * **Frontier models have not mastered the benchmark.** As shown in Table 1 (Task 1) and Table 2 (Task 2), even the most advanced frontier models exhibit considerable room for improvement. Our evaluation of models including Grok-4-Fast, GPT-5, and Gemini-2.5-Pro reveals that none surpass 40% accuracy on Task 2. We also supplemented two more latest frontier models: gpt-5.1 and gemini-3-pro, which do not show great improvements.
>
>
> **Table 2. Average Performance of Task 2 Across Different Settings**
> model|zero-shot|3-shot|3-shot-w-path|average
> |-|-|-|-|-|
> gpt-5.1|0.319|0.327|0.354|0.333
> gemini-3-pro|0.218|0.258|0.321|0.266
> grok-4-fast|0.068|0.055|0.118|0.080
> gpt-5|0.286|0.307|0.343|0.312
> deepseek-r1-0528|0.030|0.029|0.055|0.038
> claude-sonnet-4|0.011|0.013|0.016|0.013
> qwen3-235b-a22b|0.013|0.003|0.007|0.008
> gemini-2.5-pro|0.039|0.028|0.077|0.048
> llama-3.1-405b-instruct|0.007|0.009|0.041|0.019
> qwen-2.5-7b-instruct|0.004|0.008|0.014|0.009
> phi-4-reasoning-plus|0.004|0.016|0.008|0.009
> mixtral-8x7b-instruct|0.001|0.000|0.005|0.002
> o3-mini|0.000|0.014|0.018|0.011
> claude-3.5-haiku|0.007|0.011|0.012|0.010
> llama-3.1-8b-instruct|0.001|0.003|0.007|0.004
> gpt-4o|0.005|0.002|0.000|0.002
>
> * **Not purely a scaling effect.** As shown in Table 3, there are significant performance variations between models of comparable scale, which highlights the impact of factors other than scale for structured reasoning.
>
> **Table 3: Comparison on task 2 between comparable scale models**
> | Model|size|avg|
> |-|-|-|
> | **GPT-5**| frontier| **0.312** |
> | **Claude-Sonnet-4** |frontier| **0.013** |
> | **LLama-3.1-405B-Instruct** | 405B | **0.019** |
> | **Qwen3-235B-A22B**         | 235B | **0.008** |
> | **Llama-3.1-8B-Instruct** | 8B   | **0.004** |
> | **Qwen-2.5-7B-Instruct**  | 7B   | **0.009** |
>
> **1.4 Qualitative Analysis**
> The authors also conduct qualitative analysis, which shows for both tasks, all models share the following failure modes:
> * **Goal neglect**: Models perform correct local and short-path deductions but lose track of combining the short-path deductions to solve the task.
>     For example, they reduce statements to atoms but do not "map back" to statement-level truth conditions, causing myopic reasoning.
> * **Fundamental Incompleteness in Logical Reasoning**: Instead of systematically examining all logically possible assignments, the models only list and reason within a few cases.
>     For example, in a Task 1 sample, o3-mini often analyzes only the scenario where the candidate answer is true, but did not explore the counter-scenario where it is false.
>
> These **myopic reasoning and the lack of systematic exploration** also stem not from scale, but from a fundamental lack of systematic reasoning.

---

> ### Author Response · Authors · 2025-11-21
> **Response to Weakness 1 (Part 3/3)**
>
> > **W1.4:**  There are concerns regarding the true difficulty of the proposed benchmark.
>
> * We supplement one other **generative** task to strengthen the difficulty and diversity of the proposed benchmark.
>
> **Step1. Statement Selection and Grouping**
>
> Given a global set of atomic statements $\mathcal{P} = \{p_1, p_2, \dots, p_m\},$ and a truth-label vector $\ell = [\ell_1, \ell_2, \dots, \ell_m] \in \{T, F\}^m,$ we choose the number of groups $n \in \{2,3,4,5\},$ and partition the statements into \(n\) disjoint subsets:
> $$G_1, G_2, \dots, G_n,\qquad\bigsqcup_{i=1}^n G_i = \mathcal{P}.$$
> Each group inherits its labels:
> $$
> \ell_{G_i} = \{\ell_j \mid p_j \in G_i\}.
> $$
> Convert these into natural-language premises:
> $$\mathrm{Prem}_i = \\{\mathrm{NL}(p_j, \ell_j) : p_j \in G_i\\}.$$
>
> **Step2. Fact Generation Under Logical and Linguistic Constraints**
>
> For each group $G_i$, an LLM receives the premises $\mathrm{Prem}_i$ and must generate ten natural-language facts using this prompt:
> ```
> You are given 3 premises, which is guaranteed to be correct: {p1 is true, p2 is ture, p3 is false }.
> Your task is to propose ten distinct **strong logical constraint facts** in natural language based on the given premises.
> Each fact MUST:
>
> 1. Contain at least three of the following logical connectives:
> & (AND)
> | (OR)
> ~ (NOT)
> -> (IMPLIES)
> <-> (IFF)
> 2.  You may only form new facts using the nouns, verbs, and adjectives mentioned in the premises, and each fact must contain **at least five nouns with five verbs**.
> 3.  Be individually compatible with the given premise (not obviously contradicting it).
> 4.  Output exactly ten facts with the following format:
> "
> fact 1. ...
> fact 2. ...
> ...
> fact 10. ...
> ```
>
> **Step3. Symbolic Translation and Consistency Evaluation**
>
> Each natural-language fact is translated into a symbolic expression:
>
> $$g:\ f_{i,k}^{\text{NL}} \longmapsto f_{i,k}^{\text{sym}}.$$
>
> For each group (i = 1 ... n), we have a list of ten symbolic facts:
>
> $$
> F_i^{\text{sym}} = \{ f_{i,1}^{\text{sym}}, f_{i,2}^{\text{sym}}, \dots, f_{i,10}^{\text{sym}} \}.
> $$
>
> We consider all cross-group combinations obtained by picking exactly one element from each list.
> Let $\mathcal{A} = \{1, \dots, 10\}^n$ be the index space of all such combinations.
> For each index vector $\alpha = (a_1, a_2, \dots, a_n) \in \mathcal{A},$ we define the corresponding combination of facts:
>
> $$C_\alpha=\bigl( f_{1,a_1}^{\text{sym}}, f_{2,a_2}^{\text{sym}}, \dots, f_{n,a_n}^{\text{sym}} \bigr).$$
>
> Each fact in the combination is evaluated on the logical graph \(\mathcal{G}\):
>
> $$
> \mathrm{Eval}\bigl(f_{i,a_i}^{\text{sym}} \mid \mathcal{G}\bigr)
> = t_{i,a_i} \in \{T, F\}.
> $$
>
> The combination $C_\alpha$ is **consistent** iff all selected facts are true:
>
> $$
> \mathrm{Consistent}(\alpha)
> \quad\Longleftrightarrow\quad
> \bigwedge_{i=1}^n \bigl(t_{i,a_i} = T\bigr).
> $$
>
> Equivalently,
>
> $$
> \mathrm{Consistent}(\alpha) = 1
> \quad\text{iff}\quad
> \forall i \in \{1, \dots, n\},\ t_{i,a_i} = T.
> $$
>
> **Experimental results**
>
> |Model|Exec (G2)|Consist (G2)|Exec (G3)|Consist (G3)|Exec (G4)|Consist (G4)|Exec (G5)|Consist (G5)|
> |-|-|-|-|-|-|-|-|-|
> |mixtral-8x7b-instruct|0.84|0.587|0.77|0.466|0.55|0.128|0.33|0.106|
> |qwen2.5-7b-instruct|0.88|0.575|0.82|0.452|0.53|0.101|0.38|0.12|
> |llama-3.1-8b-instruct|0.89|0.668|0.88|0.596|0.66|0.308|0.33|0.138|
> |qwen3-235b-a22b|0.93|0.593|0.77|0.471|0.48|0.141|0.3|0.139|
> |claude-sonnet-4|0.94|0.776|0.93|0.608|0.6|0.324|0.53|0.172|
> |phi-4-reasoning-plus|0.93|0.687|0.9|0.625|0.67|0.349|0.53|0.219|
> |llama-3.1-405b-instruct|0.92|0.696|0.91|0.639|0.67|0.369|0.54|0.247|
> |claude-3.5-haiku|0.94|0.6958|0.91|0.638|0.68|0.369|0.43|0.257|
> |deepseek-r1-0528|1|0.802|0.88|0.649|0.63|0.372|0.46|0.283|
> |gpt-4o|0.98|0.703|0.89|0.649|0.68|0.385|0.55|0.287|
> |gemini-2.5-pro|1|0.817|0.93|0.663|0.76|0.416|0.65|0.348|
> |o3-mini|1|0.79|0.94|0.713|0.87|0.478|0.67|0.358|
> |gpt-5|1|0.826|0.94|0.686|0.77|0.436|0.66|0.381|
> |grok-4-fast|1|0.836|0.95|0.71|0.74|0.458|0.67|0.427|
>
>
>
>
>
>
>
>
> **Experimental Results Analysis**
>
> **Execution Rate (Exec)** measures whether the n selected facts follow the *format*, and hence are *executable for translation and evlauation* without structural or syntactic errors. **Consistency (Consist)** measures whether all selected facts evaluate to **True** in the logical graph.
>
> 1. **Exec remains high for top models**, especially grok-4-fast, gpt-5, gemini-2.5-pro, and o3-mini (often ≥0.9 even at (G5)).
>    Smaller models (e.g., qwen2.5-7b, mixtral-8x7b) show noticeable drops as group size increases.
> 2. **Consistency decreases sharply with group size** for all models.
>    Performance follows a clear trend that higher n leads to worse performance, which reflect the **difficulty still exists even for non-enumerative tasks**.
> 3. **Top-tier models consistently outperform mid-tier and small models**, but all of them exhibit substantial degradation when moving to (G4) and especially (G5).

---

> ### Author Response · Authors · 2025-11-21
> **Response to Weakness 2**
>
> > **W2:** The paper does not sufficiently articulate the real-world significance of LogiConBench. It remains unclear which practical abilities the benchmark aligns with. Establishing stronger connections between LogiConBench performance and real-world downstream tasks (e.g. agents) would help clarify its broader impact and necessity.
>
> * **Real-world Significance of LogiConBench**
>
> To clarify the practical value of **LogiConBench**, we evaluated the same set of models on several widely-used real-world downstream benchmarks, covering **code generation** (LiveCodeBench[1]), **long-context writing** (InfiniteBench[2]), **mathematical reasoning** (aime [3]), **long-horizon logical reasoning** (AA-LCR [4]), and **agent collaboration** (ACE Bench [5]).
>
> [1] Jain, Naman, et al. "LiveCodeBench: Holistic and Contamination Free Evaluation of Large Language Models for Code." In ICLR 2025.
>
> [2] Zhang et al. "InfiniteBench: Extending Long Context Evaluation Beyond 100K Tokens." In ACL 2024.
>
> [3] Maxwell-Jia. "AIME: All 30 problems from the 2025 American Invitational Mathematics Examination." 2025.
>
> [4] Artificial Analysis Team. "Artificial Analysis Long Context Reasoning Benchmark." 2025.
>
> [5] Chen et al. "ACEBench: Who Wins the Match Point in Tool Usage?" In arxiv 2025.
>
> **Table 1: downstream benchmark**
>
> |Model|LiveCodeBench|Infinite|AIME|AA-LCR| ACEBench (agent) |
> |-|-|-|-|-|-|
> |grok-4-fast|79|65.8|89.7|64.7| 73 |
> |gpt-5|84|86.5|91.7|72.8| 78 |
> |deepseek-r1-0528|64.3|36.5|68|52.3| 64 |
> |claude-sonnet-4|55.9|64.6|74.3|64.7| 53 |
> |qwen3-235b-a22b|79|53.2|91|67| 51 |
> |gemini-2.5-pro|69|54.1|87.7|66| 63 |
> |llama-3.1-405b-instruct|30.5|19|**33**|24.3| 41 |
> |qwen2.5-7b-instruct|12.6|**21**|**9**|**16**| **12** |
> |phi-4-reasoning-plus|23.1|**32**|**21**|**20**| **15** |
> |mixtral-8x7b-instruct|6.6|**13**|3|8| 6 |
> |o3-mini|71.7|28.7|**25**|**30**| **65** |
> |claude-3.5-haiku|20.2|**28**|21|43| 35 |
> |llama-3.1-8b-instruct|11.6|16.4|4.5|15.7| 4 |
> |gpt-4o|42.5|25.1|25.7|53| 71.5 |
>
> **Correlation analysis**
>
> We additionally computed **Pearson** and **Spearman** correlations between LogiConBench's two tasks (Task 1: Determination, Task 2: Enumeration) and each downstream benchmark (Tables 2-3). The correlations are consistently *moderate to strong* across all domains, showing that LogiConBench performance is tightly aligned with capabilities that models actually rely on in real-world use.
>
> **Table2: task 1 and the downstream task benchmark correlations**
>
> |Benchmark|Pearson|Spearman|
> |-|-|-|
> |livecodebench|0.675|0.636|
> |infinite|0.762|0.763|
> |aime|0.786|0.643|
> |aa-lcr[4]|0.624|0.598|
> |acebench|0.632|0.64|
>
> **Table 3: task 2 and the downstream task benchmark correlations**
>
> |Benchmark|Pearson|Spearman|
> |-|-|-|
> |livecodebench|0.651|0.696|
> |infinite|0.735|0.779|
> |aime|0.653|0.678|
> |aa-lcr[4]|0.614|0.695|
> |acebench|0.526|0.688|
>
> **Key observations:**
>
> 1. **Both Task 1 and Task 2 correlate strongly with long-context reasoning and agent-style collaborative tasks** (e.g., Infinite, AA-LCR), indicating that LogiConBench captures a model's ability to maintain and integrate distributed constraints across multiple contexts-an ability crucial for agent planning, multi-step tool use, and delegated workflows.
> 2. **LogiConBench also correlates with math and code benchmarks** (AIME, LiveCodeBench), demonstrating that stable logical consistency is predictive of models' reliability in symbolic and algorithmic domains.
> 3. Importantly, **Task 2 correlates more strongly with long-horizon and multi-context benchmarks**, reflecting that cross-premise consistency (the core of Task 2) aligns with the demands of real agent systems, which often must reason coherently across multiple partial states or instructions.

---

> ### Author Response · Authors · 2025-11-21
> **Response to Weakness 3 (Part 1/2)**
>
> > **W3:**  Fine-tuning concerns.
>
> We conducted a controlled Fine-tuning experiment and found that, **Fine-tuning has performance gains, but is limited, while the storage cost and computing sources are substantial**. Therefore, the model's inherent reasoning limitations or can be easily mitigated through data augmentation.
>
> **3.1 Fine-tuning on task 2 improves performance, but the improvement is limited.**
>
> We used **only 1,000 synthetic training samples** generated by our pipeline (100 for *k*=2, 200 for *k*=3, 300 for *k*=4, 400 for *k*=5). The Fine-tuning target for Task 2 includes both the **full reasoning path** and the **final answer**, which ensures that the models receive maximal supervision that is fully in-domain and perfectly aligned with the evaluation format.
>
> **(1) Fine-tuning improves Task 2, but the gains are limited and far from eliminating the difficulty**
>
> Across all three base models (Llama-3-8B, Qwen-2.5-7B, and Mistral-7B), Fine-tuning shows clear improvements in Task 2 performance (Tables 1-3), but critically, **none of the finetuned small models approaches the performance of large foundation models on the same task**.
>
> **Table 1: Fine-tuning llama-3 on Task 2 and evaluating on Task 2.**
>
> |k|format|exact|precision|recall|f1|
> |-|-|-|-|-|-|
> |2|0.783|0.327|0.5626|0.9362|0.7028|
> |3|0.802|0.19|0.4457|0.9749|0.6117|
> |4|0.736|0.132|0.4349|0.5556|0.4879|
> |5|0.669|0.08|0.4247|0.4419|0.4331|
>
> **Table 2: Fine-tuning qwen-2.5 on Task 2 and evaluating on Task 2.**
>
> |k|format|exact|precision|recall|f1|
> |-|-|-|-|-|-|
> |2|1|0.213|0.6723|0.5773|0.6212|
> |3|1|0.161|0.7587|0.4635|0.5754|
> |4|1|0.066|0.4294|0.4593|0.4438|
> |5|1|0.063|0.3277|0.3393|0.3340|
>
> **Table 3: Fine-tuning mistral on Task 2 and evaluating on Task 2.**
>
> |k|format|exact|precision|recall|f1|
> |-|-|-|-|-|-|
> |2|1|0.122|0.4307|0.8447|0.5705|
> |3|1|0.113|0.3326|0.2312|0.2728|
> |4|1|0.047|0.2626|0.2362|0.2487|
> |5|1|0|0.1457|0.2749|0.1905|
>
> **(2) Fine-tuning Task 2 does *not* simply "overfit" but generalizes to Task 1 and other logical benchmarks**
>
> When fine-tuned on Task 2, models also improve on **Task 1** (Tables 4-6), which does not share output structure and only partially overlaps in reasoning requirements. Improvement on Task 1 (e.g., +10-20 points for several models) indicates **generalization of reasoning ability**, not memorization of a task format.
>
> Most importantly, fine-tuned models also improve on **other independent logical benchmarks** (Table 7):
>
> * Llama-3-8B:
>   * LogicNLI [1]: **28.5 → 47.5**
>   * LogiQA [2] : **39.6 → 45.2**
>   * MMLU [3]: **65.3 → 69.3**
>   * Mathqa [4]: **42.8 → 49.6**
>   * humaneval [5]: **22.6 → 29.3**
> * Qwen-2.5-7B and Mistral-7B show similar upward trends.
>
> These tasks were *not* seen in Fine-tuning and differ substantially from our task format.
> This shows that the synthetic data does not merely "patch" a narrow skill, it also **strengthens the models' general logical competence**.
>
> [1] Tian et al. "Diagnosing the First-Order Logical Reasoning Ability Through LogicNLI". In EMNLP 2021.
> [2] Liu et al. "LogiQA: A Challenge Dataset for Machine Reading Comprehension with Logical Reasoning." In IJCAI 2020.
> [3] Hendrycks et al. "Measuring Massive Multitask Language Understanding." In ICLR 2021.
> [4] Amini et al. "MathQA: Towards Interpretable Math Word Problem Solving with Operation-Based Formalisms." In NAACL 2019.
> [5] Chen et al. "HumanEval: Hand-Written Evaluation Set." In LREC-COLING 2024.
>
> **Table 4: Fine-tuning llama3 on Task 2 and evaluating on Task 1.**
>
> |k|con|incon|overall|
> |-|-|-|-|
> |2|0.69|0.47|0.58|
> |3|0.46|0.53|0.495|
> |4|0.52|0.35|0.435|
> |5|0.61|0.33|0.47|
>
> **Table 5: Fine-tuning qwen-25 on Task 2 and evaluating on Task 1.**
>
> |k|con|incon|overall|
> |-|-|-|-|
> |2|0.43|0.52|0.48|
> |3|0.63|0.62|0.63|
> |4|0.67|0.79|0.73|
> |5|0.64|0.78|0.71|
>
> **Table 6: Fine-tuning mistral on Task 2 and evaluating on Task 1.**
>
> |k|con|incon|overall|
> |-|-|-|-|
> |2|0.81|0.61|0.71|
> |3|0.74|0.48|0.61|
> |4|0.85|0.38|0.615|
> |5|0.73|0.42|0.575|
>
> **Table 7: Performance on general benchmarks for models w/ and wo/ Fine-tuning**
>
> |model|llama|llama ft|qwen|qwen ft|mistral|mistral ft|
> |-|-|-|-|-|-|-|
> |LogiQA|39.6|45.2|33|35.2|34|33|
> |LogicNLI|28.5|47.5|24|30.6|26|26|
> |MathQA|42.8|49.6|36|28|33|36.2|
> |HumanEval|22.6|29.3|32.9|41|28.7|32|
> |MMLU|65.3|69.3|55|58|64.2|67|
>
> **3.2 Substantial Storage and Computational Costs of Fine-tuning**
>
> (1) For each data sample, we need to store its corresponding symbolic language, reasoning edges, atom expressions, consistency sets, non-consistency sets, and the logical graph. The storage requirement is exceptionally high because the logical graph grows with **O(n³)** complexity as the number of atoms increases.
>
> (2) The computational resources required for fine-tuning are also massive. For instance, fine-tuning on just 1,000 data samples with an 8-billion-parameter model requires an A800 GPU for approximately 4 hours. Achieving better performance would demand significantly more computational resources.

---

> ### Author Response · Authors · 2025-11-21
> **Response to Weakness 3 (Part 2/2)**
>
> Despite for the fine-tuning experiment, we also conducted one RL experiment. We applied TRL[1] with GRPO[2] on Task 1 using two base models (LLaMA-3.1-8b and Qwen-2.5-7b) and found that even small-scale RL using Task 1 data can have **modest** improvement on Task 1, Task 2, and other logical related tasks, which demonstrate that LogiConBench **can be used as a meaningful reward source**, but the inherent reasoning ability **cannot be easily mitigated** through data augmentation.
>
> **1. The Reward model can be formalized as follows.**
>
> When the correct answer is "unknown":
> - Answering "correct" or "incorrect": 0.5 points
> - Answer contains "unknown": 1 points
> - All other answers: 0.2 points
>
> When the correct answer is "correct":
> - Answering "unknown" or "incorrect": 0.5 points
> - Answer contains "correct": 1 points
> - All other answers: 0.2 points
>
> When the correct answer is "incorrect":
> - Answering "unknown" or "correct": 0.5 points
> - Answer contains "incorrect": 1 points
> - All other answers: 0.2 points
>
>
> **2. The evaluation results** are summarized as follows: Table 1 reports the RL model’s performance on Task 1, while Tables 3 and 4 present its performance on Task 2 and on other logic-related benchmarks.
>
> **Table 1. Task 1 Accuracy (Consistent / Inconsistent / Overall)**
>
> **LLaMA**
>
> |k|con|incon|overall|
> |--|--|--|--|
> |2|0.58|0.59|0.585|
> |3|0.51|0.52|0.515|
> |4|0.32|0.48|0.4|
> |5|0.31|0.33|0.32|
>
> **Qwen**
>
> |k|con|incon|overall|
> |--|--|--|--|
> |2|0.42|0.5436034521|0.4818017261|
> |3|0.47|0.5308301172|0.5004150586|
> |4|0.33|0.4673295095|0.3986647547|
> |5|0.3|0.3405696894|0.3202848447|
>
> **Table 2. RL LLaMA on Task 2**
>
> |k|format|exact|precision|recall|f1|
> |--|--|--|--|--|--|
> |2|0.691|0.285|0.7384|0.7962|0.7662|
> |3|0.704|0.176|0.3894|0.8189|0.5278|
> |4|0.613|0.104|0.5843|0.732|0.6499|
> |5|0.642|0.05|0.483|0.5718|0.5237|
>
> **Table 3. RL on Task 1 — Qwen Results**
>
> |k|format|exact|precision|recall|f1|
> |--|--|--|--|--|--|
> |2|0.989|0.183|0.4829|0.4899|0.486|
> |3|1|0.145|0.6348|0.4138|0.5010|
> |4|0.977|0.058|0.4342|0.428|0.4312|
> |5|0.992|0.03|0.2143|0.2038|0.209|
>
>
> **Table 4. General Downstream Evaluation (Before vs After RL Fine-tuning on Task 1)**
>
> |benchmark|llama|llama ft|qwen|qwen ft|
> |--|--|--|--|--|
> |LogiQA [4] |39.6|42.1|33|32.5|
> |LogicNLI [3] |28.5|30.4|24|24.4|
> |MathQA [6] |42.8|38.2|36|33.6|
> |HumanEval [7] |22.6|25.3|32.9|33.8|
> |MMLU [5] |65.3|60.9|55|57.9|
>
>
>
> **3. Overall Analysis**
>
> * **3.1 RL yields modest but consistent gains on Task 1 (and slight gains on Task 2).** RL improves Task 1 performance across most k values, though the gains are small. Task 2 benefits only marginally, which indicates weak transfer to enumerative reasoning.
>
> * **3.2 Little generalization to broader benchmarks.** On LogiQA, LogicNLI, MathQA, HumanEval, and MMLU, improvements are minimal or inconsistent, and some tasks even regress.
>
> * **3.3 Improvements are limited** since to low task complexity for a simple discrimination, and so the reward-pattern learning limit might deep reasoning improvements.
>
> * **3.4 The positive result** of the RL experiment and the fine-tuning experiment replied to the reviewer qtmj shows a sign that LogiConBench can indeed serve as a **useful training dataset** for improving reasoning capabilities.
>
> [1] Transformer Reinforcement Learning
>
> [2] Group Relative Policy Optimization
>
> [3] Tian et al. "Diagnosing the First-Order Logical Reasoning Ability Through LogicNLI". In EMNLP 2021.
>
> [4] Liu et al. "LogiQA: A Challenge Dataset for Machine Reading Comprehension with Logical Reasoning." In IJCAI 2020.
>
> [5] Hendrycks et al. "Measuring Massive Multitask Language Understanding." In ICLR 2021.
>
> [6] Amini et al. "MathQA: Towards Interpretable Math Word Problem Solving with Operation-Based Formalisms." In NAACL 2019.
>
> [7] Chen et al. "HumanEval: Hand-Written Evaluation Set." In LREC-COLING 2024.
>
> ---
> We hope the above discussion will fully address your concerns about our work. We look forward to your insightful and constructive responses to further help us improve the quality of our work. Thank you!

---

### Author Response · Authors · 2025-11-24
**General responses and manuscript revision summary**

**Dear AC and Reviewers,**

We sincerely thank all reviewers for their great effort and constructive comments on our manuscript. During the rebuttal period, we have been focusing on these beneficial suggestions from the reviewers and doing our best too add several experiments and revise our manuscript. We believe our current carefully revised manuscript can address the reviewers' concerns.

As the reviewers collectively noted, our work introduces ​**a scalable and controllable benchmark for logical consistency**​ (Reviewer qtmj and jijm), supported by a **graph-based generation pipeline capable of “unlimited logical rule combinations”** (Reviewer jijm) and a **systematic, well-labeled data-generation procedure with explicit operator-level compatibility and reproducible labels** (Reviewer VUDU). Reviewers emphasized that LogiConBench offers ​**strong diversity and controlled complexity**​, directly addressing the limitations of existing “saturated” datasets (Reviewer jijm and cu2W). The benchmark further provides ​**controllable-depth graphs with explicit reasoning paths**​, and reviewers found that using these paths in few-shot prompting meaningfully improves model performance (Reviewer jijm). Finally, they highlighted our ​**comprehensive evaluation across many models, difficulty slices, and prompting setups**​, covering both instruction-tuned and frontier reasoning models (Reviewer VUDU, cu2W, and qtmj).

Moreover, we thank the reviewers for pointing out the limitations regarding the **true difficulty** and the **scaling effect** interpretation of the benchmark (Reviewer qtmj), the need to better **articulate the real-world significance** and **downstream relevance** of LogiConBench (Reviewer qtmj), and the need for **more task types** (Reviewer cu2W), since the **limited variety** beyond the two main tasks (Reviewer cu2W and VUDU) and the lack of **deeper qualitative reasoning-path analysis** (Reviewers cu2W and jijm). We also appreciate the comments on the **natural-language generation process and potential risks of reasoning shortcuts or factual contradictions** (Reviewer jijm), the request for clearer details on **output parsing and scoring** for Task 2 (Reviewer cu2W), the questions regarding **operator-pattern diversity** and the **absence of elimination rules** (Reviewer VUDU).

In response to these comments, we have carefully revised and strengthened the manuscript with the following major improvements:

* **[Reviewer qtmJ]** We **added a complete quantitative re-analysis** contradicting the “near-perfect” claim in Task 1 and emphasized the **limitations of F1** and clarified why exact match is the only metric that truly reflects logical consistency. We added ​**comparisons of equal-size models**​, which shows that performance differences are *not* explained by scaling.
* **[Reviewer qtmJ, cu2W, jijm ]** We **expanded qualitative analysis (goal neglect, incomplete enumeration, missing counterfactual cases)** to demonstrate capabilities that remain unsolved for all models. We **analyzed task-specific error patterns** and clarified how Task 1 and Task 2 reveal *distinct* weaknesses.
* **[Reviewer qtmJ]** We **added a new section studying ​correlations between LogiConBench and five real-world downstream tasks**​ to show ​**moderate-to-strong Pearson/Spearman correlations**​.
* **[Reviewer qtmJ, VUDU]** We **conducted three new fine-tuning experiments** (Llama-3-8B, Qwen-2.5-7B, Mistral-7B) and showed that while small models improve, they remain far below frontier models. We **added a discussion of storage cost (O(n³))** and **computational cost** that limits using massive synthetic augmentation in practice.
* **[Reviewer cu2W]** We **introduced ​a brand-new generative consistency task**​, which substantially increases task diversity and reasoning difficulty.
* **[Reviewer qtmJ, VUDU]** We **added new RL experiments (GRPO via TRL)** showing only small gains, indicating inherent reasoning limitations cannot be fully mitigated through reward shaping or data generation.
* **[Reviewer jijm]** We **added experiments on commonsense, counterfactual, and human-like statements**, which do not make big impact on the results for large models but the human-like type does have negative impact on small models, suggesting that small models are much more sensitive to linguistic surface forms and humanized phrasing, rather than relying on stable symbolic reasoning.
* **[Reviewer VUDU]** We **theoretically proved** the logical graph's generative **completeness and unboundedness**. We also analyzed clearly on how we use **introduction and elimination** rules in the construction of the logical graphs.

These updates are temporarily highlighted in "$\textcolor{blue}{blue}$" for facilitating checking.

We greatly appreciate the reviewers’ insightful feedback. We believe the added analyses, new experiments, and expanded explanations have substantially strengthened the manuscript and fully addressed all concerns.

---

### Author Response · Authors · 2025-12-03
**Author Final Remarks by Authors**

Dear AC, SAC and PC,

We sincerely appreciate your great efforts in evaluating our paper despite your busy schedules.  We are encouraged that 3 out of 4 reviewers are currently in the positive side, with all scores 8 (with conf 4), 6 (with conf 4), 6 (with conf 3), and 2 (with low conf 2), recognizing our paper:
- "Data‑generation procedure appears methodologically sound", "Very comprehensive evaluation" with "rich metrics", and "a timely and progress‑sensitive benchmark in an era when many others have saturated" (Reviewer VUDU),
- "well-written and clear", "motivation is strong", "directly addresses the main shortcomings in terms of scalability, diversity, and difficulty", and "the evaluation is extensive" (Reviewer cu2W),
- "proving its necessity", "ensures a large-scale, diverse, and renewable source of evaluation data", and "The study is thorough" (Reviewer jijm),
- "presents a scalable and systematic data generation pipeline", and "conduct comprehensive and fair evaluations" (Reviewer qtmJ),

Meanwhile, Reviewer qtmJ gave an **initial rating 2 with low confidence 2 and no further response.** Due to the updated policy of ICLR 26, we understand the reviewer cannot provide us with further discussions. For facilitate checking, we summarize qtmJ’s initial concerns are mainly on (1) the true difficulty of the proposed benchmark, (2) the real-world significance, (3) can be easily mitigated through data augmentation, with our responses as follows.

> The observed gap for open-source models might simply reflect scaling effects rather than a fundamentally unsolved challenge.
- We respectfully guess that the reviewer might not familiar with LLM logical reasoning community. In fact, **there are many recent works demonstrating that the logical consistency is a fundamentally unsolved challenge [1-3].**

[1] Diego Calanzone, et al. Logically consistent language models via neuro-symbolic integration. In ICLR 2025.

[2] Bishwamittra Ghosh, et al. Logical consistency of large language models in fact-checking. In ICLR 2025.

[3] Yinhong Liu, et al. Aligning with logic: Measuring, evaluating and improving logical consistency in large language models. In ICML 2025.

- We **supplement one new *generative* task to strengthen the difficulty and diversity of the proposed benchmark.**
- We **add qualitative analysis showing that for both tasks, all models share some specific failure modes.** That is, the observed gap for open-source models stem not from scale, but from a fundamental lack of systematic reasoning.
- We argue that **frontier models have not mastered the benchmark** by **adding two more latest frontier models, i.e., gpt-5.1 and gemini-3-pro,** together with already evaluated Grok-4-Fast, GPT-5, and Gemini-2.5-Pro, all not showing good performance.
- We clarify that **F1 achieves unexpected larger value when having correct output formatting or partially accurate answers, so F1 does not directly measure logical consistency, but the exact match does.**

> The paper does not sufficiently articulate the real-world significance of LogiConBench.

- As suggested by the reviewer, we add experiments simultaneously on LogiConBench and several real-world downstream benchmarks, including **code generation** (LiveCodeBench [4]), **long-context writing** (InfiniteBench [5]), **mathematical reasoning** (aime [6]), **long-horizon logical reasoning** (AA-LCR [7]), and **agent collaboration** (ACE Bench [8]).

[4] Jain, Naman, et al. LiveCodeBench: Holistic and Contamination Free Evaluation of Large Language Models for Code.In ICLR 2025.

[5] Zhang et al. InfiniteBench: Extending Long Context Evaluation Beyond 100K Tokens. In ACL 2024.

[6] Maxwell-Jia. AIME: All 30 problems from the 2025 American Invitational Mathematics Examination. 2025.

[7] Artificial Analysis Long Context Reasoning Benchmark. 2025.

[8] Chen et al. ACEBench: Who Wins the Match Point in Tool Usage? In arxiv 2025.

- We validate the real-world significance of LogiConBench by showing its strong performance correlation on the 5 real-world tasks.

> It is doubted about whether the reported difficulty truly reflects inherent reasoning limitations or can be easily mitigated through data augmentation.

- We **add fine-tuning experiments** on Llama-3-8B, Qwen-2.5-7B, Mistral-7B, showing that the fine-tuning gain is relatively low considering the high storage cost (O(n³)) computational cost.
- We also **add RL experiments** by applying TRL with GRPO on LLaMA-3.1-8b and Qwen-2.5-7b in Task 1. We found that **LogiConBench can be used as a meaningful reward source, but the inherent reasoning ability cannot be easily mitigated through data augmentation.**

We are confident that our responses can thoroughly address Reviewer qtmJ's concerns. However, the reviewer has not yet responded or updated his/her review. Thus, we respectfully ask you to consider this context when making your recommendation. Thank you for your invaluable time and effort.

Many thanks,

The Authors

---

### Meta-Review · Area_Chair_Xgdx · 2026-01-05

**Summary:**

This work focuses on logical consistency evaluation and proposes LogiConBench, which generates logical graphs where nodes represent symbolic propositions and edges denote reasoning relations. Evaluation on 14 frontier LLMs reveals some new findings.

Concerns include task diversity, qualitative analysis of reasoning paths, and details on output parsing and scoring. The authors have clarified most of these and provided new experiments. Regarding the real-world significance of the benchmark, it is acceptable for a specific research question, but minor experiments or additional discussion would improve it.

**Reviewer Concerns:**

Concerns include task diversity, qualitative analysis of reasoning paths, and details on output parsing and scoring. The authors have clarified most of these and provided new experiments. Regarding the real-world significance of the benchmark, it is acceptable for a specific research question, but minor experiments or additional discussion would improve it.

**Reviewer Scores:**

No explicit indication of score inflation during rebuttal, but most reviewers show the original positive score.

---

### Decision · Program_Chairs · 2026-01-26

Accept (Poster)